# Capping protein regulates endosomal trafficking by controlling F-actin density around endocytic vesicles and recruiting RAB5 effectors

**Dawei Wang[1,2†], Zuodong Ye[1,2†], Wenjie Wei[3†], Jingting Yu[1†], Lihong Huang[1,2], Hongmin Zhang[4], Jianbo Yue[1,2,5]***

[1]City University of Hong Kong Shenzhen Research Institute, Shenzhen, China; [2]Department of Biomedical Sciences, City University of Hong Kong, Hong Kong, China; [3]Core Research Facilities, Southern University of Science and Technology, Shenzhen, China; [4]Department of Biology, Guangdong Provincial Key Laboratory of Cell Microenvironment and Disease Research and Shenzhen Key Laboratory of Cell Microenvironment, Southern University of Science and Technology, Shenzhen, China; [5]City University of Hong Kong Chengdu Research Institute, Chengdu, China

**Abstract** Actin filaments (F-actin) have been implicated in various steps of endosomal trafficking, and the length of F-actin is controlled by actin capping proteins, such as CapZ, which is a stable heterodimeric protein complex consisting of α and β subunits. However, the role of these capping proteins in endosomal trafficking remains elusive. Here, we found that CapZ docks to endocytic vesicles via its C-terminal actin-binding motif. CapZ knockout significantly increases the F-actin density around immature early endosomes, and this impedes fusion between these vesicles, manifested by the accumulation of small endocytic vesicles in CapZ-knockout cells. CapZ also recruits several RAB5 effectors, such as Rabaptin-5 and Rabex-5, to RAB5-positive early endosomes via its N-terminal domain, and this further activates RAB5. Collectively, our results indicate that CapZ regulates endosomal trafficking by controlling actin density around early endosomes and recruiting RAB5 effectors.

**\*For correspondence:**
jianbyue@cityu.edu.hk

[†]These authors contributed equally to this work

**Competing interest:** The authors declare that no competing interests exist.

## Editor's evaluation

In this article, Yue and colleagues have uncovered role of the actin binding and capping protein CapZ in fusion and maturation of early endosomes. They show that the actin capping factor localizes to early endosomes and regulates F-actin density on early endosomes. The authors also find another role of CapZ in facilitating recruitment of Rabaptin-5 and Rabex-5 on early endosomes and promote Rab5 activation. The work provides an unexplored and somewhat unexpected perspective on how endosomal localized actin polymerization impact on endosome maturation and fusion.

## Introduction

Endosomal trafficking plays a vital role in various physiological functions, including proliferation, differentiation, immunity, and neurotransmission. Dysregulation of endosomal trafficking has been implicated in multiple human diseases, including autoimmune diseases, neurodegeneration, diabetes, and cancer (*Mendoza et al., 2014*; *Parachoniak and Park, 2012*; *Fletcher and Rappoport, 2010*). The trafficking, fusion, and sorting of the endosomes are tightly controlled by a complicated network of signaling molecules and cytoskeleton structures (*Epp et al., 2011*; *Elkin et al., 2016*). Among

the signaling molecules, recruitment, activation, and inactivation of different RABs play an essential role in controlling the identity and maturation of endosomes (*Barr and Lambright, 2010*; *Bhuin and Roy, 2014*; *Novick, 2016*). Likewise, the actin network has been implicated in almost every step of endosomal trafficking from internalization, trafficking, to maturation (*Simonetti and Cullen, 2019*; *Pol et al., 1997*; *Nakagawa and Miyamoto, 1998*; *Taunton et al., 2000*; *Huckaba et al., 2004*; *Gauthier et al., 2007*; *Morel et al., 2009*; *Muriel et al., 2016*; *Derivery et al., 2009*; *Mooren et al., 2012*; *Kaksonen et al., 2005*).

The RAB family contains more than 70 members, which are key regulators for intracellular trafficking, including endocytosis (*Barr and Lambright, 2010*; *Bhuin and Roy, 2014*; *Novick, 2016*). Like other small GTPases, RAB-GTP is active whereas RAB-GDP is inactive, and the switch from RAB-GDP to RAB-GTP and vice versa is catalyzed by guanine-nucleotide-exchange factors (GEFs) and GTPase-activating proteins (GAPs), respectively. GDP dissociation inhibitor (GDI) extracts inactive RAB-GDP from the membrane and escorts it to the cytosol, whereas GDP dissociation factor (GDF) can release RAB-GDP from GDI (*Bhuin and Roy, 2014*). Different RABs have their own specific GEFs and GAPs, and some RABs (e.g., RAB5) also contain intrinsic GTPase activity (*Müller and Goody, 2018*). RAB-GTP targets various effectors (e.g., tethering proteins, sorting molecules, and both protein and lipid kinases) to regulate vesicle trafficking. Among them, RAB5 is responsible for early endosome maturation; RAB4 and RAB11 are required for recycling endocytosis; RAB6 is involved in retrograde trafficking between early endosomes and the Golgi complex; and RAB7 is essential for the maturation of late endosomes and their subsequent fusion with lysosomes (*Müller and Goody, 2018*). All of these RABs are under strict temporal and spatial regulation, and in this way, they control endosomal trafficking. For example, the recruitment of RAB5 from the cytosol into endosomes and its subsequent activation promotes the maturation of early endosomes. However, the inactivation of RAB5 is followed by the recruitment and activation of RAB7, which is essential for the transition from early endosomes to late endosomes (*Rink et al., 2005*). The recruitment of RAB5 to endosomes is promoted by its GEF, Rabex-5 (also called RABEX5 or RABGEF1). The active form of RAB5 subsequently recruits more effectors, including Rabex-5, Rabaptin-5 (also called RABEP1), EEA1, and VPS34, to the endosomes. Here, they are activated and thus a positive feedback loop is established to induce early endosome maturation (*Langemeyer et al., 2018*). On the contrary, Mon1-Ccz1 replaces Rabex-5 on endosomes and inactivates RAB5; it simultaneously interacts with the HOPS complex to recruit and activate RAB7, and thereby promotes the early to late endosome transition (*Kinchen and Ravichandran, 2010*; *Poteryaev et al., 2010*). Although much progress has been made on the regulation of RABs during endocytosis, the detailed mechanisms on their temporal and spatial regulation still remain elusive.

When endocytosis is initiated, the clathrin-dependent or independent effectors are recruited to the invaginated plasma membrane. For clathrin-dependent endocytosis, as the membrane curvature increases to form a deeply invaginated pit, endocytosis assembly factors recruit actin nucleation factors (NPFs), for example, WASP, and this activates Arp2/3 to assemble actin filaments around the invaginated pit (*Merrifield et al., 2002*; *Robertson et al., 2009*). The scission between the invaginated pit and plasma membrane is executed by dynamin, and actin filament might provide additional forces to promote this process. Short-lived actin comet tails around the endocytic vesicle might propel the movement of the free vesicle away from the plasma membrane (*Taunton et al., 2000*; *Galletta and Cooper, 2009*). In addition to regulating this very early stage of endocytosis, actin polymerization driven by Arp2/3 complex and another NPF, WASH, on the surface of early endosomes regulate the endosome shape and tubulation (*Derivery et al., 2009*), and the length of the branches is controlled by actin capping proteins, for example, CapZ (*Mooren et al., 2012*; *Girao et al., 2008*).

CapZ is a stable heterodimeric protein complex (consisting of α and β subunits), which binds to the barbed ends of actin filaments to prevent further filament elongation (*Wear and Cooper, 2004*; *Casella et al., 1989*). In vertebrates, three CapZβ isoforms (β1, β2, and β3) are generated by the alternative splicing of one single gene, whereas CapZα is encoded by three distinct genes (α1, α2, and α3), with α3 being a germ cell-specific gene (*Hart and Cooper, 1999*). CapZβ and CapZα structurally resemble each other, although their primary sequences are quite distinct (*Cooper and Sept, 2008*; *Yamashita et al., 2003*). Some proteins (e.g., formins and Ena/VASP) compete with CapZ to interact with the barbed ends of actin. Other proteins interact with CapZ directly, and they inhibit the capping activity by either steric blocking (e.g., V-1/myotrophin), or allosteric inhibition (e.g., CARMIL proteins) (*Edwards et al., 2014*; *Shekhar et al., 2016*). CapZ has been shown to participate in various

cellular processes, such as forming filopodia and lamellipodia, and regulating the mechanical properties of cells and tissues, via fine-tuning dynamics of actin assembly (*Edwards et al., 2014*; *Sinnar et al., 2014*; *Pocaterra et al., 2019*). Interestingly, CapZ has also been reported to regulate spindle assembly during mitosis independent of actin assembly (*di Pietro et al., 2017*). Although extensive works have documented the contribution of the actin filaments in the endocytosis pathway, the precise role of CapZ in endosomal trafficking, especially early endosome maturation, remains elusive. Here, we found that CapZ is a dual-functional factor during endosomal trafficking: it controls the actin density around premature early endosomes, and recruits RAB5 effectors to early endosomes.

## Results

### CapZ is required for early endosome maturation

To investigate the role of CapZ in endocytosis, we examined the subcellular localization of CapZ by confocal imaging. We showed that in CapZβ-GFP-expressing cells, CapZ was localized both diffusely and in puncta, and the majority of the CapZ puncta was colocalized with F-actin puncta, which were labeled with phalloidin (top panel in *Figure 1—figure supplement 1A*). Interestingly, in cells treated with vacuolin-1, which can induce homotypic fusion of endosomes to form large vacuoles (*Cerny et al., 2004*; *Ye et al., 2021*), some of the CapZ puncta appeared to be organized into large vesicular structures (bottom panel in *Figure 1—figure supplement 1A*). To determine the identity of these CapZ-positive vesicular structures, we examined if the CapZ puncta might be localized at any known cellular organelle. Our data showed that the CapZ puncta exhibited weak colocalization with Calnexin (an ER protein), GM130 (a cis-Golgi protein), and TGN46 (a trans-Golgi protein) (*Figure 1—figure supplement 1B*). However, these CapZ puncta were strongly associated with the early endosome markers, EEA1 or RAB5 (*Figure 1A*). Interestingly, treatment of cells with YM-201636 (a PIKfyve inhibitor which blocks the conversion of PI(3)P to PI(3,5)P2) (*Osborne et al., 2008*), not only enlarged early endosome but also strikingly induced the accumulation of CapZ at the surface of the enlarged endosomes (*Figure 1B*). Of note, CapZ has previously been shown to preferably bind with PI(3)P rather than with other phosphoinositides in vitro (*Schafer et al., 1996*; *Mi et al., 2015*). We also performed N-SIM S super-resolution imaging of the HeLa cells co-transfected with CapZβ-mCherry and RAB5A-GFP, and confirmed the colocalization of CapZ with RAB5A at the endosome (*Figure 1C*). To further validate the possible association of CapZ with early endosomes, we performed coimmunoprecipitation (co-IP) experiments to assess whether CapZ interacts with RAB5. We found that in twinstrep-CapZβ-expressing cells, streptavidin beads not only pulled down CapZ, but also brought down RAB5A (*Figure 1D*). Likewise, CapZ was found in the RAB5 immunocomplex (*Figure 1E*). In addition, recombinant GST-RAB5 protein was able to pull down CapZ from HEK293 cell extracts (*Figure 1F*). Interestingly, CapZ exhibited no direct interaction with RAB5A as shown by an isothermal titration calorimetry (ITC) assay (*Figure 1—figure supplement 1C*). We also assessed the colocalization of CapZ with other endosome markers, for example, RAB7 (a late endosome protein), RAB11/RAB4 (two recycling endosome proteins), and LAMP1 (a late endosome/lysosome protein). We showed that CapZ showed strong colocalization with RAB5, and weak colocalization with RAB7, RAB11, RAB4, and LAMP1 (*Figure 1—figure supplement 1D and E*). Similarly, by co-IP assay, we detected the association of CapZ with RAB5, RAB7, and RAB11, but not LAMP1 and RAB4 (*Figure 1—figure supplement 1F–1I*). Taken together, these results suggest that CapZ might preferentially associate with early endosomes.

We knocked out the expression of CapZβ in HeLa cells by CRISPR/Cas9, and also reconstituted the expression of CapZβ by infecting these cells with a lentivirus encoding CapZβ (*Figure 2A*). Similar to the previous study by *Sinnar et al., 2014*, CapZβ knockout in HeLa cells changed cell morphology and inhibited cell migration (*Figure 2—figure supplement 1A and B*). CapZβ knockout also increased cell size (*Figure 2—figure supplement 1A*). We then performed an epidermal growth factor receptor (EGFR) degradation assay to examine CapZ's role in the endosomal trafficking pathway. As shown in *Figure 2B*, *Figure 2—figure supplement 1C and D*, after EGF binds to its receptors (EGFR), the receptor complex was internalized normally in control, CapZβ-knockout, and CapZβ-reconstituted cells, but its degradation was significantly inhibited in CapZβ-knockout cells when compared to the control cells and CapZβ-reconstituted cells. Similarly, we examined the EGFR levels after EGF pulse in control or CapZβ-knockout HeLa cells by EGFR immunoblotting. We showed that EGF treatment

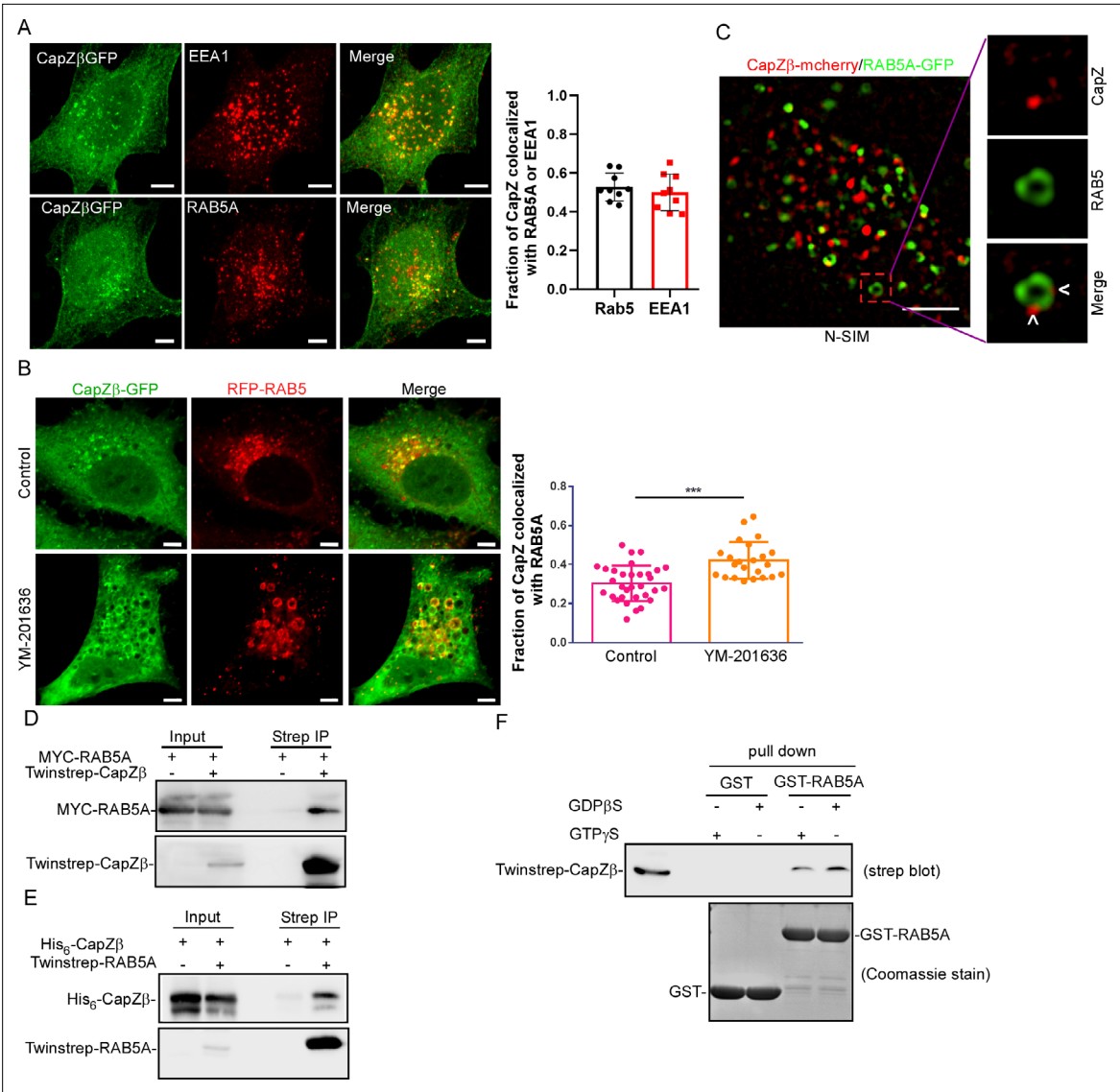

**Figure 1.** CapZ is associated with early endosomes. (**A**) CapZβ-GFP-expressing HeLa cells were immunostained with antibodies against EEA1 (top panel) or RAB5A (lower panel). Confocal images taken along the z-axis were projected (Maximum Intensity Projection). The colocalization coefficients (MCC) of CapZβ-GFP and EEA1 or CapZβ-GFP and RAB5A were quantified. (**B**) CapZβ-GFP/RFP-RAB5A-expressing HeLa cells were treated with or without YM-201636 (5 µM) for 8 hr, followed by confocal imaging. The colocalization coefficients (MCC) of mRFP-RAB5A and CapZβ-GFP were quantified. (**C**) The N-SIM S Super-resolution imaging of CapZβ-mCherry/RAB5A-GFP-expressing HeLa cells. (**D**) HEK293 cells were transiently transfected with MYC-RAB5A, and/or Twinstrep-CapZβ, and the cell lysates were incubated with Strep-Tactin Sepharose. The pulldowns were subjected to immunoblotting analysis against CapZβ and RAB5A. (**E**) HEK293 cells were transiently transfected with $His_6$-CapZβ and/or Twinstrep-RAB5A, and the cell lysates were then incubated with Strep-Tactin Sepharose. The pulldowns were subjected to immunoblot analysis against CapZβ and RAB5A. (**F**) The lysates of Twinstrep-CapZβ-expressing HEK293 cells were incubated with GST or GST-RAB5A recombinant proteins in the presence of GDPβS or GTP$\gamma$S, followed by GST pulldown and strep immunoblotting. The blots, images, and graphs represent data from at least three independent experiments. The data are expressed as mean ± SD. The difference between the two groups was analyzed using two-tailed Student's t-test, p<0.05 was considered statistically significant. *** p<0.001. Scale bar, 5 µm. MCC, Manders colocalization coefficient.

The online version of this article includes the following figure supplement(s) for figure 1:

**Source data 1.** Original immunoblots.

**Source data 2.** Data for *Figure 1A, B*.

**Figure supplement 1.** CapZ is associated with early endosomes.

**Figure supplement 1—source data 1.** Original immunoblots.

**Figure supplement 1—source data 2.** Data for 1C, 1E.

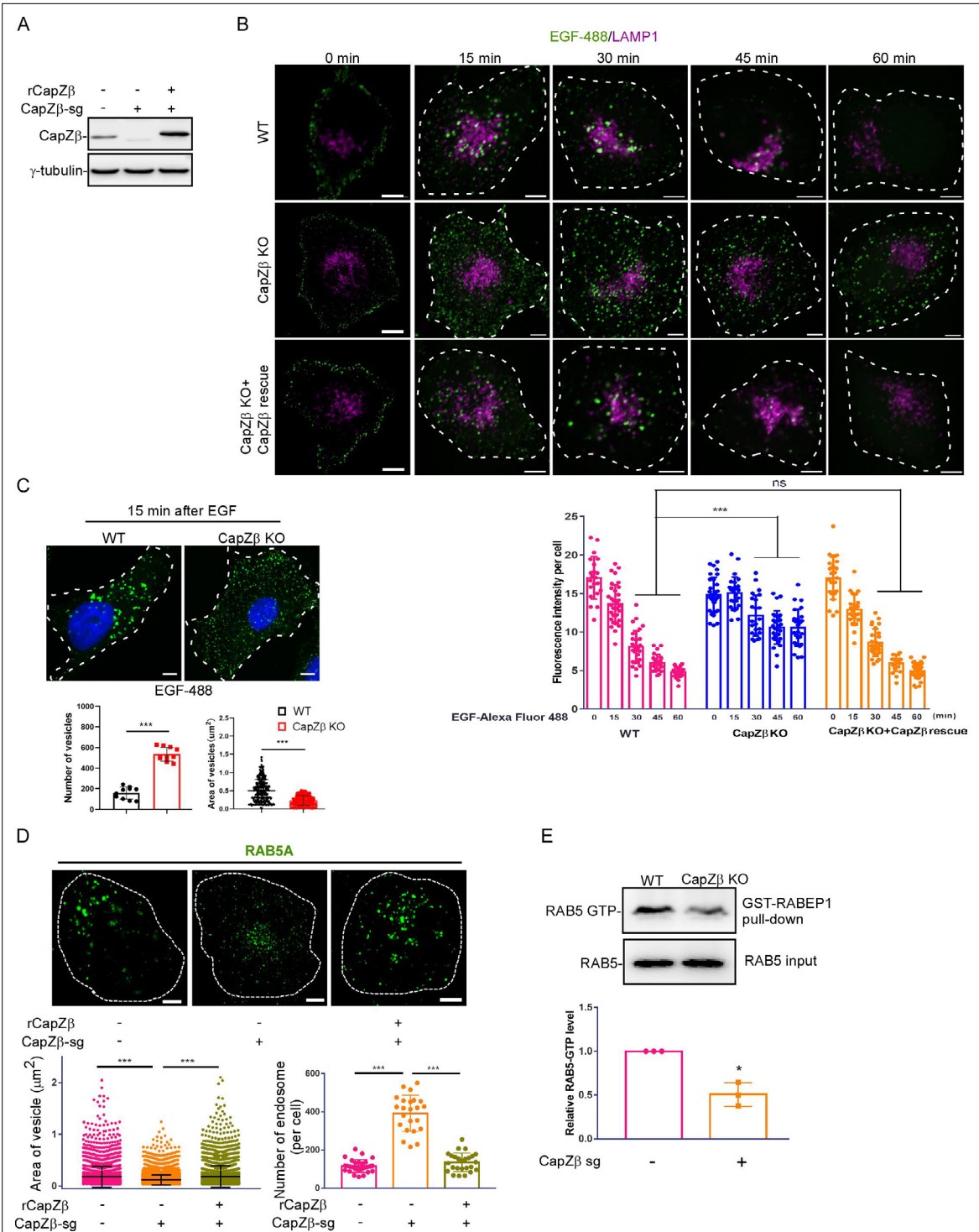

**Figure 2.** CapZ is required for early endosome maturation. (**A**) The expression of CapZβ in control, CapZβ-knockout, or CapZβ-reconstituted HeLa cells. (**B**) Control, CapZβ-knockout, or CapZβ-reconstituted HeLa cells were treated with EGF-488 for the indicated times, and then processed for immunofluorescence analysis. (**C**) Control or CapZβ-knockout HeLa cells were treated with EGF-488 for 15 min, and the number and size of EGF-positive vesicles in these cells were quantified. (**D**) Control, CapZβ-knockout, or CapZβ-reconstituted HeLa cells were immunostained with an anti-RAB5A antibody. The number and size of the early endosomes in these cells were quantified. (**E**) Active RAB5 in control or CapZβ knockout cells was examined with a GST-R5BD pulldown assay. The images and graphs represent data from three independent experiments. The data are expressed as mean ± SD. The difference between the two groups was analyzed using two-tailed Student's t-test, p<0.05 was considered statistically significant. *** p<0.001. Scale bar, 5 μm.

*Figure 2 continued on next page*

*Figure 2 continued*

The online version of this article includes the following figure supplement(s) for figure 2:

**Source data 1.** Original immunoblots.

**Source data 2.** Data for 2B–2E.

**Figure supplement 1.** CapZ is required for early endosome maturation.

**Figure supplement 1—source data 1.** Original immunoblots.

**Figure supplement 1—source data 2.** Data for 1B, 1D-1F, 1 H, 1I.

triggered EGFR degradation in control cells, yet its degradation in CapZβ-knockout cells was inhibited (*Figure 2—figure supplement 1E*). Of note, the basal level of EGFR protein or mRNA in CapZβ-knockout cells was lower than in control cells, and reconstituting CapZβ restored the expression of EGFR (*Figure 2—figure supplement 1F–1G*). This result suggests that CapZ or F-actin dynamics positively regulate the transcription of EGFR. Along this line, actin dynamics have been shown to modulate gene transcription (*Olson and Nordheim, 2010*; *Knöll, 2010*). In addition, we measured transferrin recycling in control or CapZβ-knockout HeLa cells. In transferrin recycling assay, iron-bound transferrin interacts with its receptor (TfR) to trigger the internalization of the iron-transferrin-TfR complex via clathrin-mediated endocytosis. Irons are subsequently dissociated from their ligands in the acidic endosomes and released into the cytoplasm. Thereafter, the transferrin-TfR complex is recycled back to the plasma membrane, and transferrin is finally released from its receptor into the extracellular space. Thus, the fluorescence intensity of fluorescent transferrin conjugates-labeled cells could reflect the recycling endocytic trafficking process. We showed that CapZβ knockout significantly inhibited transferrin recycling (*Figure 2—figure supplement 1H*), suggesting that CapZ might be involved in the fast recycling from the early endosome or RAB11-positive recycling endosomes. Of note, fluorescent transferrin labeling was reduced in CapZβ-knockout cells, which might be due to the decreased expression levels of TfR in CapZβ-knockout cells. In summary, these results indicate that CapZ participates in endocytosis after the internalization of endocytic vesicles.

Since EGFR-containing endocytic vesicles in CapZβ-knockout cells appear to be smaller and more abundant when compared to the control cells (*Figure 2B and C*), we next examined whether the association of CapZ with endosomes is required for the maturation of early endosomes by comparing the size and the number of endosomes in CapZβ-knockout cells with the same parameters in control cells. We showed that the size of the RAB5-positive early endosomes was much smaller, whereas the number of endosomes was significantly higher in the CapZβ-knockout cells than in the control cells (*Figure 2D* and *Figure 2—figure supplement 1I*). Addback of CapZβ to CapZβ-knockout HeLa cells restored the size and number of early endosomes to those in control cells (*Figure 2D*). This suggests that the fusion of small endocytic vesicles in the CapZβ-knockout cells is blocked or inhibited. Since CapZβ knockout exhibited a similar endosomal phenotype to that of RAB5-S34N overexpression (a GDP-bound dominant-negative mutant, RAB5 inactive form) (*Galperin and Sorkin, 2003*), we examined whether CapZβ knockout might impair the RAB5 activity by a GST-R5BD pulldown assay, in which the RAB5-binding domain (R5BD) of the RAB effector Rabaptin-5 specifically binds to GTP-bound RAB5. We showed that the level of RAB5-GTP was significantly lower in the CapZβ-knockout cells than that in the control cells (*Figure 2E*), indicating that RAB5 is less active in the knockout cells. In summary, these results indicate that CapZ participates in early endosome maturation.

## CapZ controls the F-actin density around endocytic vesicles to regulate early endosome maturation

CapZ caps the barbed end of actin filaments and thereby stops further actin polymerization (*Wear and Cooper, 2004*). As expected, in CapZβ-knockout cells, the amount of F-actin was significantly increased, and CapZβ-addack brought it down to the basal levels (*Figure 3—figure supplement 1A*). We also knocked out the expression of CapZα1 and CapZα2 by CRISPR/Cas9 in HeLa cells (*Figure 3—figure supplement 1B*). Similar to CapZβ knockout, CapZα1/2 knockout significantly increased the amount of F-actin (*Figure 3—figure supplement 1C*), and induced the accumulation of small early endosomes (*Figure 3—figure supplement 1D*). On the other hand, overexpression of both CapZα1-EGFP and CapZβ-mCherry in HeLa cells significantly decreased F-actin levels when compared to EGFP-mCherry-expressing cells (*Figure 3A*). Consistently, the size of RAB5-positive early endosomes

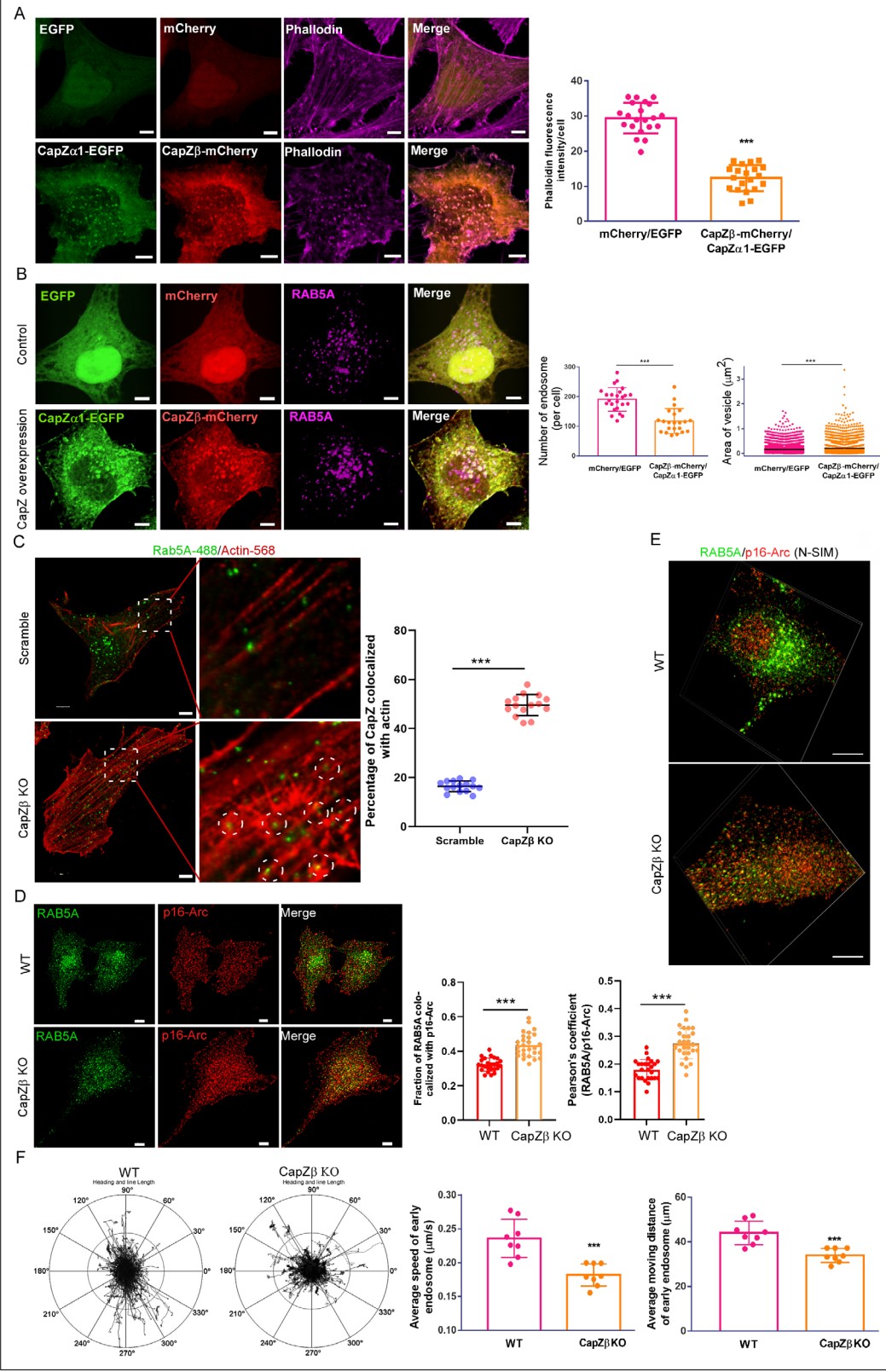

**Figure 3.** CapZ controls the actin density around early endosomes. (**A, B**) HeLa cells were stably transfected with EGFP/mCherry or CapZα1-EGFP/CapZβ-mCherry constructs. These cells were either immunostained with phalloidin (**A**), or with an anti-RAB5A antibody (**B**). The number and size of the early endosomes in these cells were quantified (**B**). (**C**) Control or CapZβ-knockout HeLa cells were immunostained with antibodies against actin and

*Figure 3 continued on next page*

*Figure 3 continued*

RAB5A. The single plane whole-cell confocal images were then captured with a resolution of 2048×2048 pixels (left panels). The selected area in the cells was further scanned with a resolution of 4096×4096 pixels (right panels). The percentage of RAB5 puncta that are actin positive against total RAB5 puncta (MCC) in control or CapZβ-knockout cells was quantified. (**D, E**) The control or CapZβ-knockout HeLa cells were stained with antibodies against RAB5 and p16-Arc, followed by confocal imaging and colocalization coefficients (MCC or PCC) analysis of RAB5/p16-Arc (**D**) or N-SIM S Super-resolution imaging (**E**). (**F**) Control or CapZβ-knockout HeLa cells were transiently transfected with RFP-RAB5A, and cells were then imaged without delay for 3 min by time-lapse Nikon high-speed confocal microscopy. 2D track analysis was performed using NIS-Elements AR software. The images and graphs represent data from at least three independent experiments. The data are presented as mean ± SD. The difference between the two groups was analyzed using two-tailed Student's t-test. Differences were considered statistically significant when $p<0.05$. * $p<0.05$, ** $p<0.01$, and *** $p<0.001$. Scale bar, 5 µm. MCC, Manders colocalization coefficient; PCC, Pearson's correlation coefficient.

The online version of this article includes the following video and figure supplement(s) for figure 3:

**Source data 1.** Data for 3A–3D, 3F.

**Figure supplement 1.** The role of actin density around endocytic vesicles in the maturation of early endosomes.

**Figure supplement 1—source data 1.** Original immunoblots.

**Figure supplement 1—source data 2.** Data for 1A, 1C–1F, 1I.

**Figure 3—video 1.** The reconstituted 3-D N-Sim images of endogenous Arp2/3 complex and endogenous RAB5 in wild-type HeLa cells.

https://elifesciences.org/articles/65910/figures#fig3video1

**Figure 3—video 2.** The reconstituted 3-D N-Sim images of endogenous Arp2/3 complex and endogenous RAB5 in CapZβ-knockout cells.

https://elifesciences.org/articles/65910/figures#fig3video2

**Figure 3—video 3.** RAB5-positive endosomes movements in wild-type HeLa cells.

https://elifesciences.org/articles/65910/figures#fig3video3

**Figure 3—video 4.** RAB5A-positive endosomes movements in CapZβ-knockout HeLa cells.

https://elifesciences.org/articles/65910/figures#fig3video4

was larger, whereas the number of endosomes was lower in the CapZ-overexpressing cells than in the EGFP-mCherry-expressing cells (*Figure 3B*). In addition, we investigated whether reducing the F-actin levels in CapZβ-knockout cells by latrunculin A (*Coué et al., 1987*) (an actin polymerization inhibitor) rescued the defects in early endosome maturation. Latrunculin A markedly decreased the F-actin levels in control HeLa cells and was also able to revert the F-actin to basal levels in CapZβ-knockout HeLa cells (*Figure 3—figure supplement 1E*). Likewise, latrunculin A not only increased the size and decreased the number of early endosomes in control cells, but also partially rescued the defects of early endosomes in CapZβ-knockout cells (*Figure 3—figure supplement 1F*). Therefore, these results suggest that CapZ knockout might increase F-actin density by promoting elongation of pre-existing actin branches to entangle around these immature RAB5-positive endocytic vesicles, and this might impede their fusion and result in the accumulation of small endosomes.

To test this possibility, we performed double immunostaining with antibodies against RAB5 and actin in control or CapZβ-knockout cells. The single plane whole-cell confocal images were captured with a resolution of 2048×2048 pixels (left panels of *Figure 3C*). To demonstrate the detail of the F-actin, we further scanned the selected area in the cells with a resolution of 4096 × 4096 pixels (right panels of *Figure 3C*). We showed that there were more RAB5-positive endosomes in actin-rich regions of the CapZβ-knockout cells than the control cells. We quantified the percentage of RAB5 puncta that are actin positive against total RAB5 puncta in control or CapZβ-knockout cells. We showed that more endosomes were actin-positive in CapZβ-knockout cells when compared to the control cells (*Figure 3C*). Similarly, we performed RAB5 and phalloidin immunostaining in control or CapZβ-knockout cells, followed by confocal images (*Figure 3—figure supplement 1G*) or N-SIM S Super-resolution imaging (*Figure 3—figure supplement 1H*). It appeared that more endosomes sit along the long F-actin in CapZβ-knockout cells than in the control cells.

We then performed RAB5A and p16-Arc (a subunit of the Arp2/3 complex, to label actin branches) immunostaining in control or CapZβ-knockout cells, followed by confocal images. We showed that

CapZβ knockout significantly increased the colocalization of RAB5 with Arp2/3 (*Figure 3D*). We also performed N-SIM S Super-resolution imaging to assess the colocalization between RAB5 and Arp2/3, and showed that more RAB5 puncta were associated with Arp2/3 puncta in CapZβ-knockout cells than in the control cells (*Figure 3E*, and *Figure 3—video 1* and *Figure 3—video 2*). Likewise, CapZβ knockout markedly increased the colocalization of N-WASP with RAB5 (*Figure 3—figure supplement 1I*). In summary, these results suggest that more F-actins are entangled with early endosomes in CapZβ-knockout cells when compared to the control cells.

We also transfected control or CapZβ-knockout HeLa cells with RAB5A-GFP, and analyzed the trafficking of RAB5-positive endosomes in these cells by high-speed confocal microscopy. In control cells, RAB5-positive vesicles rapidly moved away from the plasma membrane toward the perinuclear area and fused with each other (left panel in *Figure 3F* and *Figure 3—video 3*). Whereas, in CapZβ-knockout cells, RAB5-positive vesicles exhibited relative random and slow movements underneath the plasma membrane (right panel in *Figure 3F* and *Figure 3—video 4*). Moreover, the moving distance of RAB5-positive vesicles was shorter in CapZβ-knockout cells than in the control cells (*Figure 3F*). Therefore, these results suggest that the increased F-actin density around the immature early endosome in CapZβ-knockout cells might act as the physical barrier to impede endosomal trafficking.

To further investigate whether the increased F-actin density in CapZβ-knockout cells is responsible for the defects in early endosome maturation, we treated control and CapZβ-knockout cells with CK-636 (an inhibitor of the Arp2/3 complex) (*Nolen et al., 2009*). The Arp2/3 complex has been shown to play an important role during endosomal trafficking, from cargo sorting, fission to fusion (*Mooren et al., 2012*). We showed that CK-636 significantly decreased the colocalization between RFP-RAB5 and GFP-ARPC1B (a subunit of the Arp2/3 complex, to label actin branches) in both control and CapZβ-knockout cells (*Figure 4—figure supplement 1A*), suggesting that CK-636 treatment could lower the F-actins entangled with early endosomes. We also measured the size and numbers of RAB5-positive vesicles in control and CapZβ-knockout cells treated with or without CK-636. CK-636 not only significantly increased the size and decreased the number of early endosomes in control cells but also partially rescued the defects of early endosomes in CapZβ-knockout cells (*Figure 4A*).

In addition to CK-636 treatment, we knocked down the expression of ARP2 in control and CapZβ-knockout HeLa cells to inhibit actin branching. We first performed RAB5 and actin immunostaining in control or ARP2-knockout cells, and showed that colocalization between RAB5 and actin in ARP2-knockout cells was markedly weaker than in control cells (*Figure 4—figure supplement 1B*). We also performed RAB5 and ARPC1B immunostaining in control or ARP2-knockout cells, and found that colocalization between endogenous RAB5 and endogenous ARPC1B in ARP2-knockout cells was markedly weaker than in control cells (*Figure 4B*). These results confirmed that ARP2 knockout decreases the F-actin density around early endosomes. We then measured the size and numbers of early endosomes in control, CapZβ-knockout, ARP2-knockout, or CapZβ/ARP2-double knockout cells. As expected, ARP2 knockout significantly increased the size and decreased the number of RAB5-positive endosomes in control cells and partially rescued the defects of endosomes in CapZβ-knockout cells (*Figure 4C*). To confirm that the accumulation of large endocytic vesicles in ARP2-knockout cells is due to the increased fusion between them, we knocked out the expression of VPS8 (*Chen and Stevens, 1996*; *Horazdovsky et al., 1996*), (a subunit of CORVET complex which controls the homotypic fusion of early endosomes), in control or ARP2-knockout HeLa cells. As expected, VPS8 knockout significantly decreased the size but increased the number of RAB5-positive endosomes in ARP2-knockout cells (*Figure 4D*). Finally, we examined the colocalization of RAB5 with RAB7 in control, ARP2-knockout, or CapZβ-knockout cells. We showed that the colocalization of RAB5 with RAB7 was significantly decreased in CapZβ-knockout cells compared to the control cells, suggesting that most of the accumulated RAB5-positive vesicles are immature early endosomes. Whereas most of the enlarged endosomes in ARP2-knockout cells were RAB7 positive. Intriguingly, some of these RAB7-positive enlarged vehicles were also RAB5 positive, manifested by the increased colocalization between RAB5 and RAB7 in ARP2-knockout cells when compared to the control cells (*Figure 4—figure supplement 1C*). The enlarged RAB5-RAB7 double-positive vesicles might be the transition vesicles between early endosomes and late endosomes (*Poteryaev et al., 2010*), and the RAB7-only positive enlarged vesicles might be late endosomes. Taken together, these results indicate that CapZ knockout increases F-actin density by promoting elongation of pre-existing branches to entangle

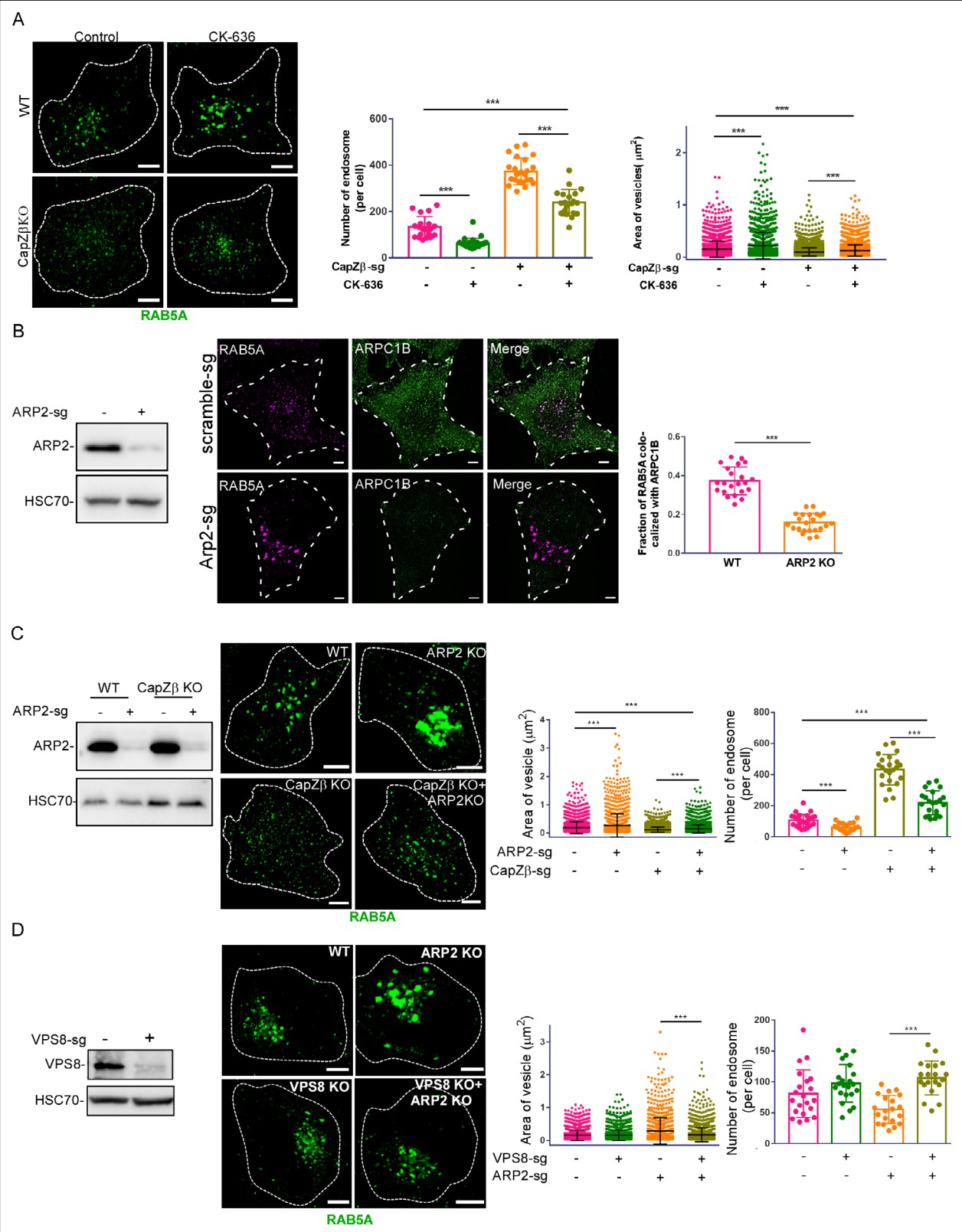

**Figure 4.** The actin density around early endosomes regulates its maturation. (**A**) Control or CapZβ-knockout HeLa cells were treated with or without CK-636 (100 µM) overnight and were then immunostained with an anti-RAB5A antibody. The number and size of the early endosomes in these cells were quantified. (**B**) Control or ARP2-knockout HeLa cells were immunostained with antibodies against RAB5A and ARPC1B, followed by confocal imaging. The colocalization coefficients (MCC) of RAB5A and ARPC1B were quantified. The expression of ARP2 in these cells was analyzed by immunoblot

*Figure 4 continued on next page*

*Figure 4 continued*

analysis. (**C**) Control, CapZβ-knockout, ARP2-knockout, or CapZβ/ARP2-double knockout HeLa cells were immunostained with an anti-RAB5A antibody, and the number and size of the early endosomes in these cells were then quantified. The expression of ARP2 in these cells was analyzed by immunoblot analysis. (**D**) Control, VPS8-knockout, ARP2-knockout, or VPS8/ARP2-double knockout HeLa cells were immunostained with an anti-RAB5A antibody, and the number and size of the early endosomes in these cells were then quantified. The expression of VPS8 in control or VPS8-knockout cells was analyzed by immunoblot analysis. The images and graphs represent data from at least three independent experiments. The data are presented as mean ± SD. The difference between the two groups was analyzed using two-tailed Student's t-test. Differences were considered statistically significant when p<0.05. * p<0.05, ** <0.01, and *** <0. 001. Scale bar, 5 µm. MCC, Manders colocalization coefficient.

The online version of this article includes the following figure supplement(s) for figure 4:

**Source data 1.** Original immunoblots.

**Source data 2.** Data for 4A–4D.

**Figure supplement 1.** The role of actin density around endocytic vesicles in the maturation of early endosomes.

**Figure supplement 1—source data 1.** Data for 1A–1C.

endocytic vesicles, thereby impeding their trafficking and fusion and resulting in the accumulation of small early endosomes.

## CapZ recruits RAB5 effectors to early endosomes

CK-636, ARP2 knockout, or latrunculin A only partially rescued the defects of early endosomes in CapZβ-knockout HeLa cells when compared to control cells (*Figure 4A and C*, and *Figure 3—figure supplement 1F*), although latrunculin A appeared to completely reverse the F-actin filaments to basal levels (*Figure 3—figure supplement 1E*) and CK-636 even decreased the actin branches around early endosomes below the basal levels (*Figure 4—figure supplement 1A*). These results suggest that CapZ might contribute to the maturation of early endosomes by other means in addition to regulating F-actin density. Interestingly, we found that in twinstrep-CapZβ-expressing cells, streptavidin beads not only pulled down CapZβ, but also brought down RAB5 effectors, for example, Rabex-5, Rabaptin-5, and VPS34 (*Figure 5A–C*). Likewise, CapZ was found in the Rabaptin-5 or VPS34 immunocomplex (*Figure 5—figure supplement 1A and B*). We then purified recombinant CapZ (CapZα1-CapZβ heterodimer) and Rabaptin-5 protein (*Figure 5—figure supplement 1C*), and performed biolayer interferometry (BLI) assay to assess the interaction between Rabaptin-5 and CapZ. We showed that Rabaptin-5 bound to immobilized CapZ in a dose-dependent manner (*Figure 5D*). ITC further confirmed the direct interaction between recombinant CapZ and Rabaptin-5 proteins (*Figure 5—figure supplement 1D*). Notably, RAB5A also directly interacted with Rabaptin-5 (*Figure 5—figure supplement 1E*).

It has been reported that the activation of RAB5 on early endosomes recruits and activates more effectors to endosomes, establishing a positive feedback loop to induce early endosome maturation (*Langemeyer et al., 2018*). We speculated that CapZ might facilitate the recruitment of these RAB5 effectors to early endosomes, thus activate RAB5 and induce early endosome maturation. We, therefore, examined the effects of CapZβ knockout on the association of RAB5 with its effectors during endocytosis by immunofluorescence staining. As expected, CapZβ knockout significantly decreased the colocalization of RAB5 with EEA1 (*Figure 5E*), Rabaptin-5 (*Figure 5F*), or Rabex-5 (*Figure 5G*). Co-IP experiment confirmed that the interaction between Rabaptin-5 and RAB5 was markedly decreased in CapZβ-knockout cells when compared to the control cells (*Figure 5H*). CapZβ knockout also markedly reduced the interaction between Rabex-5 and Rabaptin-5 (*Figure 5H and I*). Whereas knockdown of either Rabex-5 or Rabaptin-5 did not affect the ability of CapZ to interact with Rabex-5 or Rabaptin-5, respectively (*Figure 5—figure supplement 1F and G*). Since the recruitment of the Rabaptin-5 and Rabex-5 (a RAB5 GEF) complex to early endosomes is required to maintain the RAB5-GTP level for early endosome maturation, we assessed Rabaptin-5 knockout on the size and number of endosomes in control or ARP2-knockout cells. As expected, Rabaptin-5 knockout increased the number, and decreased the size of early endosomes in both control and ARP2-knockout cells (*Figure 5J*). In summary, these results suggest that CapZ facilitates the recruitment of Rabex-5 and Rabaptin-5 to endosomes, to activate RAB5 during early endosome maturation.

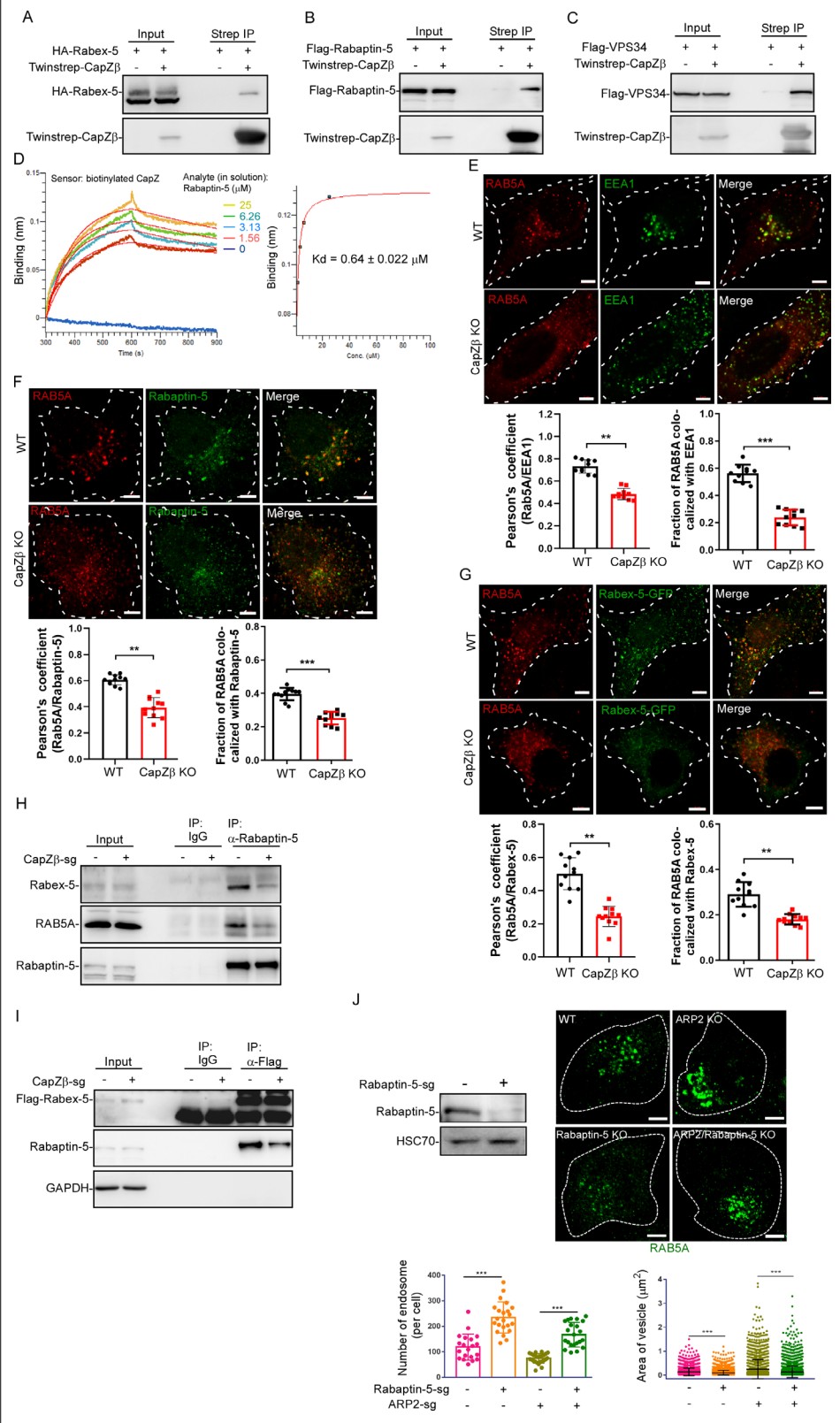

**Figure 5.** CapZ facilitates the recruitment of RAB5 effectors to early endosomes. (**A**) HEK293 cells were transiently transfected with HA-Rabex-5, and/or Twinstrep-CapZβ, and the cell lysates were then incubated with Strep-Tactin Sepharose. The pulldowns were subjected to immunoblotting analysis against CapZβ and Rabex-5. (**B, C**) HEK293 cells were transiently transfected with Flag-Rabaptin-5 (**B**), Flag-VPS34 (**C**), and/or Twinstrep-

*Figure 5 continued on next page*

*Figure 5 continued*

CapZβ, and the cell lysates were then incubated with Strep-Tactin Sepharose. The pulldowns were subjected to immunoblot analysis against CapZβ, Rabaptin-5, and/or VPS34. (**D**) Wavelength shift (nm) generated by the addition of Rapaptin-5 at indicated concentrations to biotinylated CapZ (200 µg/ml) immobilized on a streptavidin biosensor. Association was monitored for 5 min, followed by a 5 min dissociation step in assay buffer. Binding affinity constant was generated by fitting both the association and dissociation curves to a 1:1 binding model. (**E**) Control or CapZβ-knockout HeLa cells were immunostained with an anti-RAB5A or anti-EEA1 antibody, after which the colocalization coefficients (PCC or MCC) of EEA1 and RAB5A were quantified. (**F**) Control or CapZβ-knockout HeLa cells were immunostained with an anti-RAB5A or anti-Rabaptin-5 antibody. The colocalization coefficients (PCC or MCC) of Rabaptin-5 and RAB5A were quantified. (**G**) Control or CapZβ-knockout HeLa cells were transiently transfected with Rabex-5-GFP, and subjected to immunostaining with RAB5A antibody. The colocalization coefficients (PCC or MCC) of Rabex-5 and RAB5A were quantified. (**H**) The lysates of control or CapZβ-knockout cells were immunoprecipitated with an anti-Rabaptin-5 antibody, and the immunoprecipitates were then subjected to immunoblot analysis against Rabex-5, Rabaptin-5, and RAB5A. (**I**) Control or CapZβ-knockout HeLa cells were stably transfected with Flag-Rabex-5, and the cell lysates were then immunoprecipitated with an anti-Flag antibody. The immunoprecipitates were subjected to immunoblot analysis against Rabex-5 and Rabaptin-5. (**J**) Control, ARP2-knockout, Rabaptin-5-knockout, or ARP2/Rabaptin-5-double knockout HeLa cells were immunostained with an anti-RAB5A antibody, and the number and size of the early endosomes in these cells were then quantified. The expression of Rapaptin-5 in control or Rapaptin-5-knockout HeLa cells was analyzed by immunoblot analysis. The BLI data (D) represent two independent experiments, and the rest of the blots, images, and graphs represent data from at least three independent experiments. The data are presented as mean ± SD. The difference between the two groups was assessed using two-tailed Student's t-test. Differences were considered statistically significant when p<0.05. ** p<0.01, and *** p<0.001. Scale bar, 5 µm. BLI, biolayer interferometry; MCC, Manders colocalization coefficient; PCC, Pearson's correlation coefficient.

The online version of this article includes the following figure supplement(s) for figure 5:

**Source data 1.** Original immunoblots.

**Source data 2.** Data for 5D–5G, 5J.

**Figure supplement 1.** The role of CapZ in the recruitment of RAB5 effectors to early endosomes.

**Figure supplement 1—source data 1.** Original immunoblots.

**Figure supplement 1—source data 2.** Data for 1D, 1E.

## Recruitment of RAB5 effectors to the early endosome by CapZ depends on its actin capping activity

Although RAB5 did not directly interact with CapZ in vitro (*Figure 1—figure supplement 1C*), in Twinstrep-actin-expressing cells, streptavidin beads pulled down both actin and RAB5A (*Figure 6—figure supplement 1A*). Early endosomes are known to be associated with F-actin (*Taunton et al., 2000*; *Kaksonen et al., 2005*), and several proteins are required for their association, such as Annexin A2 and RN-tre (a GAP for Rab5) (*Morel et al., 2009*; *Lanzetti et al., 2004*). We, thus, speculated that CapZ might carry Rabaptin-5 and/or Rabex-5 to RAB5-positive early endosomes by capping the barbed end of F-actin. It has been shown that the C-terminal 29–34 amino acids of CapZβ or CapZα bind to the barbed ends of actin filaments with high affinity (*Figure 6A*; *Yamashita et al., 2003*; *Narita et al., 2006*), and that the capping activity of a CapZβ mutant where the C-terminal tail is deleted, is at least 300-fold lower than the wild-type protein (*Wear et al., 2003*). To investigate the capping activity of CapZ in the maturation of early endosomes, we constructed a CapZβ deletion mutant by removing the last 34 amino acids at its C-terminal (CapZβ-ΔC34). We subsequently transfected it back to CapZβ-knockout HeLa cells and selected the clones that mimic the endogenous CapZβ expression levels (*Figure 6B*). We showed that CapZβ-ΔC34 addback not only failed to revert the high level of F-actin induced by CapZβ knockout to that in control cells, but also further increased the F-actin levels, suggesting that this mutant might act as a dominant-negative mutant to prevent other actin capping proteins from binding to the barbed ends of actin (*Figure 6—figure supplement 1B*). As expected, CapZβ-ΔC34 addback did not restore the size and number of early endosomes in CapZβ-knockout cells to those in control cells, and it further induced the accumulation of small endosomes when compared to the CapZβ-knockout cells (*Figure 6C*). Likewise, the depletion of EGF-488 signals was significantly inhibited in CapZβ-ΔC34-reconstituted cells (*Figure 6D* and *Figure 6—figure supplement 1C*). Moreover, CapZβ-ΔC34-GFP exhibited diffused expression pattern whereas

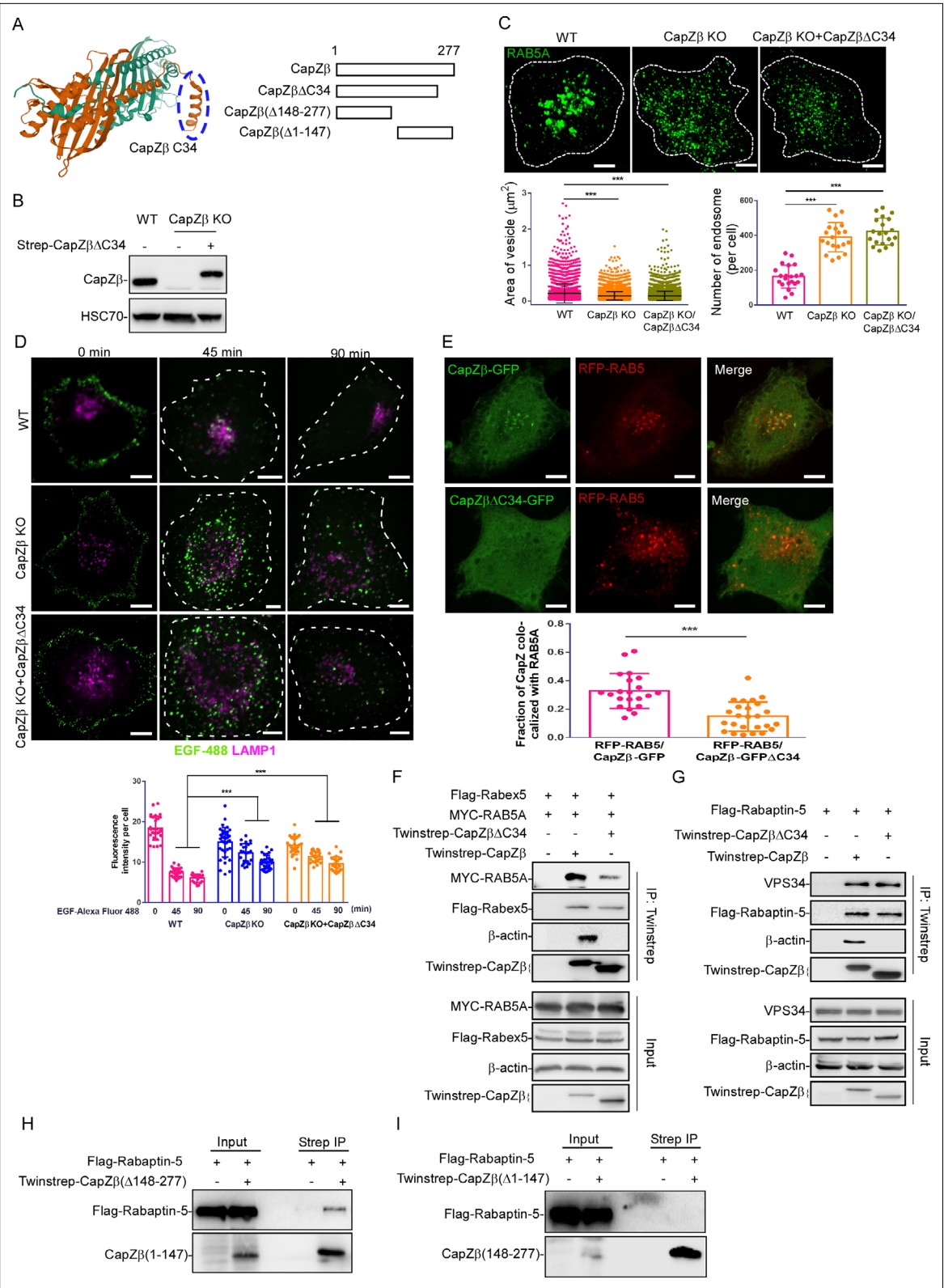

**Figure 6.** Recruitment of RAB5 effectors to the early endosomes by CapZ depends on its actin capping activity. (**A**) The left panel shows the actin-binding motif in the CapZ structure generated from the protein data bank (PDB ID: 1IZN). Cyan color indicated CapZα structure, and brown color indicated CapZβ structure. The right panel shows various CapZβ deletion constructs. (**B**) The expression of CapZβ in control, CapZβ-knockout, or Twinstrep-CapZβ-ΔC34-reconstituted HeLa cells was determined by CapZβ immunoblotting. (**C**) Control, CapZβ-knockout, or Twinstrep-CapZβ-ΔC34-

*Figure 6 continued on next page*

*Figure 6 continued*

reconstituted HeLa cells were stained with an anti-RAB5A antibody. The number and size of the early endosomes in these cells were quantified. (**D**) Control, CapZβ-knockout, or Twinstrep-CapZβ-ΔC34-reconstituted HeLa cells were treated with EGF-488 for the indicated times, and processed for immunofluorescence analysis. (**E**) HeLa cells were transfected with RFP-RAB5A, and CapZβ-GFP or CapZβ-ΔC34-GFP, and subjected to confocal imaging 24 hr later. The colocalization coefficients (MCC) of RFP-RAB5A and CapZβ-GFP or RFP-RAB5A and CapZβΔC34-GFP were quantified. (**F**) HEK293 cells were transfected with Twinstrep-CapZβ or Twinstrep-CapZβ-ΔC34, MYC-RAB5A, and Flag-Rabex-5, and the cell lysates were then incubated with Strep-Tactin Sepharose. The pulldowns were subjected to immunoblotting analysis against the indicated antibodies. (**G**) HEK293 cells were transfected with Twinstrep-CapZβ or Twinstrep-CapZβ-ΔC34, and Flag-Rabaptin-5, and the cell lysates were then incubated with Strep-Tactin Sepharose. The pulldowns were subjected to immunoblotting analysis against the indicated antibodies. (**H, I**) HEK293 cells were transfected with Twinstrep-CapZβ(Δ148–277) (**H**) or Twinstrep-CapZβ(Δ1–147) (**I**), and Flag-Rabaptin-5, and the cell lysates were then incubated with Strep-Tactin Sepharose. The pulldowns were subjected to an immunoblotting analysis against the indicated antibodies. The blots, images, and graphs represent data from at least three independent experiments. The data are presented as mean ± SD. The difference between the two groups was calculated using two-tailed Student's t-test. Differences were considered statistically significant when p<0.05. *** p<0.001. Scale bar, 5 μm. MCC, Manders colocalization coefficient.

The online version of this article includes the following figure supplement(s) for figure 6:

**Source data 1.** Original immunoblots.

**Source data 2.** Data for 6C–6E.

**Figure supplement 1.** The role of the actin-binding motif of CapZ in endocytosis.

**Figure supplement 1—source data 1.** Original immunoblots.

**Figure supplement 1—source data 2.** Data for 1B.

CapZβ-GFP showed both diffused and puncta expression pattern; and the colocalization between CapZβ-ΔC34-GFP and RFP-RAB5 was significantly weaker than that for CapZβ-GFP and RFP-RAB5 (*Figure 6E*). Thus, these results indicate that the actin capping activity of CapZis essential for its role in endosomal trafficking. To further understand the role of actin capping activity of CapZ in the regulation of endosome maturation, we assessed the interaction of CapZβΔC34 with actin, RAB5, and RAB5 effectors by Co-IP experiments. We showed that not only the binding of β-actin with CapZβΔC34 was abolished, but also the interaction between CapZβΔC34 and RAB5 was markedly diminished (*Figure 6F*). However, this C34 deletion had little effect on the interaction between CapZβ and the RAB5 effectors such as Rabex-5 (*Figure 6F*), Rabaptin-5, and VPS34 (*Figure 6G*). Similarly, the deletion of the C terminal 148–277 amino acids of CapZβ did not affect its interaction with Rabaptin-5 (*Figure 6H*), whereas the deletion of the N-terminal 1–147 amino acids of CapZβ abolished its interaction with Rabaptin-5 (*Figure 6I*). Interestingly, CK-636 (an inhibitor of the Arp2/3 complex) had no effects on CapZ's interaction with actin, RAB5, Rabex-5, Rabaptin-5, or VPS34 (*Figure 6—figure supplement 1D and E*). Taken together, these results suggest that CapZ recruits the RAB5 effectors via its N-terminal domain, and docks to the early endosomes by capping the barbed ends of F-actin via its C-terminal actin-binding motif.

## Discussion

During the early step of endocytosis, actin filaments facilitate the budding of RAB5-positive vesicles by acting as scaffolds for the plasma membrane invagination and providing force for scission of the vesicle against turgor pressure (*Girao et al., 2008*; *Bucci et al., 1992*; *McLauchlan et al., 1998*). The actin filaments might also help these internalized vesicles move away from the invaginated fusion site (*Taunton et al., 2000*; *Merrifield et al., 2002*). Unexpectedly, here we found that the increased F-actin density around endocytic vesicles caused by CapZ knockout inhibited the maturation of early endosomes, manifested by the accumulation of small endocytic vesicles in CapZ-knockout cells (*Figure 2C and D*, *Figure 2—figure supplement 1I*, and *Figure 3—figure supplement 1D*). Whereas the decreased F-actin density, caused by CapZ overexpression (*Figure 3A and B*), latrunculin A treatment (*Figure 3—figure supplement 1F*), CK-636 treatment (*Figure 4A*), or ARP2 knockout (*Figure 4C*), increased the size and/or maturation of early endosomes. Therefore, these results suggest that the actin filaments around the endocytic vesicles might act as the physical barrier to inhibit homotypic fusion of these vesicles. CapZ impedes actin filaments growth, which in turn can facilitate actin filaments to be depolymerized by other factors (e.g., cofilin), to facilitate the fusion. This possibility is further supported by the inability of CapZβ-ΔC34 (a mutant lacking actin capping

activity) to rescue the defects of endolysosomal trafficking in CapZβ-knockout cells (*Figure 6C and D*, and *Figure 6—figure supplement 1C*).

Besides binding to actin (*Figure 6F and G*), CapZ complex was associated with RAB5 (*Figure 1D and E*), and GST-RAB5 was able to pull down CapZ from cell extracts (*Figure 1F*). Interestingly, RAB5-GDP pulled down more CapZ from cell extracts than RAB5-GTP (*Figure 1F*), and RAB5 was less active in the CapZ-knockout cells when compared with the control cells (*Figure 2E*). These results suggest that CapZ might interact with RAB5-GDP to promote the production of RAB5-GTP. However, CapZ contains no known GEF domain, such as VPS9 or DENN (*Barr and Lambright, 2010*), and it did not directly interact with RAB5 in vitro (*Figure 1—figure supplement 1C*). Therefore, it is likely that CapZ activates RAB5 via other known RAB5 GEFs. Rabex-5, perhaps the best known of the RAB5 GEFs, normally functions with Rabaptin-5, to activate RAB5 and promote the maturation of early endosomes (*Barr and Lambright, 2010*; *Langemeyer et al., 2018*). Indeed, we demonstrated the presence of both Rabex-5 and Rabaptin-5 in the CapZ complex (*Figure 5A and B*, and *Figure 5—figure supplement 1A*), and CapZ directly interacted with Rabaptin-5 in vitro (*Figure 5D* and *Figure 5—figure supplement 1D*). CapZβ-ΔC34 also retained its binding capabilities toward Rabex-5 and Rabaptin-5, not actin and RAB5 (*Figure 6F and G*), whereas CapZβ(Δ1–147) lost its binding affinity with Rabaptin-5 (*Figure 6I*). In addition, when CapZβ was knocked out, the interaction or association between Rabex-5 or Rabaptin-5 and RAB5 was reduced (*Figure 5F-I*). In contrast, knockdown of either Rabex-5 and Rabaptin-5 did not affect the ability of CapZ to interact with Rabaptin-5 or Rabex-5, respectively (*Figure 5—figure supplement 1F and G*). These results suggest that when CapZ caps the F-actin filaments (via its C-terminal tail), it might function as a scaffold protein to carry Rabaptin-5 and Rabex-5 (via its N-terminal domain) to RAB5-GDP-positive early endosomes, which are associated with F-actin filaments. This could induce the production of RAB5-GTP, and the active RAB5 then recruits more Rabaptin-5, Rabex-5, and other effectors to initiate a positive feedback loop to further increase its activity.

In the CapZ-knockout cells, the number of early endosomes was significantly increased but their size was markedly smaller, when compared with the control cells (*Figure 2C and D*, *Figure 2—figure supplement 1I*, and *Figure 3—figure supplement 1D*). In addition, RAB5 activity was significantly decreased in CapZ-knockout cells than in the control cells (*Figure 2E*). These results suggest that CapZ is required for the fusion or maturation of early endosomes. Another explanation of the increased small endocytic vesicles in CapZ-knockout cells is that CapZ deficiency might also promote the fission of early endosomes. Interestingly, CapZ was found in the WASH complex. WASH was shown to facilitate the nucleation of actin onto endosomes to control the fission of early endosomes for endosomal receptor recycling. WASH knockdown markedly induced tabulation of the endosomes and inhibited transferrin recycling (*Derivery et al., 2009*). Of note, CapZ knockout inhibited transferrin recycling as well (*Figure 2—figure supplement 1G*).

In mammalian cells, both actin filament and microtubule serve as tracks for the movement of endocytic vesicles. Myosin-VI drives early endosomes along actin filaments for short-range movements near cell cortex; and the long-range endosome motility is associated with microtubules, which mediates the movement of endosomes toward the perinuclear region (*Girao et al., 2008*; *Soldati and Schliwa, 2006*). However, a switch from microfilament- to microtubule-based movements during endosome translocating from cell surface to cell center remains unknown. Interestingly, CapZ knockout did not disrupt the move of the immature early endosomes away from the cell cortex, but it significantly slowed down the motility, shortened the moving distance, and disrupted the moving direction of these immature vesicles (*Figure 3F*). Therefore, CapZ might decrease the density of actin filaments around endocytic vesicles to facilitate the attachment of these vesicles to microtubules to render their fast movement toward the perinuclear region. Interestingly, deletion of CapZ in yeast cells reduced the moving rate of endocytic vesicles from the membrane invagination sites, suggesting that actin meshwork around the membrane invagination facilitates the subsequent vesicle scission and its rapid movement away from the plasma membrane (*Kaksonen et al., 2005*).

Taken together, our results illustrate a dual-functional role of CapZ in endosomal trafficking: it controls actin filament density around immature early endosomes via its C-terminal domain to facilitate the homotypic fusion of the endocytic vesicles; and it functions as a scaffold protein via its N-terminal domain to recruit RAB5 effectors, for example, Rabex-5 and Rabaptin-5, to early endosomes, thereby establishing a positive feedback loop to induce early endosome maturation.

# Materials and methods

**Key resources table**

| Reagent type (species) or resource | Designation | Source or reference | Identifiers | Additional information |
|---|---|---|---|---|
| Antibody | APRC1B (Rabbit polyclonal) | Sigma-Aldrich | HPA004832 | IF (1:100) |
| Antibody | ARP2 (Rabbit polyclonal) | ProteinTech | 10922-1-AP | WB (1:2000) |
| Antibody | Pan Actin (Mouse monoclonal) | Cytoskeleton | AAN02-s | IF (1:500) |
| Antibody | CapZ $\alpha$ (Mouse monoclonal) | DSHB | mAb 5B12.3 | WB (1:100) |
| Antibody | CapZβ (Rabbit polyclonal) | ProteinTech | 25043-1-AP | WB (1:2000) |
| Antibody | EGFR (Mouse monoclonal) | Santa Cruz | sc-373746 | WB (1:100) |
| Antibody | EEA1 (Rabbit monoclonal) | CST | 3,288 | IF (1:100) |
| Antibody | Flag-tag (Mouse monoclonal) | Sigma-Aldrich | F3165 | WB (1:1000) |
| Antibody | HA-tag (Rat monoclonal) | Sigma-Aldrich | 11867423001 | WB (1:3000) |
| Antibody | His-tag (Mouse monoclonal) | ProteinTech | 66005-1-Ig | WB (1:3000) |
| Antibody | HSC70 (Mouse monoclonal) | Santa Cruz | sc-7298 | WB (1:10,000) |
| Antibody | LAMP1 (Rabbit monoclonal) | CST | 9,091 | IF (1:200) |
| Antibody | MYC-tag (Mouse monoclonal) | ProteinTech | 67447-1-Ig | WB (1:3000) |
| Antibody | N-WASP (Rabbit polyclonal) | Novus Biologicals | NBP1-82512 | IF (0.25–2 µg/ml) |
| Antibody | p16-Arc (Rabbit monoclonal) | Abcam | ab51243 | IF (1:100) |
| Antibody | Rabaptin-5 (Rabbit polyclonal) | ProteinTech | 14350-1-AP | WB (1:1000) |
| Antibody | Rabex-5 (Mouse monoclonal) | Santa Cruz | sc-166049 | WB (1:500) |
| Antibody | StrepMAB-Classic HRP conjugate (Mouse monoclonal) | IBA | 2-1509-001 | WB (1:3000) |
| Antibody | VPS8 (Rabbit polyclonal) | ProteinTech | 15079-1-AP | WB (1:1000) |
| Antibody | RAB5A (Rabbit polyclonal) | ProteinTech | 11947-1-AP | WB (1:1000) |
| Antibody | RAB5A (Rabbit monoclonal) | CST | 3547 | IF (1:200) |
| Antibody | RAB5A (Mouse monoclonal) | CST | 46449 | IF (1:200) WB (1:1000) |
| Chemical compound, drug | CK-636 | Selleck | S7497 | |

*Continued on next page*

*Continued*

| Reagent type (species) or resource | Designation | Source or reference | Identifiers | Additional information |
|---|---|---|---|---|
| Chemical compound, drug | Latrunculin A | Santa Cruz | sc-202691 | |
| Chemical compound, drug | EGF | Bio-Rad | PHP030A | |
| Chemical compound, drug | GDP-β-S | Sigma-Aldrich | G7637 | |
| Chemical compound, drug | GTP-γ-S | Jena Bioscience | NU-412-2 | |
| Other | Transferrin-Alexa Fluor 594 | Invitrogen | T13343 | (25 µg/mL) |
| Other | EGF-Alexa Fluor 488 | Invitrogen | E13345 | (1 µg/ml) |
| Other | Phalloidin-iFluor 647 | Abcam | ab176759 | (1:1000) in 30 µl stock solution |
| Other | Texas Red-X Phalloidin | Invitrogen | T7471 | (1:40) in 1.5 ml stock solution |
| Cell line (*Homo sapiens*) | HeLa cells | ATCC | CRL-CCL-2 | |
| Cell line (*H. sapiens*) | HEK293 cells | ATCC | CRL-1573 | |

## Cell culture and virus propagation

HeLa (CRL-CCL-2) and HEK293 (CRL-1573) cells were obtained from ATCC. The identity of these cells has been authenticated by STR, and they are negative for mycoplasma contamination. Where not otherwise specified, all cells were maintained in Dulbecco's modified Eagle's medium (DMEM; Invitrogen, 12800-017) containing 10% fetal bovine serum (Invitrogen, 10270-106) and 100 units/ml of penicillin/streptomycin (Invitrogen, 15140-122) at 5% $CO_2$ and 37°C.

## Antibodies

The primary antibodies were used as follows: GAPDH (Proteintech, 60004-1-Ig), CapZβ (Proteintech, 25043-1-AP), ARPC1B (Sigma-Aldrich, HPA004832), p16-Arc (Abcam, ab51243), N-WASP (Novus Biologicals, NBP1-82512), His-tag (ProteinTech, 66005-1-Ig), β-actin (ProteinTech, 60008-1-Ig), α-tublin (ProteinTech, 66031-1-Ig), VPS8 (ProteinTech, 15079-1-AP), $\lambda$-tublin (Sigma-Aldrich, T6557), HSC70 (Santa Cruz, sc-24), VPS34 (CST, 4263), CapZa1/a2 (Developmental Studies Hybridoma Bank, mAb 5B12.3), anti-pan actin (Cytoskeleton, AAN02-s), RAB5A (CST, 3547 or 46449), Calnexin (ProteinTech, 10427-2-AP), LAMP1 (CST, 9091), TGN46 (Bio-Rad, AHP500GT), EEA1 (CST, 3288), MYC-tag (ProteinTech, 67447-1-Ig), Flag-tag (Sigma-Aldrich, F3165), HA-tag (Sigma-Aldrich,11867423001), StrepMAB-Classic HRP conjugate (IBA, 2-1509-001), Rabex-5 (Santa Cruz, sc-166049), Rabaptin-5 (ProteinTech, 14350-1-AP), EGFR (Santa Cruz, sc-373746), goat anti-mouse IgG (H+L) secondary antibody (Invitrogen, 31430), goat anti-rabbit IgG (H+L) secondary antibody (Invitrogen, 31460), and goat anti-rat IgG (H+L) secondary antibody (Invitrogen, 31470).

## Reagents and chemicals

Transferrin-Alexa Fluor 594 (Invitrogen, T13343), EGF-Alexa Fluor 488 (Invitrogen, E13345), CK-636 (Selleck, S7497), Latrunculin A (Santa Cruz, sc-202691), Texas Red-X Phalloidin (Invitrogen, T7471), Phalloidin-iFluor 647 (Abcam, ab176759), EGF (Bio-Rad, PHP030A), GDP-β-S (Sigma-Aldrich, G7637), GTP-γ-S (Jena Bioscience, NU-412-2), ProLong Diamond Antifade (Thermo Fisher Scientific, P36970), RIPA lysis buffer (Beyotime, P0013), Bradford (Bio-Rad, 5000006), Fatty-free bovine serum albumin (BSA; Sangon Biotech, A602448-0050), anti-flag magnetic beads (Bimake, B26101), Strep-Tactin Sepharose (IBA Lifesciences, 12846436), Western Chemiluminescent HRP Substrate (Millipore,

WBKLS0500), Ni-NTA Agarose (Qiagen, 30210), and Glutathione Sepharose High Performance (GE, 17527901).

## Immunoblotting

Total protein lysates were prepared in RIPA buffer and quantified using the Bradford Protein Assay. Proteins were separated by SDS-PAGE gels, transferred onto PVDF membranes, and blocked with 3% BSA in TBST (20 mM Tris base, 137 mM NaCl, 0.1% Tween-20, pH 7.6) for 1 hr. The membranes were probed with primary antibodies at 4°C overnight, and then washed with TBST followed by incubation with HRP secondary antibodies for 1 hr at room temperature. Immunoreactivity was visualized by chemiluminescent HRP substrate.

## Co-Immunoprecipitation

For co-IP performed in transfected cells, cells were lysed in NP-40 buffer before being incubated with anti-Flag magnetic beads (Bimake, B26101) or Strep-Tactin Sepharose (IBA, 2-1201-010) at 4°C overnight. The beads were then washed at least three times with lysis buffer before immunoblotting analysis. For endogenous co-IP, Hela cells were lysed in NP-40 buffer followed by incubation with the indicated antibody or normal IgG at 4°C overnight, followed by incubation with Protein A/G magnetic beads (Bimake, B23201) for another 2 hr. The beads were then washed at least three times with lysis buffer before immunoblotting analysis.

## CRISPR/Cas9 knockout

sgRNA sequences were designed by CHOPCHOP (*Montague et al., 2014*). To generate the CapZβ-KO cell lines, CapZβsgRNA (5'-CCAGGTCGATCAGGTCGCTG-3') was firstly cloned into pSpCas9(BB)-2A-GFP vector (*Ran et al., 2013*), and then sgRNA constructs were transfected into HeLa cells for 24 hr. GFP-positive cells were individually sorted and expanded. The knockout efficiency was determined by immunoblot analysis. Lentivirus Cas9 mediated gene knockout was achieved by lentiCRISPRv2 plasmids, which were inserted with VPS8 sgRNA (5'-CACCGCTTCGCTCGTCTTGGCACAG-3'), ARP2 sgRNA (5'-CACCGGGTGTGCGACAACGGCACCG-3'), or Rabaptin-5 sgRNA (5'-CACCGGTATAGA GAATCCGCAGAGA-3'). After lentivirus infection and antibiotics selection, the knockout efficiency was determined by immunoblot analysis.

## Plasmid constructions

Plasmids were constructed by conventional restriction enzyme-based cloning or by the Gateway recombination system (Invitrogen). All the constructs were verified by sequencing. The plasmids used in this study are listed in *Table 1*.

## GST pulldown assays

The recombinant GST-Rab5A protein was incubated with either GTPγS (1 mM) or GDPβS (1 mM) in binding buffer (150 mM NaCl, 5 mM EDTA, and 25 mM Tris-HCl, pH 7.4) at 37°C for 30 min, as

**Table 1.** Plasmids used in this study.

| | | |
|---|---|---|
| pCDNA3.1-Flag-Rabaptin-5 | pRK5-mRFP-FYVE | pCDNA3.1-HA-Rabex-5 |
| pCDNA3.1-MYC-RAB5A | plenti-CapZβ-GFP | pCDNA3.1-CapZβ-His6 |
| pRP-CapZβ-TwinStrep | pGEX-4-R5BD | pCDNA3.1-Flag-VPS34 |
| pCDNA3.1-Twinstrep-RAB5A | iRFP-FRB-RAB5A | CFP-FKBP-CapZβ |
| mRFP-RAB5A | pAGW-GFP-RAB5A | plenti-CapZβ-mcherry |
| pGEX4T-1-RAB5A | pGEX4T-1 | plenti-Flag-Rabex-5 |
| Plenti-GFP-ML1N2* | plenti-CapZβ | pRSFDuet-1-CapZβ/CapZα1 |
| plenti-CapZβ-Twinstrep | psPAX2 | pMD2.G |
| pGEX4T-1-Rabaptin-5 | pCDNA3.1-β-actin-Twinstrep | plenti-CapZβ(1-147)-Twinstrep |
| plenti-CapZβ(148-277)-Twinstrep | mCherry-ARPC1B | |

described previously (*Shinde and Maddika, 2016*). Then preloaded GST-RAB5A was incubated with HEK 293 cell lysates containing 10 mM MgCl$_2$, for 2–4 hr at 4°C. Thereafter, the beads were washed with wash buffer (150 mM NaCl, 2.5 mM MgCl$_2$, and 25 mM Tris-HCl, pH 7.4) and subjected to immunoblot analysis with respective antibodies.

## Recombinant protein purification

*Escherichia coli* BL21(DE3) pLysS cells (TransGen Biotech) transformed with expression vectors containing GST-tagged RAB5A, GST-tagged Rabaptin-5, or His$_6$-tagged CapZβ/ CapZα1 were grown at 37°C to an OD$_{600}$ of 0.6–0.8 and then induced with 0.5 mM IPTG at 16°C overnight. Induced cells were harvested by centrifugation at 4000 rpm and resuspended in lysis buffer A (50 mM Tris-HCl pH 7.5 and 150 mM NaCl) for GST-tagged proteins or lysis buffer B (50 mM Tris-HCl pH 7.5, 1 mM DTT, 1 mM EDTA, 1% Triton-x100 and 1 mM phenylmethylsulphonyl fluoride) for His$_6$-tagged proteins. Resuspended cells were lysed by the French press method and centrifuged at 18,000 rpm for 1 hr to remove cell debris. For GST-tagged proteins, supernatants were incubated with Glutathione Sepharose (GE Healthcare) for 2 hr at 4°C, and nonspecific proteins were washed out with buffer C (50 mM Tris-HCl pH 7.5 and 500 mM NaCl) before elution with buffer D (50 mM Tris-HCl pH 8.0, 50 mM NaCl, and 15 mM reduced glutathione). For His$_6$-tagged proteins, supernatants were loaded onto a HisTrap excel column before washing with buffer E (50 mM Tris-HCl pH 7.5, 50 mM NaCl, and 20 mM imidazole). To elute the His$_6$-tagged proteins, buffer F (50 mM Tris-HCl pH 7.5, 100 mM imidazole, and 50 mM NaCl) was used. GST-tagged proteins and His$_6$-tagged proteins were dialyzed with 3C protease-containing buffer (50 mM Tris-HCl pH 7.5, 50 mM NaCl, and 5 mM EDTA) to cleave off GST-tag and His$_6$-tag. The digestion mixtures were applied to Glutathione Sepharose or loaded to a HisTrap excel column to remove released GTS-tag or His$_6$-tag, and the flow-through fractions containing the target proteins were collected. The flow-through fractions were further purified by a HiTrap Q HP column and gel filtration chromatography using Superdex 75 or 200 pg 16/60 column (GE Healthcare). The purified proteins were analyzed by SDS-PAGE and concentrated to about 20 mg/ml using 10 or 30 kDa Amicon Ultra centrifugal filters (Millipore).

## Live-cell image

Cells were plated on 35 mm glass-bottom dishes (Ibidi, 181212/5) and transfected with RFP-RAB5 plasmids. After 24 hr, images were obtained with a Nikon A1HD25 high-speed and large Field of view confocal microscope using a 100× oil objective lens. RFP fluorescence was captured using 558 nm excitation filters. Images were captured for 3 min using the 'no delay' mode (the 'Real' interval between each frame is around 0.5 s), and after data acquisition, the raw images were processed with the Nikon NIS software. Briefly, the endosomes in each frame were detected by the bright different sizes method in the NIS-Elements AR software; the typical diameter and contrast parameters were set appropriately to ensure most objects were selected. The objects were then tracked with a constant speed motion model. The 'linking' modules were selected, which allowed three maximum gap sizes in tracks, to facilitate the same endosomes to be tracked between frames. The average speed is the average speed of each endosome in each frame. The average moving distance is the average moving distance of each endosome in each frame X (multiply) total frames.

## Immunofluorescence staining

Cells were fixed by 4% paraformaldehyde and permeabilized with 0.1% Triton X-100 in phosphate-buffered saline (PBS) for 15 min. Nonspecific antigens were then blocked by 1% BSA for 30 min at room temperature followed by incubation with the indicated primary antibodies at 4°C overnight. The next day, the cells were incubated with the appropriate fluorescence-conjugated secondary antibody for 1 hr at room temperature and washed with PBS before mounting coverslip with ProLong Diamond Antifade. Images were acquired with the Zeiss LSM 880 or Nikon A1HD25 confocal microscope and then analyzed with the ZEISS ZEN or ImageJ software. All the confocal images were taken and analyzed in a single plane if not otherwise specially stated.

ZEISS ZEN software was used to examine RAB5-positive endocytic vesicles, and the min/max size of endosomes were set to the same value in the same set of experiments (e.g., 30,000 nm$^2$(min)/10,000,000 nm$^2$(max)). The intensity cutoffs were set automatically through Otsu threshold

methods with manual checks to ensure most endosomes to be appropriately circulated. The overlapping endosomes were separated by the watersheds method.

Super-resolution microscopy was performed with a Nikon N-SIM system in 3D structured illumination mode on an Eclipse Ti-E microscope equipped with a 100×/1.49 NA oil immersion objective, 488 and 561 nm solid-state lasers (coherent), and an EM-CCD camera (DU-897, Andor Technology). To generate a SIM image, 15 raw images acquired at different orientation of the structured illumination were collected and then reconstructed into a final super-resolved image using the N-SIM module in NIS-Elements software. To generate a 3D SIM image, z-series imaging of cells were taken with a 0.12-µm step size. The Z stack was then reconstructed into a 3D image using the N-SIM module in NIS-Elements software.

## EGF endocytosis and EGFR degradation assay

For EGFR degradation, cells maintained in six-well plates were washed with PBS before incubating in serum-free DMEM medium overnight at 37°C. EGFR endocytosis was stimulated by the addition of EGF (100 ng/ml) in DMEM. At indicated time points after EGF stimulation, the cells were lysed in SDS–PAGE sample buffer, and subjected to immunoblot analysis with an anti-EGFR antibody.

GST-R5BD pulldown assay- The preparation of GST-R5BD linked to glutathione-agarose beads was performed as described previously (*Qi et al., 2015*). Cells were lysed in lysis buffer (25 mM HEPES-KOH (pH 7.4), 100 mM NaCl, 5 mM $MgCl_2$, 0.1% NP40, 10% Glycerol, and 1 mM DTT). Equal amounts of protein were then added to the GST-R5BD beads and incubated at 4°C for 2 hr. The beads were washed with lysis buffer without detergent and then boiled in sample loading buffer, followed by immunoblot analysis with the anti-RAB5A antibody.

## Biolayer interferometry

The binding affinity between CapZ and Rabaptin5 was measured by BLI on an OctetRED96 instrument (ForteBio). The assays were performed on black 96-well plates (Greiner Bio-One) with a shaking speed of 1000 rpm at 25°C. CapZ was biotinylated by EZ-Link Sulfo-NHS-LC-Biotinylation Kit (Thermo Fisher Scientific) in binding buffer (1× PBS). The streptavidin (SA) biosensors (ForteBio) were blocked for 10 min with 5% skimmed milk powder in 1× PBS, and washed with PBS for three times (10 min per time) to reduce nonspecific interactions. SA biosensors with immobilized biotinylated CapZ (200 µg/ml) were then exposed to various concentrations of Rabaptin-5 protein, and one well without Rabaptin5 protein was used as a negative control. The biotinylated Rab5A-loaded SA sensor exposed to Rabaptin5 protein (50 µM) was used as a positive control, and was run for each experiment. Data were analyzed by Octet Data Analysis HT 10.0.3.7 software (Pall ForteBio LLC). The binding affinity constant was generated by fitting both the association and dissociation curves to a 1:1 binding model.

## Isothermal titration calorimetry

ITC experiments were performed on a MicroCal PEAQ-ITC system (Malvern Panalytical) at 25°C. All proteins were separately exchanged into ITC buffer (50 mM Tris-HCl pH 7.5, 150 mM NaCl, and 5 mM EDTA) by extensive dialysis at 4°C for 16 hr. Rabaptin-5 (400 µM) and GST-RAB5A (500 µM) were respectively injected into 300 µl ITC buffer containing 40 µM CapZ complex via a 40-µl syringe with 19 injections of 2 µl each. The same experiment was performed with injections of 2 µl GST-RAB5A (300 µM) into 300 µl Rabaptin-5 (30 µM) as a positive control. A background titration was carried out by respectively titrating Rabaptin-5 (400 µM) or GST- RAB5A (500 µM) to the sample cell containing 300 µl ITC buffer. Titration curves were fitted by MicroCal PEAQ-ITC analysis software (Malvern Panalytical) using a one-site binding model.

## Image colocalization analysis

Manders colocalization coefficient (MCC) or Pearson's correlation coefficient (PCC) are two major metrics of colocalization analysis of two probes. In general, MCC or PCC methods based on the workflow described by *Dunn et al., 2011* were performed on single plane images (*Dunn et al., 2011*). Briefly, to analyze the colocalization between RAB5 and its effectors (e.g., EEA1, Rabex-5, or Rabaptin-5), PCC was used because RAB5 and its effectors mostly co-occur on the same cellular structures (early endosomes), and their fluorescent signals fit a single linear relationship, which satisfies the requirements of PCC. On the other hand, if images were required to set a threshold to distinguish an object from a

background (e.g., CapZ), or only a small proportion of two objects (such as p16-Arc vs. RAB5, CapZ vs. RAB5, or RAB5 vs. Actin) were codistributed in the same compartment, MCC was used.

## Statistical analysis

All statistical analysis was performed by Graphpad software. The difference between the two groups was assessed using an unpaired and parametric Student's t-test. The endosome size data in control, CapZ-knockout, or ARP2-knockout cells could also be transformed with log10 before t-test. Differences were considered statistically significant when $p < 0.05$.

## Acknowledgements

The authors thank members of Yue lab for their advice on the preparation of this manuscript. This work was supported by Hong Kong Research Grant Council (RGC) grants (11101717 and 11103620), NSFC (21778045, 32070702, and 82161128014), ITF (MRP/064/21 and GHP/097/20GD), research grants from Shenzhen Science and Technology Innovation Committee (SGDX20201103093201010, JSGG20200225150702770, and JCYJ20210324134007020).

## Additional information

### Funding

| Funder | Grant reference number | Author |
|---|---|---|
| Research Grants Council, University Grants Committee | 11101717 | Jianbo Yue |
| Research Grants Council, University Grants Committee | 11103620 | Jianbo Yue |
| National Natural Science Foundation of China | 21778045 | Jianbo Yue |
| National Natural Science Foundation of China | 2070702 | Jianbo Yue |
| Science, Technology and Innovation Commission of Shenzhen Municipality | JCYJ20160229165235739 | Jianbo Yue |
| Science, Technology and Innovation Commission of Shenzhen Municipality | JCYJ20170413141331470 | Jianbo Yue |

The funders had no role in study design, data collection and interpretation, or the decision to submit the work for publication.

### Author contributions

Dawei Wang, Conceptualization, Data curation, Formal analysis, Investigation, Writing – original draft; Zuodong Ye, Jingting Yu, Lihong Huang, Data curation, Investigation, Methodology; Wenjie Wei, Investigation; Hongmin Zhang, Supervision; Jianbo Yue, Conceptualization, Data curation, Formal analysis, Funding acquisition, Resources, Supervision, Writing – original draft, Writing – review and editing

### Author ORCIDs

Dawei Wang (iD) http://orcid.org/0000-0002-7868-775X
Jingting Yu (iD) http://orcid.org/0000-0002-2631-8434
Hongmin Zhang (iD) http://orcid.org/0000-0003-4356-3615
Jianbo Yue (iD) http://orcid.org/0000-0001-6384-5447

### Decision letter and Author response

Decision letter https://doi.org/10.7554/eLife.65910.sa1
Author response https://doi.org/10.7554/eLife.65910.sa2

## Additional files

### Supplementary files
• Transparent reporting form

### Data availability
All data generated or analysed during this study are included in the manuscript and supporting files.

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
