## [Editor Report]

In this article, Yue and colleagues have uncovered role of the actin binding and capping protein CapZ in fusion and maturation of early endosomes. They show that the actin capping factor localizes to early endosomes and regulates F-actin density on early endosomes. The authors also find another role of CapZ in facilitating recruitment of Rabaptin-5 and Rabex-5 on early endosomes and promote Rab5 activation. The work provides an unexplored and somewhat unexpected perspective on how endosomal localized actin polymerization impact on endosome maturation and fusion.

---

## [Decision Letter]

Thank you for submitting your article "Capping protein regulates endosomal trafficking by capping F-actin on endocytic vesicles and recruiting RAB5 effectors" for consideration by *eLife*. Your article has been reviewed by 3 peer reviewers, one of whom is a member of our Board of Reviewing Editors, and the evaluation has been overseen by Anna Akhmanova as the Senior Editor. The following individual involved in review of your submission has agreed to reveal their identity: Giorgio Scita (Reviewer #2).

The reviewers have discussed their reviews with one another, and the Reviewing Editor has drafted "essential revisions" section incorporating comments from all the three reviewers to help you prepare a revised submission. The individual recommendations from the three reviewers are also listed below.

Essential revisions:

The manuscript by Wang et al., describes the role of actin capping protein CapZ in early endosome maturation and cargo trafficking. CapZβ knock out HeLa cells show defect in endosome maturation and a delay in EGFR degradation. The early endosome maturation defects in CapZβ-knockout cells are partially rescued by inhibiting actin polymerization. CAPZ is also shown to mediate a direct interaction with basic components of the RAB5-endosomal fusion machinery and to control through them RAB5A activation and endosome fusion, with an impact on Zika virus infection. Overall, the experiments are well conducted and documented and support the mode of action of CapZ proposed on endosome maturation. However, there are several concerns that must be adequately addressed by additional experimentation to support the major conclusions of this study:

Please find the points covering the essential revision requirements below:

1) CapZ is an essential protein in almost all actin dependent mechanisms, and as a consequence the KO of CapZ will undoubtedly affects a number of different actin-dependent processes that influence several conclusions of the paper. For example, The KO of CapZ is known to influence cell morphology (Sinnar et al., MBOC 2014). Thus, authors should investigate parameters such as cell spreading and/or cell size in CapZ KO Hela cells. Several endosomes metrics should also be weighted by cell size, such as the number of endosomes. This could markedly change the significance of several findings. Further the rate of constitutive endocytosis should be measured in CapZβ-knockout cells.

2) A central assertion of the paper is that CapZ knockout might increase the formation of new actin branches to entangle around these immature RAB5-positive endocytic vesicles, and this might impede their fusion and result in the accumulation of small endosomes" (lines 113-115). However, there is surprisingly no imaging experiments performed with actin markers.

a) High resolution imaging of actin and Rab5 endosomes should provide better estimation of excessive actin polymerization on these endosomes. Subcellular fractionation experiments can also be performed to test actin levels in the endosomal fractions under these conditions.

b) Are the enlarged endosomes seen in the ArpC1B KO completely devoid of actin?

c) Can the impaired motility of Rab5 positive vesicles be rescued by Arp2/3 complex inhibition and or actin depolymerization?

3) It is unclear whether there is any preference for CAPZ binding to early (RAB5 positive) versus Late Rab7- or RAB4/RAB11-recycling endosomes. The Golgi-like localization of GFP-CapZ suggests it localizes to lysosomes or late endosomes. To that end, please quantify the degree of colocalization of CapZ with Lamp1 (images are in the supplementary Figure 1B but not quantified), Rab7 and Rab11/Rab4. Rab4 is particularly important since it can directly interaction with Rabaptin-5 (Vitale et al., EMBO J. 1998).

4) Similarly, while the co-immunoprecipitation between CAPZ and RAB5 is documented, it is unclear whether this interaction is specific (do other RABs, and specifically RAB7 or RAB11, interact with CAPZ?) and whether there is any preferential binding to the GTP or GDP bound form of RAB5. One key tenet, here, is that CapZ interaction with RAB5 and endosome should be mediated by the branched actin network around these structures. Does inhibiting actin polymerization either with Latrunculin or CK-639 impair the interaction between CapZ and RAB5 but not some of the component of endosomal fusion machineries reported in Figure 3 (as later shown using CapZ mutants)? Further interaction between Rab5 GEF and Rab5 also seems to be dependent on CapZ actin capping activity. The potential reason/ basis of these observations can be added to the Discussion section of the manuscript.

5) In Figure 1F, the trafficking dynamics of fluorescently labeled EGF is examined in CAPZ null versus reconstituted cells. CAPZ removal results in a persistent accumulation of EGF in intracellular vesicles, leading the author to conclude that EGFR degradation is severely delayed by CAPZ Knocked down. However, to support this contention, EGFR receptor levels (also through western blotting) should be monitored over time following a pulse of EGF. Indeed, the accumulation of EGF/EGFR could be also caused by inhibition of recycling begging the question as to whether CapZ is also involved EGF/EGFR recycling? Additionally, one might expect EGFR signalling to be perturbed given the tight relationship between endosomal fluxes, number and size and signalling (as shown by recent work by Zerial group that proposed the notion of endosomes act as quanta signalling platforms thereby influencing ERK1/2 activity).

6) ArpC1B-mCherry does not appear localize to the lamella in Figure 2B and suggests that ectopic expression of ArpC1B-mCherry does not faithfully recapitulate the localization of the endogenous Arp2/3 complex. a. As a positive control, the authors should show that ArpC1B-mCherry colocalizes with endogenous Arp2/3 complex.

b. Figure 2C shows N-Sim of ArpC1B-mCherry with endogenous Rab5. This analysis would be best served by comparing this colocalization with the colocalization of endogenous Arp2/3 complex and endogenous Rab5.

c. The representative images in 2B and 2C do not show the same focal plane for WT and KO cells and raises questions about the fairness of the colocalization analysis. Please show representative images of WT and CapZbKO cells at the same focal plane and indicate how the colocalization analysis was performed.

d. Also, how well does the overexpressed ArpC1B-mCherry colocalize with GFP-CapZ, and does CapZ KO disrupt ArpC1B-mCherry localization?

e. An alternative to the conclusion that inhibition of Arp2/3 complex leads to larger endosomes is that inhibition of Arp2/3 complex leads to impaired emergence of nascent endosomes from the plasma membrane. This would reduce the population of small vesicles and as a consequence would inherently overly inflate the contribution of larger vesicles to the overall mean vesicle area. Again, showing the distribution of endosome sizes would be helpful in distinguishing these possibilities.

f. Localization of the Arp2/3 complex to these vesicular populations does not imply that it is active. Please determine, if this CapZ KO-driven localization is accompanied by enrichment of endogenous N-WASP on these enlarged vesicles.

7) The in vitro binding of Rabaptin-5 to CapZ is unconvincing, mainly because of the low heat of interaction coupled to the high concentrations of proteins used in ITC experiment. To that end, the authors should be extra critical of the ITC results and provide a secondary measure of interaction (like a pulldown with purified proteins). Based on the evidence presented in Figure 2E and 2F, it would seem that CapZ KO-driven loss of Rabaptin-5 is the main driving force behind the small Rab5-positive vesicles in CapZ KO cells since this effect could not be rescued with either CK-636 nor ArpC1B KO. However, endosome number/size should be quantified for CK-636/latrunculin-treated cells where Rabaptin-5 has been ablated.

8) The authors examine the potential pathophysiological consequences of the loss of CAPZ using Zika Virus infection. However, this part of the manuscript is not thoroughly investigated. For instance, CapZ has also been implicated in autophagy, where ablation of CapZ leads to impaired autophagosome formation (Mi et al., 2015). Additionally, autophagy is well-established to promote ZIKV infection (as reviewed in Gratton et al., Int. J. Mol Sci 2019 and Chiramel & Best, Viruses 2018). Justifying the author's conclusion would requires distinguishing the contributions of CapZ in endosomal maturation and autophagy. Thus, these results are best served for another publication where this conclusion can be thoroughly justified. Instead, examining the physiological signalling and trafficking biology of an endocytic cargo such as EGFR receptor might provide more insights onto the impact of CapZ on endosome maturation.

*Reviewer #1:*

Wang et al., investigated the role of actin Capping factor, CapZ, particularly, CapZβ in regulating endosomal trafficking. The authors show that CapZβ localizes to Rab5 -positive endosomes and also interacts with Rab5. CapZβ KO HeLa cells show defects in endosome maturation and a delay in EGFR degradation. The early endosome maturation defects CapZβ-knockout cells are partially rescued by inhibiting actin polymerization. Notably the authors also observe that active Rab5 was lower in the CapZβ-knockout cells. Recruitment of Rabex-5 and Rabaptin5 to Rab5 endosomes is reduced in CapZβ-knockout cells, explaining the lower activation of Rab5 in these cells. Finally, the authors show that endosome maturation role of CapZβ is crucial for ZIKA virus infection, as CapZβ-knockout delays the release of viral genome from the endosomes.

The experiments in this manuscript are rigorous, well controlled and well-analyzed. However, it is unclear whether the effect on Rab5 activation and endosome maturation is a direct role of CapZ pool localized at the early endosomes. To address this question, it would be important to determine the rate of endocytosis in CapZβ knockout cells to conclude that CapZβ acts downstream of cargo endocytosis. Also, it would be important to determine how actin-capping activity of CapZβ is mediating the endosomal localization of Rab5 GEF, Rabex-5 and Rabaptin-5.

The endosomal localization of actin capping factor, as reported here is an interesting observation. This study suggests that localized polymerization and depolymerization of actin has important consequences for endosomal maturation and transport.

The manuscript by Wang et al., describes the role of actin capping protein CapZ in early endosome maturation and cargo trafficking. The authors show that CapZβ localizes to Rab5 -positive endosomes and also interacts with Rab5. CapZβ KO HeLa cells show defect in endosome maturation and a delay in EGFR degradation. Colocalization of ARPC1B (a subunit of the Arp2/3 complex) with Rab5 is increased in CapZβ-knockout cells and motility of Rab5 vesicles is reduced. The early endosome maturation defects CapZβ-knockout cells are partially rescued by inhibiting actin polymerization. Notably the authors also observe active Rab5 was lower in the CapZβ-knockout cells. Recruitment of Rabex-5 and Rabaptin5 to Rab5 endosomes is reduced in CapZβ-knockout cells, explaining the lower activation of Rab5 in these cells. Finally, the authors show that endosome maturation role of CapZβ is crucial for ZIKA virus infection, as CapZβ-knockout delays the release of viral genome from the endosomes.

The manuscript is interesting but there are few major concerns (as described below) that pertain to how a general actin regulatory factor such as CapZ (that alters the status of actin cytoskeleton at the whole cell level) has a specific role in early endosome maturation.

– Figure 1A and Supplementary Figure 2A- It appears that a small pool of CapZβ localizes to early endosomes while most of the protein shows an actin-like localization. Further actin cytoskeleton is strikingly altered in CapZβ-knockout cells. The authors need to alleviate the concern that effects shown on endosome maturation are not an indirect effect of altering the actin cytoskeleton at the whole-cell level. What is the rate of constitutive endocytosis in CapZβ-knockout cells? From the EGF data (Figure 1F) it appears that even endocytosis of the receptor is reduced in CapZβ-knockout cells.

– Line # 269, the authors state "We, thus, speculated that CapZ might carry Rabaptin-5 and/or Rabex-5 to RAB5-positive early endosomes by capping the barbed end of

F-actin." It is confusing and not clear how CapZ actin capping activity regulates recruitment of Rab5 GEF and therefore Rab5 activation. The CapZΔC34 mutant continues to interact with Rabex-5 and Rabaptin-5 but is unable to recruit Rab5 and actin in this complex (Figure 4F). The underlying basis of these observations is not clear. How does excessive actin accumulation prevent Rab5 GEF interaction with Rab5? Does Rabex/ Rabaptin-5 have any particular domains that respond to actin levels?

– It would be relevant to test whether actin polymerization and Rab5 activation are directly linked, for instance, what is the status of Rab5 activation in ARPC1B knockout cells or cells treated with latrunculin A or CK-636? Similarly, what is the localization of Rabex-5 and Rabaptin-5 in ARPC1B knockout cells or cells treated with latrunculin A or CK-636? Does Rabex-5 and Rabaptin-5 association with Rab5 and recruitment to endosomes depends on actin polymerization? Does constitutively active Rab5 rescue the endosome maturation defects in CapZβ-knockout cells and what is status of actin in these cells?

– I also find the evidence that excessive actin accumulation leads to block/delay in endosome maturation far less convincing. Probably live-cell imaging of actin and Rab5 endosomes should provide a better estimation of their continued association in control and CapZβ-knockout cells. Further, by using inhibitors such as latrunculin A in live-cell imaging, authors can measure Rab5 endosome size/motility before and after treatment. Subcellular fractionation experiments can also be performed to test actin levels in the endosomal fractions under these conditions.

– It is important to show the endogenous distribution of CapZβ as only overexpression approach is used. Is CapZβ selectively localized to early endosomes or to other organelles?

*Reviewer #2:*

In this manuscript, a role of the actin binding and capping protein CapZ is investigated. Using CRISPR and cellular biochemistry approaches, it is cogently shown that CAPZ accumulates on early endosome, in conjunction with F-actin. There, through loss of function experiments CapZ is shown to regulate the amount of F-actin around the early endosome. Its removal causes a hyper-polymerization of actin that impedes endosome motility, fusion and possibly maturation. CapZ is shown to bind directly to the Rab5 effector Rabex-5, possibly through region that are independent from its ability to cap actin filaments. Functionally, removal of Capz reduces the efficiency of Zika Virus infection possibly by trapping the viral particle in immature endosomes.

Overall, the experiments are well conducted and documented and support the mode of action of CapZ proposed on endosome maturation, albeit the mechanistic aspects and specifically those related to the interaction between CapZ and Rabex-5 are not developed or exploited to their potential. Similarly, there are very few experiments testing the impact of CapZ perturbations on endosomal trafficking and signalling in physiological conditions. In this latter respect, exploiting the plethora of information on EGFR trafficking and signalling would seem an easy road to take to complete and expand this study.

In figure 1 and SI1, the association between CAPZbeta and early endosomes is convincingly documented. It is however unclear whether there is any preference for CAPZ binding to early (RAB5 positive) versus Late Rab7- or RAB4/RAB11-recycling endosomes. Similarly, while the co-immunoprecipitation between CAPz and RAB5 is documented, it is unclear whether this interaction is specific (do other RABs, and specifically RAB7 or RAB11, interact with CAPZ?) and whether there is any preferential binding to the GTP or GDP bound form of RAB5. One key tenet, here, is that CapZ interaction with RAB5 and endosome should be mediated by the branched actin network around these structures. If so, one would expect that inhibiting actin polymerization either with Latrunculin or CK-639 impair the interaction between CapZ and RAB5 but not some of the component of endosomal fusion machineries reported in Figure 3 (as later shown using CapZ mutants).

– In Figure 1F, the trafficking dynamics of fluorescently labeled EGF is examined in CAPZ null versus reconstituted cells. CAPZ removal results in a persistent accumulation of EGF in intracellular vesicles, leading the author to conclude that EGFR degradation is severely delayed by CAPZ Knocked down. However, to support this contention, EGFR receptor levels (also through western blotting) should be monitored over time following a pulse of EGF. Indeed, the accumulation of EGF/EGFR could be also caused by inhibition of recycling begging the question as to whether CapZ is also involved EGF/EGFR recycling?

– In Figure 2A, there a number of interesting and convincing experiments. One aspect that might deserve some attention is the impact of CapZ ablation on early endosome motility, which is severely compromised. Is this a specific effect on early endosome or the large increase in actin network documented in supplementary figure 2 F is the culprit here? In other words, some caution in interpreting this finding should be used as it remains unclear whether the purported increase F-actin around endosome is the cause of their reduced motility.

– In Figure 3, a direct interaction between CAPz and Rabex-5 one of the effectors of RAB5 in early endosome is shown. These findings are somewhat surprising, but interesting and well documented. They also beg the question as to the nature of this molecular interactions and of the relative contribution of CAPz to endosome biology through actin branched polymerization or binding and assembly of the fusion machinery. The author made progress in this direction by showing that a CAPz mutant no longer able to bind to barbed ends fails to restore CAPz KO phenotype, is no longer able to bind to RAB5 but retain the ability to associate with Rabex-5. Conversely a n-terminal N-terminal 1-147 deletion abolishes the association with Rabex-5. What remains unclear is whether the latter mutant is still capable of binding to barbed ends and whether its expression can restore the phenotypes due to loss of CapZ. Thus, the interaction between Capz and Rabex-5 potentially uncover interesting and unexpected mechanistic scenario, which are however not investigated in sufficient details.

– Finally, the authors examine the potential pathophysiological consequences of the loss of CAPz using Zika Virus infection. This is fine; however, it would seem that examining the physiological signalling and trafficking biology of the EGFR receptor might provide significant more insights onto some of the missing aspects of the model proposed. For example, is EGFR degradation, recycling affected? Additionally, if the model is correct one might expect EGFR signalling to be perturbed given the tight relationship between endosomal fluxes, number and size and signalling (as shown by recent work by Zerial group that proposed the notion of endosomes act as quanta signalling platforms thereby influencing ERK1/2 activity). Exploiting and investigating these aspects would go a long way in establishing the consequences of the purported impact of CapZ on endosome maturation

*Reviewer #3:*

The proposed manuscript, titled "Capping protein regulates endosomal trafficking by capping F-actin on endocytic vesicles and recruiting RAB5 effectors," attempts to uncover a potential role for the barbed-end actin capping protein, CapZ, in the fusion and maturation of early endosomes. To that end the authors show that ablation of CapZ leads to the accumulation of small Rab5-positive vesicle that are believed to represent unfused early endosomes. In an attempt to establish the underling mechanism, the authors find that loss of CapZ leads to enhanced recruitment of Arp2/3 complex to Rab5 positive vesicles, which is assumed to be driving unregulated actin assembly on these endosome, that ultimately impairs their fusion. The authors also find that CapZ may directly interact with the Rab5 effector, Rabaptin-5, which in itself could potentiate the recruitment of Rab5. Finally, they also demonstrate that impairing endosome maturation through CapZ ablation can impair infection of ZIKV RNA.

The authors use a wide array of experimental approaches to put forward a number of interesting observations that could have considerable impact on understanding how the actin dynamics is coupled to endosomal maturation. However, based on the current data, it is unclear if the loss of CapZ leads to impaired fusion of nascent endosomes with the early endosomes because of excessive actin assembly on the early endosomes or it it is due to a loss of Rab5 activity and/or Rab5 dependent vesicle fusion factors, like Rabaptin-5. Also, it's unclear if the dependence of Rab5 localization to early endosomes is entirely influenced by a local populations of CapZ on endosomes emerging from the plasma membrane, or if it's a result of global changes in the actin dynamics and membrane trafficking in response to knocking-out a crucial element of the actin cytoskeleton. This is particularly important when considering viral infectivity, which relies on the convergence of multiple actin cytoskeleton dependent processes.

1. Data presentation though-out the figures need to be improved and better explained. As it stands, it is not evident that anyone could reproduce the data presented in the paper.

a. Error bars need description. I'm assuming they are presented as SEM, however this could be entirely inappropriate if the data is not normally distributed.

b. Please report the individual measurements in addition to the bar graphs so the reader can accurately assess and confirm the appropriateness of the statistical treatments. Also, presenting the endosome number and size distributions would be considerably more informative than the current mean and SEM presentation.

c. For imaging analysis:

i. Please indicate, for all experiments, whether conventional or SIM-mode was used.

ii. Please describe and justify the nature of the images analyzed (single-plane, all-planes, max projection, etc.) and indicate if only selected regions were analyzed within in those images.

iii. There are several instances where Mander's overlap coefficient is reported, but not Pearson's, and it's not clear why. This is particularly worrisome, because Mander's is heavily weighted by pixel intensity and small endosomes will have considerably less Rab5 intensity compared to large endosomes. In almost all cases, Pearson's correlation coefficient should be used. Importantly, please indicate on the Y-axis of each correlation bar graph the type of correlation.

iv. Please describe the criteria (min/max size, intensity cut-offs, circularity, etc..) used to automatically identify endosomes for the image analysis and how overlapping endosomes were resolved.

v. Please indicate how single endosomes were tracked in Figure 2D. There is no method listed for this. It's not enough to say, "performed using NIS-Elements AR software". What steps were taken to ensure the same endosome was being tracked between frames? This is particularly important when considering there is an abundance of small endosomes in CapZ KO cells, which will be less bright compared to those in WT cells.

vi. Was the Live-cell imaging performed in super-resolution mode? What was the 'real' acquisition speed of the experiment (this is missing in the caption and in the figure legend). Again, this is important for faithfully tracking endosomes.

vii. Please explain how "Average speed" and "Average moving distance" was calculated.

viii. What was the average time a single endosome could be tracked?

ix. It's stated that endosomes from 14 cells were tracked in the figure caption, but it's not stated how many endosomes were tracked per cell.

d. For ITC analysis:

i. Please report binding model used for the fit, and ALL fit parameters, along with fit errors. Indicate which fit parameters were fixed.

ii. Indicate if the experiment was performed only once or multiple times, If the latter, report the experimental average and associated error.

2. The paper assumes all Rab5-positive vesicles but does not explore the possibility that it may influence other endosomal populations. The authors argue that Rab5 is recruited indirectly by CapZ to early endosomes through Rab5 effectors and would therefore suggest the small Rab5-postivie vesicles observed in the CapZ KO cells would not be early endosomes. Indeed, the authors show in Figure 5E that Rab5A poorly colocalizes with EEA1 in CapZ KO cells. Furthermore, EGF trafficking assays show EGF-positive endosomes swelling as they accumulate in the CapZKO cells, suggesting that it's not early endosome formation that is impaired, rather the Rab5-to-Rab7 switch or lysosomal fusion is impaired. Therefore, the authors should present additional evidence to support their conclusion about the role of CapZ on early endosome maturation.

a. Given the diffuse localization of CapZ, It's unclear if the colocalization of CapZ with EEA1 or Rab5 is significant without a basis of comparison. An excellent positive control would be to show the colocalization of EEA1 with Rab5 (as done in Figure 3E).

b. With that in mind, how do these sizes of these Rab5 positive vesicles compare with EEA1 positive vesicles in WT and CapZ KO Cells.

c. The Golgi-like localization of GFP-CapZ suggests it localizes to lysosomes or late endosomes. To that end, please quantify the degree of colocalization of CapZ with Lamp1 (images are in the supplementary Figure 1B but not quantified), Rab7 and Rab11/Rab4. Rab4 is particularly important since it can directly interaction with Rabaptin-5 (Vitale et al., EMBO J. 1998). Also, do any of these markers pulldown IP with CapZ?

d. Despite CapZ appearing to have strong colocalization with Lamp1, the authors consider this to be "weak" colocalization? What is the basis for considering this to be weak colocalization? It should be noted that there is a precedent for a role of CapZ with autophagosomes (Mi et al., 2015), which frequently positioned next to lysosomes.

3. CapZ is an essential protein in almost all actin dependent mechanisms, and as a consequence the KO of CapZ will undoubtedly affects a number of different actin-dependent processes that influence several conclusions of the paper. For example, The KO of CapZ is known to influence cell morphology (Sinnar et al., MBOC 2014). This important, because endosome number and dynamics depend directly on cell morphology.

a. Please provide determine if cell spreading and/or cell size was affected by CapZ KO in Hela cells.

b. Several endosomes metrics should also be weighted by cell size, such as the number of endosomes. This could markedly change the significance of several findings.

c. An increase in stress fibers in supplementary Figure S1A does not necessarily imply an increase in F-actin. Please quantify the G-actin to F-actin ratio in WT and CapZ-KO cells.

4. ArpC1B-mCherry does not appear localize to the lamella in Figure 2B and suggests that ectopic expression of ArpC1B-mCherry does not faithfully recapitulate the localization of the endogenous Arp2/3 complex. This is not surprising, given that the Arp2/3 complex is a 7-subunit complex, and only 1 subunit is being overexpressed, however, this does shed some doubt as to the relevance of these experiments. Thus, several control experiments are needed to bolster the significance of these findings.

a. As a positive control, the authors should show that ArpC1B-mCherry colocalizes with endogenous Arp2/3 complex.

b. Figure 2C shows N-Sim of ArpC1B-mCherry with endogenous Rab5. This analysis would be best served by comparing this colocalization with the colocalization of endogenous Arp2/3 complex and endogenous Rab5.

c. The representative images in 2B and 2C do not show the same focal plane for WT and KO cells and raises questions about the fairness of the colocalization analysis. Please show representative images of WT and CapZbKO cells at the same focal plane and indicate how the colocalization analysis was performed.

d. Also, how well does the overexpressed ArpC1B-mCherry colocalize with GFP-CapZ, and does CapZ KO disrupt AprC1B-mCherry localization?

e. Can the impaired motility of Rab5 positive vesicles be rescued by Arp2/3 complex inhibition and or actin depolymerization?

f. An alternative to the conclusion that inhibition of Arp2/3 complex leads to larger endosomes is that inhibition of Arp2/3 complex leads to impaired emergence of nascent endosomes from the plasma membrane. This would reduce the population of small vesicles and as a consequence would inherently overly inflate the contribution of larger vesicles to the overall mean vesicle area. Again, showing the distribution of endosome sizes would be helpful in distinguishing these possibilities.

g. localization of the Arp2/3 complex to these vesicular populations does not imply that it is active. Please determine, if this CapZ KO-driven localization is accompanied by enrichment of endogenous N-WASP on these enlarged vesicles.

5. A central assertion of the paper is that CapZ knockout might increase the formation of new actin branches to entangle around these immature RAB5-positive endocytic vesicles, and this might impede their fusion and result in the accumulation of small endosomes" (lines 113-115). However, there is surprisingly no imaging experiments of any kind performed with actin markers.

a. Please show that KO of CapZ does in fact lead to excessive actin polymerization on these endosomes.

b. Also, are the enlarged endosomes seen in the ArpC1B KO completely devoid of actin?

6. Rabaptin-5 interaction with Rab5 has been shown to be important for docking and fusion of early endosomes (Stenmark et al., Cell 1995). This raises the question whether impaired vesicle fusion observed in CapZ KO cells, is due to a loss of Rabaptin-5, from excessive branched actin nucleation, or a combination of the two? Based on the evidence presented in Figure 2E and 2F, it would seem that CapZ KO-driven loss of Rabaptin-5 is the main driving force behind the small Rab5-positive vesicles in CapZ KO cells since this effect could not be rescued with either CK-636 nor ArpC1B KO. However, endosome number/size should be quantified for CK-636/latrunculin-treated cells where Rabaptin-5 has been ablated.

7. The invitro binding of Rabaptin-5 to CapZ is unconvincing, mainly because of the low heat of interaction coupled to the high concentrations of proteins used in ITC experiment. To that end, the authors should be extra critical of the ITC results and provide a secondary measure of interaction (like a pulldown with purified proteins).

a. Given the high concentrations used in all the ITC experiments (400-500 mM!), please show the titration of the titrand into a cell containing buffer in the ITC figures as suggested to have been done in the methods. Frequently, the titration of highly concentrated proteins will give or take in heat due to dissociation of weak interactions, therefore the experimentally determined background should be used in the analysis, not just the end point.

b. The integrated heats of injection should be shown in the right panel. It's not clear if the "Heat changer per injection" (should be written as "Heat change per injection") represents the integrated heats of injection.

c. I'm concerned about the analysis that leads to the supposed integrated heats on the right. I'm assuming the first injection is omitted since the right panel shows the integration of 18 injections instead of all 19 shown in the left panel. If that is the case, the first two injection should not have the same integrated heats if they were integrated over the same time window. If the integrated heats follow the trend in the left figure, then the Kd would be considerably weaker than suggested. This would be consistent with the incredibly low heat of interaction (Assuming the reaction truly reaches saturation, then the DH of the reaction would be ~-0.6 kcal/mol, which is not even the energy found in a single isolated hydrogen bond!)

d. What is the rationale behind the high Kd but low DH? That would imply the reaction is entropically driven?

e. Please indicate and justify the binding model was assumed for the fit, and what is the experimental error for the Kd? If it was only performed 1 time, please report that is the case and report the fitting error.

8. The conclusion that inhibition of endosomal maturation leads to impaired ZIKV infection is not completely justified by the presented data. CapZ has also been implicated in autophagy, where ablation of CapZ leads to impaired autophagosome formation (Mi et al., 2015). Autophagy is also known to be accompanied by Cathepsin B cleavage and is impaired by Vacuolin-1 (as per the corresponding authors previous work), Chloroquine and Bafilomycin. Additionally, autophagy is well-established to promote ZIKV infection (as reviewed in Gratton et al., Int. J. Mol Sci 2019 and Chiramel & Best, Viruses 2018). Justifying the author's conclusion would requires distinguishing the contributions of CapZ in endosomal maturation and autophagy. For this person, I believe these results are best served for another publication where this conclusion can be thoroughly justified.

[Editors' note: further revisions were suggested prior to acceptance, as described below.]

Thank you for resubmitting your work entitled "Capping protein regulates endosomal trafficking by capping F-actin on endocytic vesicles and recruiting RAB5 effectors" for further consideration by *eLife*. Your revised article has been evaluated by the original reviewers and Anna Akhmanova (Senior Editor) and Mahak Sharma (Reviewing Editor). The reviewers have discussed their reviews with one another, and the Reviewing Editor has drafted this to help you prepare a revised submission.

The manuscript has been improved but there are few substantial issues that remain to be addressed, as outlined below:

1. On the F-actin around endosome after CAPZBeta KO: Both reviewers find that the new data does not provide clear evidence of actin association with endosomes as the global actin polymerization is altered in CAPZBeta KO, such as increased stress fibers. It is unclear whether individual endosome do display more branched actin around when CAPZbeta is KO or silenced. Direct imaging of actin on endosomes and further quantification could address these concerns.

2. One main point in the revision was the measurements of the rate of constitutive endocytosis in CapZβ-knock out cells. The authors provided some measurements of transferring recycling but not of the rate of endocytosis. The authors should quantify the amount of TfR or EGFR at the surface of cells shortly after the chase. CapZ knockdown was previously shown to impair the initial steps of endocytosis, however the authors explicitly say that CapZ affects steps after the formation of endocytosis.

3. The conclusions are based on the assumption that all the Rab5 positive structures are "endosomes". This is problematic, as not all Rab5 positive structures are endosomes and it's not evident (by methodology nor the presented experiments) that the authors have taken the appropriate steps to distinguish endosomes from other organelles. It is important to compare the size distributions of EGF or Transferrin endosomes in scramble and CapZ KO cells. The authors only measure the size of Rab5 positive structures and show that Rab5 colocalization with an early endosome marker is reduced in CapZ KO cells, suggesting that the smaller endosomes may not be early endosomes.

4. On EGFR: The total amount of EGFR is severely reduced after CAPZbeta KO, despite a significant reduction in its degradation. While studying the reason for this is probably beyond the scope of the present study, it would be easy and relevant to assess the total amount of mRNA of EGFR and whether reconstituting the expression of CAPZB restores the levels of EGFR. Also, in the EGFR degradation assay, it is surprising not to see the band corresponding to EGFR shifted as after stimulation with EGF, the receptors became heavily phosphorylated at multiple sites. Some clarification is needed here.

5. One explanation for the loss of localization Rab5 to EEA1 positive early endosomes in CapZ-KO cells is that there is increase in the rate of maturation of these endosomes. The authors should check if the colocalization of Rab5 with Rab7 affected by CapZ KO? Similarly, are the larger endosomes seen in the Arp2 KO cells, more positive for Rab7? This seems to be particularly true for the CK-636 treated cells in Supplemental Figure 4A, which have a dramatic Golgi-like localization.

6. The novel interaction between CapZ and the Rabaptin/Rabex GEF complex is the real strength of the paper. The authors have done an excellent job in validating these interactions, however the mechanism proposed of how CapZ regulates Rabaptin/Rabex GEF complex is confusing.

a) The authors propose that CapZ regulates Rab5 indirectly by regulating the localization of Rabaptin-5/Rabex-5, but it's unclear if CapZ solely effects localization and/or complex formation. For example, does CapZ facilitate the formation of a functional Rabaptin-Rabex complex? Does CapZ modulate binding of the Rabaptin-Rabex complex to Rab5?

b) Does Rabaptin-5 and Rabex-5 colocalizes with CapZDC34 in cells? Presumably, If the interaction between CapZ with said Rab5 GEFs is independent of F-actin, then CapZDC34 should localize equally well to the Rab5 GEF complex as full-length CapZ.

c) The increase in Rab5 colocalization with the Arp2/3 complex in CapZ KO cells is quite unexpected for two reasons. (1) it contradicts the assumption that CapZ recruits Rab5 (indirectly through Rabaptin-5/Rabex-5) to these endosomal actin patches. (2) it's indicated that Rabaptin-5 (Figure 5J) is delivered by CapZ (Figure 5H) and needed to facilitate the growth of these endosomes (presumably through fusion). However, without Rab5 GEFs it's impossible to promote the positive feedback loop needed to drive endosomal fusion and/or maturation. Can the authors provide some insight into why this is the case?

d) The impact of CapZ on early endosomes could also be explained by changes in protein expression of Rab5, Rabaptin-5, Rabex-5, VPS34, etc. Please provide evidence that the expression levels of these proteins have not changed with CapZ KO.

7. The authors need to include a discussion with how their observations fit into the wealth of data on what is known about CapZ in endocytosis and trafficking. The potential role of CapZ in endosome fusion versus fission should be discussed.

In addition to the points mentioned above, please address the comment #4 (a-f) and #5 of reviewer 3 on the colocalization analysis and methodology and provide the complete information in the manuscript text. Please also address the minor concerns listed by reviewer #3.

*Reviewer #2:*

The authors have performed a number of additional experiments to address each of the main and minor point raised. Overall, the revision of this manuscript is satisfactory. Few points merit further attention or discussion.

1. One main point requested was the measurements of the rate of constitutive endocytosis in CapZβ-knock out cells.

The authors provided some measurements of transferring recycling not of the rate of endocytosis. This should be clarified.

2. On the F-actin around endosome after CAPZBeta KO: The authors provided confocal images or N-SIM S Super-resolution imaging to show that more endosomes sit along the long F-actin in CapZ KO cells than the control cells. However, in the silenced cells there is an apparent increase in stress fibers that might confound the interpretation. In addition, while they documented the increase colocalization of endosomes with branched actin using ARC1B, it is unclear whether individual endosome do display more branched actin around when CAPZbeta is KO or silenced.

3. On EGFR: The total amount of EGFR is severely reduced after CAPZbeta KO, despite a significant reduction in its degradation. While studying the reason for this is probably beyond the scope of the present study, it would be easy and relevant to assess the total amount of mRNA of EGFR and whether reconstituting the expression of CAPZB restores the levels of EGFR. Also, in the EGFR degradation assay, it is surprising not to see the band corresponding to EGFR shifted as after stimulation with EGF, the receptors became heavily phosphorylated at multiple sites. Some clarification is needed here

*Reviewer #3:*

The revised manuscript, titled "Capping protein regulates endosomal trafficking by capping F-actin on endocytic vesicles and recruiting RAB5 effectors," attempts to establish a role for Capping protein (CapZ) and branched actin assembly in the fusion and maturation of early endosomes. In this revision, the authors have streamlined the paper by omitting the less-developed role of CapZ in ZIKV infection, and primarily focused on the role of CapZ in endosomal trafficking/maturation. To that end the authors have provided some additional experiments to show that CapZ influences trafficking of plasma membrane receptors and to show that interfering with branched actin assembly by the Arp2/3 complex leads to morphological changes in endosomes. Overall, these experiments help to build a case for the role of CapZ influencing the activity and/or localization of Rab5 to endosomes, however the conclusions drawn from these experiments are not well supported by the evidence and often contradict well documented roles of CapZ and actin in endocytosis. For example, while the receptor trafficking experiments are performed well they do not however provide additional support the authors conclusion that CapZ regulates endosomal fusion. In fact, the results of these experiments can be attributed to well-established functions of capping protein in the formation of endosomes. Similarly, the authors provided some additional experiments to interrogate the role of branched actin assembly in endosomal maturation, however these experiments are ultimately indirect, and the results can be interpreted that actin plays a role in endosomal fission. Surprising the authors do not capitalize on the most exciting aspect of the paper involving the interaction of CapZ with the Rabaptin-5/Rabex-5 GEF complex. This interaction could have been exploited in order to build a direct role of CapZ in regulating Rab5 function on endosomes. Ultimate, the manuscript needs a few more additional experiments to distinguish the role of CapZ and actin in endosome formation, fusion and maturation, reanalysis of their current data, and a critical assessment of their results with respect to the literature.

1. One of the major conclusions of this paper is "CapZ-KO increases the formation of new actin branches to entangle around these immature Rab5-positive endocytic vesicles." However, there are 2 major problems with this conclusion.

a. First, this is at odds with what is known about how CapZ regulates branched actin assembly. It's generally accepted that capping protein promotes actin branching by antagonizing formin-dependent actin elongation (Akin & Mullins, Cell 2008; Hu & Papoian, Biophys. J. 2010; Bombardier et al., Sinnar et al., MBoC 2014; Nat. Commun 2015; Shekhar et al., Nat Commun. 2015, Billault-Chaumartin & Martin Curr. Biol 2019). Furthermore, the authors present evidence that support the accepted role of capping protein, such as an increase cellular stress fibers (known to be assembled by Formins) with CapZ KO that are conversely impaired when CapZ is overexpressed. Similarly, the reduced motility of Rab5-positive vesicles beneath the plasma membrane is consistent with reduced branched actin nucleation (again please refer to Akin & Mullins, Cell 2008). Ultimately, the authors do not present any clear evidence that overturns what has been established about the role of CapZ in branched actin nucleation, and therefore these concluding statements should be modified to reflect what is known in the literature.

b. The proposed mechanism that CapZ directly regulates branched actin caging of endosomes is largely based on indirect evidence. The authors primarily show that CapZ KO has a global effect on cellular actin polymerization, including impaired cell spreading (rebuttal), impaired cell motility (rebuttal), increased stress fibers (Figure 3A), and loss of endosome motility (Figure 3E), and the connection between CapZ and branched actin assembly is built out of the coincidence that Arp2/3 complex inhibition leads to swollen Rab5 structures (Supplemental fig). However, this interpretation is problematic (see below). Instead, the authors should make some attempt to directly visualize actin on endosomes, such as using live cell imaging approaches or electron microscopy.

c. With regards to the above point, why have the authors ruled out the possibility that CapZ regulates endosome fusion by controlling Rab5 localization with EEA1? The authors demonstrate that there is less colocalization of Rab5 with EEA1 in CapZ KO cells (Figure 5E). EEA1 is a known be important for the docking and fusion of early endosomes (Christoforidis et al., Nature 1999; Mishra et al., PNAS 2010). Importantly, the tethering activity of EEA1 is regulated by nucleotide state of Rab5.

d. An alternative or additional explanation for the enlarged Rab5 structures resulting from inhibition of Arp2/3 complex (either through KO or CK-636 treatment), is that endosomal fission has been impaired (please refer to the review by Gautreau and colleagues, CSH Perspectives 2014). How do the authors distinguish between these two possibilities? This latter mechanism could also explain why CapZ-KO and VPS8 only partially rescues the Arp2-KO (Figure 4 C & D) and why CK-636 treatment had no effect on the interaction of CapZ with Rab5 and the Rab5 GEF complex (Supplemental Figure 6D&E).

e. With regards to the above point, It's difficult to make strong conclusions from the actin drug experiments. It's stated that cells were treated overnight with 1 and 100 μm latrunculin and CK-636, respectively. This is far longer than what is recommended for these inhibitors (Nolen et al., Nature 2009), and such extreme treatments can wreak havoc on a number of critical cellular processes like cell division (see discussion in Ilatovskaya et al., Cell Tissue Res 2013). What was the rational for extreme treatments? It would be more appropriate to apply these drugs in a more acute matter, like during the Tfn/EGF pulse chase experiment.

f. The double guide experiments with Arp2 and CapZ are perplexing. How does CapZ-KO promote branched actin cagin when Arp2 is knocked out or in the presence of CK-636?

g. In general, the authors need to include a discussion with how their observations fit into the wealth of data on what is known about CapZ in endocytosis and trafficking.

2. It is still not clear if CapZ regulates the formation of early endosomes, impairs endosomal fusion and/or promotes endosomal maturation. This is particularly important, because CapZ is well known to be important for the early steps of clathrin mediated endocytosis in yeast, where it aids in the formation and excision of the nascent endosome from the PM. (Kaksonen et al., Cell 2005; Kim et al., MBoC 2006; Berro & Pollard, MBoC 2014). It has also been shown to localize to the PM at the sites of CME in mammalian cells (Collins et al., Curr. Biol 2011). As such, additional experiments to clarify the situation.

a. Are all Rab5 positive vesicles in the CapZ-KO endosomes? Most cell images presented in the paper show strong localization to larger organelles, like Golgi and/or late endosomes, which can erroneously lead to inflated Mander's coefficient. How do the authors distinguish between "immature endosomes", fused early endosomes, mature endosomes or other organelles in their colocalization analysis? Given that this is a central aspect of the paper, the authors should restrict their analysis to early endosomes. One such approach would be assessing the colocalization analysis of both CapZ and Rab5 with an endosomal cargo, like wheat germ agglutinin (WGA) or EGF, as well as the colocalization of CapZ with Rab5 only on the WGA/EGF positive endosomes. Alternatively, the authors should restrict the colocalization analysis to regions expected to be occupied by early endosomes and not include the larger centrally located accumulations that cannot be reliably segmented.

b. To help support the above point, the authors should also compare the size distributions (numbers & area) of a bonafide early endosome marker, like EEA1.

c. Similarly, are these small Rab5 positive vesicles observed in cells where endocytosis has been inhibited such as by impairing dynamin function with Dynasore, in scramble and CapZ-KO cells? This would indicate that loss of CapZ promotes non-CME-dependent processes.

d. From the trafficking experiments presented in Figure 2 and Supplemental Figure 2, the authors conclude that "CapZ participates in endocytosis after the internalization of endocytic vesicles" (166-167). However, due to the long delay between time-points in these experiments, they can also be interpreted as a defect in internalization, which has been previously documented. To distinguish between these two possibilities, it would be necessary to capture the first 5 min of Tfn or EGF pulse chase experiment and quantify the loss of labelled Tfn/EGF from the plasma membrane (please refer to Daro et al., PNAS 1996).

e. Does CapZ regulate early endosomal fusion and/or endosomal maturation? One explanation for the loss of localization Rab5 to EEA1 positive early endosomes in CapZ-KO cells, is that there is increase in the rate of maturation of these endosomes. The authors should check if the colocalization of Rab5 with Rab7 affected by CapZ KO? Similarly, are the larger endosomes seen in the Arp2 KO cells, more positive for Rab7? This seems to be particularly true for the CK-636 treated cells in Supplemental Figure 4A, which have a dramatic Golgi-like localization.

f. I'm surprised that VPS8 KO had no effect on the basel size of endosomes

3. The novel interaction between CapZ and the Rabaptin/Rabex GEF complex is the real strength of the paper. The authors have done an excellent job in validating these interactions, but I'm surprised that the authors did not exploing this interaction to interrogate how CapZ regulates Rab5 on GEFs. As it stands, the presented mechanism is confusing.

a. Does CapZ regulate more than the localization of Rabaptin-Rabex complex? For example, does CapZ facilitate the formation of a functional Rabaptin-Rabex complex? Does CapZ modulate the activity Rabaptin-Rabex complex or binding of the complex to Rab5?

b. In support of the above point, does Rabaptin-5 and Rabex-5 colocalizes with CapZDC34 in cells? Presumably, If the interaction between CapZ with said Rab5 GEFs is independent of F-actin, then CapZDC34 should localize equally well to the Rab5 GEF complex as full-length CapZ.

c. The dependence of Rab5 localization to endosomes on the actin capping activity of CapZ is not clear. It's shown that removing the C-terminal 34 residues of CapZ impairs the amount of Rab5 that pulls down but does not affect it's ability to interact with Rabex-5, Rabaptin-5 and VPS34 (Figure 6F & G). However, it's also stated that Rab5 indirectly interacts with CapZ through the aforementioned Rab5 GEFs (lines 314 to 316). If this assertion was true, then the amount of Rab5 that pulled down with CapZ truncation should be similar to that of the full length CapZ. Please reconcile this discrepancy.

d. The increase in Rab5 colocalization with the Arp2/3 complex in CapZ KO cells is quite unexpected for two reasons. (1) it contradicts the assumption that CapZ recruits Rab5 (indirectly through Rabaptin-5/Rabex-5) to these endosomal actin patches. (2) it's indicated that Rabaptin-5 (Figure 5J) is delivered by CapZ (Figure 5H) and needed to facilitate the growth of these endosomes (presumably through fusion). However, without Rab5 GEFs it's impossible to promote the positive feedback loop needed to drive endosomal fusion and/or maturation. Can the authors provide some insight into why this is the case?

e. The impact of CapZ on early endosomes could also be explained by changes in protein expression of Rab5, Rabaptin-5, Rabex-5, VPS34, etc… Please provide evidence that the expression levels of these proteins has not changed with CapZ KO.

4. Much of the conclusions of the MS rely on colocalization analysis, which is surprisingly not at all described. This is particularly relevant to the above points. Please provide a detailed description as to how colocalization analysis was performed.

a. Indicate which software package was employed (presumably Image J).

b. Indicate whether the analysis was performed on single confocal plane, on Z-stack projection or in 3D? It's indicated in the rebuttal that "only single confocal plane was taken and analyzed unless otherwise stated." If this is true, indicate the criteria that was used to select this plane for analysis. Also, the methods indicate that the SIM images were reconstructed in 3D – were these image sets also analyzed in the same way as the confocal images. It's not stated how these data sets were analyzed.

c. How were the ROIs determined? Was the entire image plane or individual cells analyzed? If on the entire image plane, how many cells were in the field of view. If performed in 3D was the ROI the same in all image planes of the stack, or was segmentation performed on each pane to identify the true boundary of the cell?

d. Did the ROI exclude centrally located organelles, such as the nucleus, perinuclear region, and the Golgi? The use of Mander's to examine co-occurrence of endosomal markers is appropriate so long as it is restricted to peripheral organelles (please refer to Pike et al., Methods 2017).

e. Was the background subtracted? Which approach was used?

f. Was signal isolation performed in an automated and unbiased way, such as employing the Costes joint histogram approach?

5. The endosome size data was not statistically analyzed correctly. The data is not normally distributed and showing the mean +/- SD is not appropriate for these figures. Instead, please show a box and whisker plot on top of the data points. Also note, because the data is skewed, the appropriate statistical test is the Mann-Whitney U Test. It may be more beneficial to show this data as a frequency histogram distribution.

[Editors' note: further revisions were suggested prior to acceptance, as described below.]

Thank you for resubmitting your work entitled "Capping protein regulates endosomal trafficking by capping F-actin on endocytic vesicles and recruiting RAB5 effectors" for further consideration by *eLife*. Your revised article has been evaluated by Anna Akhmanova (Senior Editor) and a Reviewing Editor.

The manuscript has been improved but there are some remaining issues that primarily require text changes, data quantification and figure modifications. These changes are outlined below in the reviewers comments:

*Reviewer #2:*

The authors have done an additional set of experiments to address the concerns expressed. Overall, the manuscript is certainly improved and at this stage, only few points to clarify might be needed.

1. On the actin around the endosome. The new set of images in figure 3C is certainly a progress in this direction. Yet, it remains conceivable that the apparent increased chance of seeing an endosome adjacent to filamentous actin is simply a reflection of more cellular F-actin actin. Maybe one way to visualized actin, keeping in mind the effect exerted by CAPZ KO (which reduced the size and increase the number of endosomes) could have been treating cells with the Pik5 inhibitor. This said, at this stage, it would seem acceptable to use cautionary words on the increased F-Actin around the endosome following CAPZ genetic silencing w/o performing new experiments.

2. On the possible impact of CAPZ on early internalization steps. The attempts made were clearly inconclusive on the images in response to this point, it is stated that no permeabilization was done to ensure that the detected EGF would be localized at the cell surface. Specifically: "we found that staining fixed cells without permeabilization with phalloidin-647 could demonstrate plasma membrane" yet at 5 minutes there is apparently plenty of EGF signal inside the cells, which should not be detectable in the absence of permeabilization. The experiments on this issue seem inconclusive. There are more precise ways to measure the rate of endocytosis through the use of Iodinated EGF. The manuscript, however, is not focused on early steps on endocytosis, and it would seem not appropriate, at this point, to ask for these experiments. The authors were careful in commenting on the possible role of CAPZ in the early step of internalization and this might be all that is needed.

*Reviewer #3:*

The additional data provided in the rebuttal when fully included, markedly helps with interpreting the roll of CapZ in regulating early endosome fusion and/or maturation. There is a clear effect of CapZ on trafficking and it seems to be driven by the assembly on Rab-5 GEF complex, Rabex-5/Rabaptin-5. The data on this point is excellent and of general interests. However, my main concern is with this insistence to say that CapZ regulates branched actin assembly on endosomes, when it's unnecessary to do so. It also calls attention to the weak point of the paper, and this conclusion is formulated by an over-interpretation, and concerning implementation and colocalization analyses, not to mention the incorrect assumption that CapZ blocks Arp2/3 complex. Indeed, recent data provided in this recent revision suggests that Arp2/3 complex regulates maturation of early endosomes, since Arp2/3 KO leads to accumulation of Rab7 positive endosomes, that were previously misinterpreted as enlarged Rab5 endosomes. This calls into question the fairness of comparing the effects of CapZ KO with Arp2/3 complex KO on endosome number and endosome size in all of Figure 4 and Figure 5J. The correct thing to do would have been restrict the endosome size distribution analysis to early endosomes (as suggested by previously by the reviewers), which would necessarily exclude any Rab5 puncta overlapping with Rab7. Alternatively, the data can be kept as is, but the authors have to strongly state that these Rab5 positive structures also includes both early and late endosomes. Furthermore, it is quite likely, and correctly mentioned by the authors, that the Arp2/3 complex plays a role in a myriad of endosome functions that affects everything from cargo sorting, fission and fusion. As such, this data cannot be used to solely conclude that CapZ KO leads to an increase in branched actin density on early endosomes that impairs their fusion.

With regards to point 8.b. of the rebuttal letter, the authors state that they performed colocalization analysis on a single plane for the entire field of view, and each field of view consists of 5-10 cells. Is that correct? This would suggest extra-cellular debris and labeling would have been included in the analysis. Surely the authors did something to correct for this.

---

## [Author Response]

Essential revisions:The manuscript by Wang et al., describes the role of actin capping protein CapZ in early endosome maturation and cargo trafficking. CapZβ knock out HeLa cells show defect in endosome maturation and a delay in EGFR degradation. The early endosome maturation defects in CapZβ-knockout cells are partially rescued by inhibiting actin polymerization. CAPZ is also shown to mediate a direct interaction with basic components of the RAB5-endosomal fusion machinery and to control through them RAB5A activation and endosome fusion, with an impact on Zika virus infection. Overall, the experiments are well conducted and documented and support the mode of action of CapZ proposed on endosome maturation. However, there are several concerns that must be adequately addressed by additional experimentation to support the major conclusions of this study:Please find the points covering the essential revision requirements below:1) CapZ is an essential protein in almost all actin dependent mechanisms, and as a consequence the KO of CapZ will undoubtedly affects a number of different actin-dependent processes that influence several conclusions of the paper. For example, The KO of CapZ is known to influence cell morphology (Sinnar et al., MBOC 2014). Thus, authors should investigate parameters such as cell spreading and/or cell size in CapZ KO Hela cells. Several endosomes metrics should also be weighted by cell size, such as the number of endosomes. This could markedly change the significance of several findings. Further the rate of constitutive endocytosis should be measured in CapZβ-knockout cells.

Similar to the previous study by Sinnar et al., CapZ KO in HeLa cells changed cell morphology and inhibited cell migration, as shown in Figure 2—figure supplement 1A,B. CapZ KO also increased cell size as analyzed by measuring the size of the cells in Figure A via image J (Figure 2—figure supplement 1A), which is likely due to the increased actin polymerization. When the endosome parameters (size and number) are weighed by cell size, the differences of size and number of endosomes between control and KO cells are increased and decreased, respectively, yet still significant (Author response image 1) . Of note, it might be more appropriate to weigh these endosome parameters, especially number, by cell surface area, not cell size. Although CapZ KO cells appeared to be larger than the control cells, CapZ KO also decreased filopodial length (see the white lines in Author response image 1, and Figures 3F and 3G shown in Sinnar et al., 2014) (Sinnar et al., 2014). The reduced filopodial length in CapZ KO cells could mitigate, if not counteract, the changes of cell surface area induced by the increased cell size when compared to the control cells. Moreover, the causal relationship between cell size and endocytosis has not been established, although intuitively, large cells might have more endosomes than small cells. Thus, it might be more appropriate to compare endosome parameters per cell without weighing the cell size to simplify the interpretation of the results.

**Author response image 1. sa2fig1:** (A) Control or CapZβ-knockout HeLa cells were immunostained with an anti-RAB5A antibody. The number and size of the early endosomes in these cells were quantified and weighted by the cell size. (B) Control or CapZβ-knockout HeLa cells were immunostained with Red-X Phalloidin; the white dashed lines indicate filopodia.

As suggested, we measured transferrin recycling (one type of constitutive endocytosis), in control and CapZ KO cells. In transferrin recycling assay, iron-bound transferrin interacts with its receptor (TfR) to trigger the internalization of the iron-transferrin-TfR complex via clathrin-mediated endocytosis. Irons are subsequently dissociated from their ligands in the acidic endosomes and released into the cytoplasm. Thereafter, the transferrin-TfR complex is recycled back to the plasm membrane, and transferrin is finally released from its receptor into the extracellular space. Thus, the fluorescence intensity of fluorescent transferrin conjugates labeled cells could reflect the recycling endocytic trafficking process. As shown in Figure 2—figure supplement 1H, CapZ KO significantly inhibited transferrin recycling.

2) A central assertion of the paper is that CapZ knockout might increase the formation of new actin branches to entangle around these immature RAB5-positive endocytic vesicles, and this might impede their fusion and result in the accumulation of small endosomes" (lines 113-115). However, there is surprisingly no imaging experiments performed with actin markers.a) High resolution imaging of actin and Rab5 endosomes should provide better estimation of excessive actin polymerization on these endosomes. Subcellular fractionation experiments can also be performed to test actin levels in the endosomal fractions under these conditions.b) Are the enlarged endosomes seen in the ArpC1B KO completely devoid of actin?c) Can the impaired motility of Rab5 positive vesicles be rescued by Arp2/3 complex inhibition and or actin depolymerization?

a. As suggested, control or CapZ KO cells were stained with Phalloidin and RAB5, and then subjected to confocal images or N-SIM S Super-resolution imaging. As shown in Figure 3—figure supplement 1G,H, it is clear that more endosomes sit along the long F-actin in CapZ KO cells than the control cells. Of note, the intensity of short actin branches is much dimmer than long microfilaments (e.g., stress fibers) in Phalloidin-staining cells. Therefore, we transfected cells with GFP-RAB5A and mCherry-ARPC1B (a subunit of the Arp2/3 complex) to label early endosome and actin branches, respectively. We found that more GFPRAB5 puncta were colocalized with RFP-ARPC1B puncta in CapZβ-knockout cells when compared to the control cells (old Figure 2B in the original manuscript). We also performed RAB5A immunostaining in mCherry-ARPC1B-expressing HeLa cells followed by N-SIM S super-resolution imaging. We showed that more RAB5 puncta were associated with mCherry-ARPC1B puncta in CapZβ-knockout cells when compared to the control cells (old Figure 2C in the original manuscript). In addition, as suggested by the reviewer (point 6b, below), we assessed the colocalization of endogenous RAB5 with endogenous ARP2/3 by performing the anti-p16-Arc and anti-RAB5 immunostaining, and similar results were found (see Figures shown in point 6). These results are now presented as new Figures 3C and 3D in the revised manuscript.

We have also tried the subcellular fractionation experiment as suggested. We found that this experiment is technically challenging as we failed to separate the early endosomes from late endosomes, and actin is very abundant and spread all across all the fractions (Author response image 2) .

**Author response image 2. sa2fig2:** The lysates of control or CapZβ-knockout HeLa cells were subjected to sucrose gradient fractions, followed by respective immunoblot analysis.

b. As suggested, we performed RAB5 and actin immunostaining in control or ARP2 KO cells. We showed that colocalization between RAB5 and actin in ARP2 KO cells is markedly weaker than in control cells (Figure 4—figure supplement 1B). In addition, we performed RAB5 and ARPC1B (a subunit of the Arp2/3 complex, to label actin branches) immunostaining in control or ARP2 KO cells. We showed that colocalization between endogenous RAB5 and endogenous ARPC1B in ARP2 KO cells is markedly weaker than in control cells (Figure 4B).

c. As suggested, we assessed the effects of ARP2 KO on *the* motility of Rab5 positive vesicles in both control and CAPZ KO cells. We showed that ARP2 KO in control cells inhibited the motility of early endosomes, but failed to further inhibit the motility of early endosomes in CAPZ KO cells (Author response image 3) . Of note, a classic study by Taunton et al. has reported that once the endocytic vesicles invaginate and pinch off from the membrane, a burst (fast and short-lived) of F-actin tail catalyzed by the Arp2/3 complex drives these vesicles away from the plasma membrane and into the cytoplasm (Taunton et al., 2000). Thus, it is not surprising that ARP2 KO inhibits early endosome motility. We speculate that F-actin around early endosomes fine-tunes the motility of these vesicles, and too much or too little F-actin compromises its motility. The videos for these experiments are uploaded for reference as well (Author response videos 1-4). Also, the study by Taunton et al. is cited in the revised manuscript.

**Author response image 3. sa2fig3:** Control, CapZβ-knockout, ARP2-knockout, or CapZβ/ARP2 double-knockout HeLa cells were transiently transfected with RFP-RAB5A, and cells were then imaged without delay for 3 min by time-lapse Nikon high-speed confocal microscopy. 2D track analysis was performed using NIS-Elements AR software.

**Author response video 1. sa2video1:** RAB5-positive endosomes movements in wild-type HeLa cells.

**Author response video 2. sa2video2:** RAB5-positive endosomes movements in CapZb-knockout HeLa cells.

**Author response video 3. sa2video3:** RAB5-positive endosomes movements in ARP2-knockout HeLa cells.

**Author response video 4. sa2video4:** RAB5-positive endosomes movements in CapZb-ARP2 double knockout HeLa cells.

3) It is unclear whether there is any preference for CAPZ binding to early (RAB5 positive) versus Late Rab7- or RAB4/RAB11-recycling endosomes. The Golgi-like localization of GFP-CapZ suggests it localizes to lysosomes or late endosomes. To that end, please quantify the degree of colocalization of CapZ with Lamp1 (images are in the supplementary Figure 1B but not quantified), Rab7 and Rab11/Rab4. Rab4 is particularly important since it can directly interaction with Rabaptin-5 (Vitale et al., EMBO J. 1998).

As suggested, we assessed the colocalization of CapZ (CapZ_β_-GFP) with RAB7 (RAB7mcherry), RAB11 (anti-RAB11 immunostaining), LAMP1 (anti-LAMP1 immunostaining), RAB4-Flag (anti-Flag staining), or RAB5 (anti-RAB5 staining). As shown in Figure 1—figure supplement 1D and 1E , CapZ showed strong colocalization with RAB5, and weak colocalization with RAB7, RAB11, RAB4, and LAMP1. Similarly, by co-immunoprecipitation (coIP) assay suggested by the reviewer (point 4 below), we detected the association of CapZ with RAB5, RAB7, and RAB11, but not LAMP1 and RAB4. In summary, these results suggest that CapZ might preferentially associate with early endosomes. Of note, whether CapZ is involved in RAB11/4-mediated recycling endosomes and/or RAB7-mediated late endosomes is currently under investigation in the lab. If successful, these results will be submitted for future publication.

4) Similarly, while the co-immunoprecipitation between CAPZ and RAB5 is documented, it is unclear whether this interaction is specific (do other RABs, and specifically RAB7 or RAB11, interact with CAPZ?) and whether there is any preferential binding to the GTP or GDP bound form of RAB5. One key tenet, here, is that CapZ interaction with RAB5 and endosome should be mediated by the branched actin network around these structures. Does inhibiting actin polymerization either with Latrunculin or CK-639 impair the interaction between CapZ and RAB5 but not some of the component of endosomal fusion machineries reported in Figure 3 (as later shown using CapZ mutants)? Further interaction between Rab5 GEF and Rab5 also seems to be dependent on CapZ actin capping activity. The potential reason/ basis of these observations can be added to the Discussion section of the manuscript.

As mentioned above, by co-IP assay, we detected the association of CapZ with RAB5, RAB7, and RAB11, but not LAMP1 and RAB4 (Figure 1—figure supplement 1F-1I). Again, as mentioned above, more efforts are currently going on in the lab to assess whether CapZ is involved in RAB11/4-mediated recycling endosomes and/or RAB7-mediated late endosomes.

To determine whether there is any preferential binding of CapZ to the GTP or GDP bound form of RAB5, we incubated the GST or GST-RAB5A recombinant proteins in the presence of GDP_β_S or GTP_γ_S with lysates of Twinstrep-_CapZβ_-expressing HEK293 cells, followed by GST pulldown and strep immunoblotting. It appears that recombinant GDP-bound GSTRAB5 protein was able to pull down more CapZ from HeLa cell extracts than the GTP-bound RAB5 (Figure 1F). This result is discussed in the second paragraph on page 19 of the revised manuscript.

As suggested, we assessed whether CK-636 (an inhibitor of the Arp2/3 complex) impairs the interaction between CapZ and RAB5 or its effectors by co-IP experiment. We showed that CK-636 had no effects on CapZ’s interaction with actin, RAB5, and RAB5 effectors (Figure 6–—figure supplement 1D,E)**.** These results suggest that as long as CapZ is capable of capping the Factin, its interaction with RAB5 or its effectors is independent of the actin branch around the endosome. Consistently, CK-636 or ARP2 knockout resulted in enlarged early endosomes (new Figures 4A and 4C in the revised manuscript), confirming that inhibiting the branch actin formation does not affect the machinery for early endosome maturation (e.g., the recruitment of RAB5 and its effectors to early endosomes). On the other hand**,** we showed in the manuscript that CapZ_β_-_∆_C34 (which lacks the actin capping activity) lost its interaction with actin, and its association with RAB5 was significantly diminished. However, CapZ_β_-_∆_C34 retained its binding capabilities towards Rabex-5 and Rabaptin-5 (Figure 6F and 6G in the revised manuscript). In addition, when CapZ_β_ was knocked out, the interaction or association between Rabex-5 or Rabaptin-5 and RAB5 was reduced (Figure 5F-5I in the revised manuscript). In contrast, knockdown of either Rabex-5 and Rabaptin-5 did not affect the ability of CapZ to interact with Rabex-5 or Rabaptin-5, respectively (Figure 5—figure supplement 1F and 1G in the revised manuscript). These results suggest that when CapZ caps the F-actin (via its C-terminal tail), it might function as a scaffold protein to carry Rabaptin-5 and Rabex-5 (via its N-terminal domain) to RAB5-GDP-positive early endosomes, which are associated with F-actin. This could induce the production of RAB5GTP, and the active RAB5 then recruits more Rabaptin-5, Rabex-5, and other effectors to initiate a positive feedback loop to further increase its activity. We have presented the results of CK-636 on the interaction between CapZ and RAB5 or its effectors as new Figure 6—figure supplement 1D and 1E, and have discussed the role of actin capping activity of CapZ in its interaction with RAB5 and its effectors in the first paragraph on page 20 of the revised manuscript.

5) In Figure 1F, the trafficking dynamics of fluorescently labeled EGF is examined in CAPZ null versus reconstituted cells. CAPZ removal results in a persistent accumulation of EGF in intracellular vesicles, leading the author to conclude that EGFR degradation is severely delayed by CAPZ Knocked down. However, to support this contention, EGFR receptor levels (also through western blotting) should be monitored over time following a pulse of EGF. Indeed, the accumulation of EGF/EGFR could be also caused by inhibition of recycling begging the question as to whether CapZ is also involved EGF/EGFR recycling? Additionally, one might expect EGFR signalling to be perturbed given the tight relationship between endosomal fluxes, number and size and signalling (as shown by recent work by Zerial group that proposed the notion of endosomes act as quanta signalling platforms thereby influencing ERK1/2 activity).

As suggested, we examined the EGFR levels after EGF pulse in control or CapZ KO HeLa cells by EGFR immunoblotting. We showed that EGF treatment triggered EGFR degradation in control cells, yet its degradation in CapZ KO cells was inhibited. Of note, the basal EGFR level in CapZ KO cells was lower than in control cells (Figure 2—figure supplement 1E). Therefore, we repeated the EGFR degradation experiments in cells pretreated with cycloheximide (a protein synthesis inhibitor). Similar results were observed, and the basal EGFR level in CapZ KO cells was still lower than the control cells (Author response image 4A). Why CapZ KO affects EGFR levels remains to be determined. Yet, the expression of RAB5, Rebex-5, or Rabectin-5, was not affected by CapZ KO (see Figures 2E, 5H, and 5I in the revised manuscript). CapZ KO also did not affect the expression of RAB7 and VPS34 (Author response image 4C).

As suggested, we also examined the pERK activation in control or CapZ KO cells after EGF treatment. The pattern of EGF-induced ERK activation in control and CapZ KO cells is similar (Figure 2—figure supplement 1E and Author response image 4A). Since the basal pERK level was lower in CapZ KO cells than in control cells (likely due to the decreased EGFR levels in KO cells), we also normalized the p-ERKs in EGF-treated control or CapZ KO cells over respective non-treated cells. It appears that the extent of pERK activation by EGF in CapZ KO cells is greater than in control cells (Author response image 4B). However, the reduced EGFR level in CapZ KO cells might complicate the interpretation of the downstream ERK activation (because we treated both types of cells with the same concentration of EGF). Moreover, EGF/EGFRinduced ERK activation is wired by multiple positive feedback and negative feedback loops and is cell type-dependent, and the EGF-induced EGFR lysosomal degradation is just one of the negative feedback loops (Avraham and Yarden, 2011). Therefore, more efforts are needed to address whether ERK activation is affected by EGFR endosomal trafficking.

**Author response image 4. sa2fig4:** Control or CapZβ-knockout HeLa cells were treated with EGF (100 ng/ml) in the presence of cycloheximide (10 μg/ml) (A). They were then collected at the indicated time points and subjected to western blot analysis against EGFR, ERK, pERK, and Hsp70. The p-ERKs in EGF-treated control or CapZβ-knockout cells were also normalized over non-treated cells, respectively (B). (C) The expression levels of RAB5A, RAB7A, and VPS34 in control or CapZβ-knockout HeLa cells were determined by western blot analysis.

We agree that the accumulation of EGF/EGFR in early endosomes could also be caused by inhibition of recycling since CapZ KO also inhibited recycling endosomal trafficking of transferrin (as shown in Figure 2—figure supplement 1H)*.* In our assay to monitor the endosomal trafficking of EGFR, we applied EGF-488 to label the cell surface EGFR. In this assay, we only observed the delivery of the EGFR-EGF-488 complex to lysosomes, but not its recycling back to the plasma membrane (top panel in Figure 2B in the revised manuscript). It is possible that EGFR might disassociate from EGF-488 in endosomes before some EGFR is routed to recycling endosomes, thus escaping our observation. However, we could not find a suitable anti-EGFR antibody to label the cell surface EGFR to monitor its recycling after EGF treatment. Also, as mentioned above, our lab currently investigates the role and mechanism of CapZ in recycling endocytosis.

6) ArpC1B-mCherry does not appear localize to the lamella in Figure 2B and suggests that ectopic expression of ArpC1B-mCherry does not faithfully recapitulate the localization of the endogenous Arp2/3 complex. a. As a positive control, the authors should show that ArpC1B-mCherry colocalizes with endogenous Arp2/3 complex.

a. As suggested, we stained ARPC1B-mCherry- or ARPC1B-GFP-expressing HeLa cells with anti-p16-Arc antibody. It is clear that most ARPC1B puncta colocalize with endogenous Arp2/3 complex (Author response image 5). Also, as suggested by the reviewer in point b, we have used the endogenous Arp2/3 localization to replace the ARPC1B-mCherry images in the revised manuscript.

**Author response image 5. sa2fig5:** The ARPC1B-mCherry- or ARPC1B-GFP-expressing HeLa cells were stained with anti-p16-Arc antibody, followed by confocal imaging.

b. Figure 2C shows N-Sim of ArpC1B-mCherry with endogenous Rab5. This analysis would be best served by comparing this colocalization with the colocalization of endogenous Arp2/3 complex and endogenous Rab5.

b. As suggested, we performed N-Sim imaging of endogenous Arp2/3 complex (p16-Arc staining) and endogenous RAB5, and reconstituted the 3-D images. We showed that more RAB5 puncta were associated with Arp2/3 puncta in CapZβ-knockout cells when compared to the control cells (Figure 3E). These results are presented in the revised manuscript in Figure 3-Video 1 and 2.

c. The representative images in 2B and 2C do not show the same focal plane for WT and KO cells and raises questions about the fairness of the colocalization analysis. Please show representative images of WT and CapZbKO cells at the same focal plane and indicate how the colocalization analysis was performed.

c. As suggested, we repeated the similar confocal imaging analysis (Manders’ colocalization coefficient) of colocalization between endogenous RAB5 and endogenous Arp2/3 in control and CapZ KO cells. Similarly, more RAB5 puncta were colocalized with Arp2/3 puncta in CapZ_β_-knockout cells when compared to the control cells (Figure 3D). When responding to reviewer 3’s comments below, we have addressed the rationale for using Pearson’s correlation coefficient or Manders’ colocalization coefficient (point *iii*).

d. Also, how well does the overexpressed ArpC1B-mCherry colocalize with GFP-CapZ, and does CapZ KO disrupt ArpC1B-mCherry localization?

d. We also assessed the colocalization between ARPC1B-mCherry and CapZ-GFP. As shown in Author response image 6, overexpressed ARPC1B strongly associated with CapZ. Compared to control cells, CapZ KO seemed to markedly increase the ARPC1B-mCherry puncta.

**Author response image 6. sa2fig6:** The control or CapZβ-knockout HeLa cells were transiently transfected with ARPC1B-mCherry or CapZβ-GFP, followed by confocal imaging.

e. An alternative to the conclusion that inhibition of Arp2/3 complex leads to larger endosomes is that inhibition of Arp2/3 complex leads to impaired emergence of nascent endosomes from the plasma membrane. This would reduce the population of small vesicles and as a consequence would inherently overly inflate the contribution of larger vesicles to the overall mean vesicle area. Again, showing the distribution of endosome sizes would be helpful in distinguishing these possibilities.

e. We have repeated the experiments and analyzed the sizes and number of endosomes in control, CapZ KO, ARP2 KO, and CapZ/ARP2 double KO cells. The distribution of endosome sizes and numbers are presented as suggested by the reviewer (Figure 4C). It is clear that ARP2 KO did significantly increase the number of large vesicles when compared to the control cells, although it might also decrease the number of small vesicles.

f. Localization of the Arp2/3 complex to these vesicular populations does not imply that it is active. Please determine, if this CapZ KO-driven localization is accompanied by enrichment of endogenous N-WASP on these enlarged vesicles.

f. As suggested, we assessed the colocalization of RAB5 with N-WASP in control or CapZ KO cells. We showed that CapZ KO markedly increased the Colocalization of N-WASP with RAB5 Author response image 7. Of note, CapZ KO induced the smaller, not larger, vesicles when compared to the control cells.

**Author response image 7. sa2fig7:** The control or CapZβ-knockout HeLa cells were immunostained with antibodies against RAB5 and N-WASP, followed by confocal imaging.

7) The in vitro binding of Rabaptin-5 to CapZ is unconvincing, mainly because of the low heat of interaction coupled to the high concentrations of proteins used in ITC experiment. To that end, the authors should be extra critical of the ITC results and provide a secondary measure of interaction (like a pulldown with purified proteins). Based on the evidence presented in Figure 2E and 2F, it would seem that CapZ KO-driven loss of Rabaptin-5 is the main driving force behind the small Rab5-positive vesicles in CapZ KO cells since this effect could not be rescued with either CK-636 nor ArpC1B KO. However, endosome number/size should be quantified for CK-636/latrunculin-treated cells where Rabaptin-5 has been ablated.

Since GST alone was able to pulldown CapZ in vitro, we have removed the GST-tag from Rabaptin-5, and performed the Biolayer interferometry (BLI) assay to assess the interaction between Rabaptin-5 and CapZ. As shown in Figure 5D, Rabaptin-5 binds to immobilized CapZ (200 _µ_g/ml) in a dose-dependent manner. This BLI result is presented as the new Figure 5D, and the ITC data is presented as Figure 5—figure supplement 1D in the revised manuscript.

As suggested, we assessed Rabaptin-5 KO on the size and number of endosomes in control or ARP2 KO cells. As expected, Rabaptin-5 KO increased the number, and decreased the size of early endosomes in both control and ARP2 KO cells (Figure 5J).

8) The authors examine the potential pathophysiological consequences of the loss of CAPZ using Zika Virus infection. However, this part of the manuscript is not thoroughly investigated. For instance, CapZ has also been implicated in autophagy, where ablation of CapZ leads to impaired autophagosome formation (Mi et al., 2015). Additionally, autophagy is well-established to promote ZIKV infection (as reviewed in Gratton et al., Int. J. Mol Sci 2019 and Chiramel & Best, Viruses 2018). Justifying the author's conclusion would requires distinguishing the contributions of CapZ in endosomal maturation and autophagy. Thus, these results are best served for another publication where this conclusion can be thoroughly justified. Instead, examining the physiological signalling and trafficking biology of an endocytic cargo such as EGFR receptor might provide more insights onto the impact of CapZ on endosome maturation.

One of the research focuses in our lab is to systematically dissect the role of canonical autophagy machinery in flavivirus infection, including ZIKV. The pro- or/and anti-viral role of autophagy is rather complicated and is highly context-dependent. Yet, we agree with the reviewer that a more thorough study is needed to investigate the role of CapZ in ZIKV infection. Thus, we removed Figures 5 and S5 from the manuscript. The effects of CapZ KO on endosomal trafficking of EGFR and transferrin receptor are presented as Figures 2B and Figure 2—figure supplement 1A-1C in the revised manuscript.

Reviewer #3:The proposed manuscript, titled "Capping protein regulates endosomal trafficking by capping F-actin on endocytic vesicles and recruiting RAB5 effectors," attempts to uncover a potential role for the barbed-end actin capping protein, CapZ, in the fusion and maturation of early endosomes. To that end the authors show that ablation of CapZ leads to the accumulation of small Rab5-positive vesicle that are believed to represent unfused early endosomes. In an attempt to establish the underling mechanism, the authors find that loss of CapZ leads to enhanced recruitment of Arp2/3 complex to Rab5 positive vesicles, which is assumed to be driving unregulated actin assembly on these endosome, that ultimately impairs their fusion. The authors also find that CapZ may directly interact with the Rab5 effector, Rabaptin-5, which in itself could potentiate the recruitment of Rab5. Finally, they also demonstrate that impairing endosome maturation through CapZ ablation can impair infection of ZIKV RNA.The authors use a wide array of experimental approaches to put forward a number of interesting observations that could have considerable impact on understanding how the actin dynamics is coupled to endosomal maturation. However, based on the current data, it is unclear if the loss of CapZ leads to impaired fusion of nascent endosomes with the early endosomes because of excessive actin assembly on the early endosomes or it it is due to a loss of Rab5 activity and/or Rab5 dependent vesicle fusion factors, like Rabaptin-5. Also, it's unclear if the dependence of Rab5 localization to early endosomes is entirely influenced by a local populations of CapZ on endosomes emerging from the plasma membrane, or if it's a result of global changes in the actin dynamics and membrane trafficking in response to knocking-out a crucial element of the actin cytoskeleton. This is particularly important when considering viral infectivity, which relies on the convergence of multiple actin cytoskeleton dependent processes.1. Data presentation though-out the figures need to be improved and better explained. As it stands, it is not evident that anyone could reproduce the data presented in the paper.

As suggested, we have reorganized the figures (original Figure 1 is split into Figures 1 and 2; Figure 2 into Figures 3 and 4; Figure 3 into Figure 5; Figure 4 into Figure 6; Figure S1 into Figure 1—figure supplement 1 and Figure 2—figure supplement 1; Figure S2 into Figure 3figure supplement 3 and Figure 4—figure supplement 1; Figure S3 into Figure 5—figure supplement 1; and Figure S4 into Figure 6—figure supplement 1); expanded the figure legends, and added more details in the material and method sections.

a. Error bars need description. I'm assuming they are presented as SEM, however this could be entirely inappropriate if the data is not normally distributed.

As suggested, the error bars have all changed to SD in the revised manuscript.

b. Please report the individual measurements in addition to the bar graphs so the reader can accurately assess and confirm the appropriateness of the statistical treatments. Also, presenting the endosome number and size distributions would be considerably more informative than the current mean and SEM presentation.

As suggested, all bar graphs are replaced with dot blots to reflect the individual measurements, and presented as mean ± SD.

c. For imaging analysis:i. Please indicate, for all experiments, whether conventional or SIM-mode was used.

All fluorescent images were captured by conventional confocal microscope if not otherwise specially stated (the N-SIM images are clearly indicated in the respective images).

ii. Please describe and justify the nature of the images analyzed (single-plane, all-planes, max projection, etc..) and indicate if only selected regions were analyzed within in those images.

All the confocal images were taken and analyzed in a single plane if not otherwise specially stated.

iii. There are several instances where Mander's overlap coefficient is reported, but not Pearson's, and it's not clear why. This is particularly worrisome, because Mander's is heavily weighted by pixel intensity and small endosomes will have considerably less Rab5 intensity compared to large endosomes. In almost all cases, Pearson's correlation coefficient should be used. Importantly, please indicate on the Y-axis of each correlation bar graph the type of correlation.

Manders colocalization coefficient (MCC) or Pearson's correlation coefficient (PCC) are two major metrics of colocalization analysis of two probes. Both have pros and cons, and neither is superior to the other. We followed MCC or PCC methods based on the workflow described by Dunn et al. (Dunn et al., 2011). Briefly, to analyze the colocalization between RAB5 and its effectors (e.g., EEA1, Rabex-5, or Rabaptin-5), PCC was used because RAB5 and its effectors mostly co-occur on the same cellular structures (early endosomes), and their fluorescent signals fit a single linear relationship, which satisfies the requirements of PCC. On the other hand, if images were required to set a threshold to distinguish an object from a background (e.g., CapZ), or only a small proportion of two objects (such as p16-Arc vs. RAB5, CapZ vs. RAB5, or RAB5 vs. Actin) were codistributed in the same compartment, MCC was used. We have clearly labeled the MCC or PCC metrics in respective blots.

iv. Please describe the criteria (min/max size, intensity cut-offs, circularity, etc..) used to automatically identify endosomes for the image analysis and how overlapping endosomes were resolved.

It has been estimated that the early endosomes range from 200-600 nm (in diameter) by confocal and electron microscopy (Shearer and Petersen, 2019; Vermeulen et al., 2018). To maximally count all RAB5-positive endocytic vesicles, we actually set the min/max size of endosomes to the same value in the same set of experiments (e.g., 30000 nm^2^(min)/10000000 nm^2^(max)). The intensity cutoffs were set automatically through otsu threshold methods with manual checks to ensure that most endosomes were appropriately circulated. The overlapping endosomes were separated by the watersheds method. This information is also added to the “Materials and Method’ section (the third paragraph) on page 30 of the revised manuscript.

v. Please indicate how single endosomes were tracked in Figure 2D. There is no method listed for this. It's not enough to say, "performed using NIS-Elements AR software". What steps were taken to ensure the same endosome was being tracked between frames? This is particularly important when considering there is an abundance of small endosomes in CapZ KO cells, which will be less bright compared to those in WT cells.

The endosomes in each frame were detected by the bright different sizes method in the NISElements AR software; the typical diameter and contrast parameters were set appropriately to ensure most objects were selected. The objects were then tracked with a constant speed motion model. The "linking" modules were selected, which allowed three maximum gap sizes in tracks, to help the same endosomes were tracked between frames.

vi. Was the Live-cell imaging performed in super-resolution mode? What was the 'real' acquisition speed of the experiment (this is missing in the caption and in the figure legend). Again, this is important for faithfully tracking endosomes.

The live-cell imaging was performed in a high-speed confocal microscope with a 100X objective lens. The interval between each frame is around 0.5 seconds.

vii. Please explain how "Average speed" and "Average moving distance" was calculated.

The average speed is the average speed of each endosome in each frame. The average moving distance is calculated as followed: the average moving distance of each endosome in each frame X (multiply) total frames (360 frames).

viii. What was the average time a single endosome could be tracked?

The average tracking time of a single endosome was around eight seconds.

ix. It's stated that endosomes from 14 cells were tracked in the figure caption, but it's not stated how many endosomes were tracked per cell.

For WT cells, more than 3000 endosomes were tracked in each cell in a 3 minutes tracking time; for CapZ knockout cells, more than 9000 endosomes were tracked in each cell in the same period. The information for points *v-ix* is also added to the revised manuscript on page 29 (second paragraph).

d. For ITC analysis:i. Please report binding model used for the fit, and ALL fit parameters, along with fit errors. Indicate which fit parameters were fixed.ii. Indicate if the experiment was performed only once or multiple times, If the latter, report the experimental average and associated error.

Titration curves were fitted by MicroCal PEAQ-ITC analysis software (Malvern Panalytical) using a one-site binding model. The fitting parameters are N = 0.372 ±4.7e-3, KD = 378e-9 ±62.1e-9 M, ΔH = 0.581 ±1.2e-2 kcal/mol, ΔG = -8.76 kcal/mol, and -TΔS = -8.18 kcal/mol, respectively. All fit parameters are included in new Figure 5—figure supplement 1D in the revised manuscript.

The ITC experiments for binding between CapZ and Rabaptin-5 were performed twice with similar results: one with 400 µM Rabaptin-5 to 40 µM CapZ complex, and another one with 200 µM CapZ complex to 20 µM GST-Rabaptin-5 (Figure 5—figure supplement 1D ). As suggested by the reviewer, we have confirmed the interaction between CapZ and Rabaptin-5 by Biolayer interferometry (BLI) assay, which is presented as new Figure 5—figure supplement 1D in the revised manuscript.

References

Avraham, R., and Yarden, Y. (2011). Feedback regulation of EGFR signalling: decision making by early and delayed loops. Nat Rev Mol Cell Biol *12*, 104-117.

Dunn, K. W., Kamocka, M. M., and McDonald, J. H. (2011). A practical guide to evaluating colocalization in biological microscopy. American Journal of Physiology-Cell Physiology *300*, C723-C742.

Kaksonen, M., Toret, C. P., and Drubin, D. G. (2005). A modular design for the clathrin- and actinmediated endocytosis machinery. Cell *123*, 305-320.

Shearer, L. J., and Petersen, N. O. (2019). Distribution and colocalization of endosome markers in cells. Heliyon *5*, e02375.

Sinnar, S. A., Antoku, S., Saffin, J. M., Cooper, J. A., and Halpain, S. (2014). Capping protein is essential for cell migration in vivo and for filopodial morphology and dynamics. Mol Biol Cell *25*, 2152-2160. Taunton, J., Rowning, B. A., Coughlin, M. L., Wu, M., Moon, R. T., Mitchison, T. J., and Larabell, C. A. (2000). Actin-dependent propulsion of endosomes and lysosomes by recruitment of N-WASP. J Cell Biol *148*, 519-530.

Vermeulen, L. M. P., Brans, T., Samal, S. K., Dubruel, P., Demeester, J., De Smedt, S. C., Remaut, K., and Braeckmans, K. (2018). Endosomal Size and Membrane Leakiness Influence Proton Sponge-Based Rupture of Endosomal Vesicles. ACS Nano *12*, 2332-2345.

[Editors' note: further revisions were suggested prior to acceptance, as described below.]

The manuscript has been improved but there are few substantial issues that remain to be addressed, as outlined below:1. On the F-actin around endosome after CAPZBeta KO: Both reviewers find that the new data does not provide clear evidence of actin association with endosomes as the global actin polymerization is altered in CAPZBeta KO, such as increased stress fibers. It is unclear whether individual endosome do display more branched actin around when CAPZbeta is KO or silenced. Direct imaging of actin on endosomes and further quantification could address these concerns.

As suggested, we performed double immunostaining with antibodies against Rab5 (green) and Actin (red) in control or CapZ_β_ KO cells. The single plane whole-cell images were then captured by Leica SP8 confocal microscope with a 63x oil lens (NA 1.45) and a resolution of 2048x2048 pixels (left panels of Figure 3C). To demonstrate the detail of the actin branches, we further scanned the selected area in the cells with a resolution of 4096x4096 pixels (right panels of Figure 3C). We showed that more Rab5-positive endosomes are buried inside the actin branches in CapZ_β_ KO cells than the control cells. We also quantified the percentage of Rab5 puncta that are actin positive against total Rab5 puncta in control or CapZ_β_ KO cells. We showed that more endosomes are actin-positive in CapZ_β_ KO cells when compared to the control cells. These figures are presented as new Figure 3C in the revised manuscript.

From the literature, various groups have also performed immunostaining with antibodies against the subunit of the Arp2/3 complex or expressed the GFP or RFP fusion proteins of these subunits to demonstrate short actin branches [1-5]. Therefore, we performed immunostaining with an antibody against either p16-Arc or ARPC1B (both are subunits of Arp2/3 complex) to demonstrate the actin branch around endosomes. As shown in Figures 3D, 3E, and Figure 3-Video 1 and 2 in the revised manuscript, the colocalization coefficient of RAB5 puncta (RAB5 immunostaining) with Arp2/3 puncta (p16-Arc immunostaining) in CapZ_β_-knockout cells was significantly higher than the control cells. In addition, in the original manuscript, we transfected cells with mCherry-ARPC1B to label actin branches. We showed that more GFP-RAB5 puncta were colocalized with RFP-ARPC1B puncta in CapZ_β_-knockout cells when compared to the control cells (old Figure 2B in the original manuscript). We also performed RAB5A immunostaining in mCherry-ARPC1B-expressing HeLa cells followed by N-SIM S super-resolution imaging. We showed that more RAB5 puncta were associated with mCherry-ARPC1B puncta in CapZ_β_-knockout cells when compared to the control cells (old Figure 2C in the original manuscript). Take together, these results suggest that more actin branches are associated with early endosomes in CapZ_β_ KO cells when compared to the control cells.

2. One main point in the revision was the measurements of the rate of constitutive endocytosis in CapZβ-knock out cells. The authors provided some measurements of transferring recycling but not of the rate of endocytosis. The authors should quantify the amount of TfR or EGFR at the surface of cells shortly after the chase. CapZ knockdown was previously shown to impair the initial steps of endocytosis, however the authors explicitly say that CapZ affects steps after the formation of endocytosis.

As suggested, we measured the cell surface EGFR levels 1 or 5 minutes after EGF chase. Briefly, live control or CapZ_β_-knockout HeLa cells were incubated with EGF-488 on ice for 30 min. After washed, cells were incubated with the warm medium at 37 °C incubator for 1 or 5 minutes before fixation and staining with phalloidin-647 and DAPI (we found that staining fixed cells without permeabilization with phalloidin-647 could demonstrate plasma membrane). As shown in Author response image 8, the cell surface EGFR or intracellular EGFR levels at 1 min or 5 min after EGF chase in CapZ_β_-knockout cells exhibited no significant differences when compared to control cells. Due to the technical limitation of our chase assay, we could not be able to monitor EGFR seconds after EGF chase. Therefore, TIRF microscopy might be more appropriate to examine the effects of CapZ_β_ KO on the rate of constitutive endocytosis. However, we do not have this microscope in our institute. We are also not able to use it at another institute in Shenzhen (a neighboring city in mainland China), where we could find help from colleagues with expertise, because the border between Hong Kong and Shenzhen has been closed for the past 20 months in response to the current pandemic. It is worth pointing out that Kaksonen et al. has shown that deletion of CapZ in yeast cells reduced the moving rate of endocytic vesicles from the membrane invagination sites. Their results indicate that actin meshwork around the membrane invagination facilitates the subsequent vesicle scission and its rapid movement away from the plasma membrane [6]. We have commented on the role of CapZ in this early stages of endocytosis in the Discussion section (page 23) of the revised manuscript. As suggested, we also rephrased our statement in the manuscript to stress that our study has focused on the role of CapZ in the maturation of early endosomes.

**Author response image 8. sa2fig8:** Control or CapZβ-knockout HeLa cells were treated with EGF-488 for the indicated times before fixation and staining with phalloidin-647 and DAPI (without permeabilization), and then processed for immunofluorescence analysis.

3. The conclusions are based on the assumption that all the Rab5 positive structures are "endosomes". This is problematic, as not all Rab5 positive structures are endosomes and it's not evident (by methodology nor the presented experiments) that the authors have taken the appropriate steps to distinguish endosomes from other organelles. It is important to compare the size distributions of EGF or Transferrin endosomes in scramble and CapZ KO cells. The authors only measure the size of Rab5 positive structures and show that Rab5 colocalization with an early endosome marker is reduced in CapZ KO cells, suggesting that the smaller endosomes may not be early endosomes.

As suggested, we measured the size distributions of EGF-positive vesicles after EGF chase in control or CapZ_β_ KO cells. Consistently, we showed that 15 min after EGF treatment, the size and number of EGF-positive vesicles in CapZ_β_ KO cells were significantly decreased and increased, respectively, when compared to the control cells (Figure 2C).

4. On EGFR: The total amount of EGFR is severely reduced after CAPZbeta KO, despite a significant reduction in its degradation. While studying the reason for this is probably beyond the scope of the present study, it would be easy and relevant to assess the total amount of mRNA of EGFR and whether reconstituting the expression of CAPZB restores the levels of EGFR. Also, in the EGFR degradation assay, it is surprising not to see the band corresponding to EGFR shifted as after stimulation with EGF, the receptors became heavily phosphorylated at multiple sites. Some clarification is needed here.

As suggested, we examined EGFR mRNA levels in control and CapZ_β_ KO HeLa cells, and showed that mRNA level of EGFR was significantly decreased in CapZ_β_ KO cells when compared to the control cells (Figure 2—figure supplement 1G). This result suggests that CapZ or F-actin dynamics positively regulate the transcription of EGFR. Along this line, actin dynamics have been shown to modulate gene transcription [7, 8]. It is of interest to further dissect the role and mechanisms of CapZ in transcription regulation in the future.

As suggested, we examined the level of EGFR in CapZ_β_-reconstituted cells, and showed that reconstituting CapZ_β_ in CapZ_β_ KO cells restores the expression of EGFR (Figure 2—figure supplement 1F).

Regarding the mobility of EGFR in SDS-PAGE after EGF treatment, we searched the literature and found that the mobility of EGFR after EGF treatment could be manifested in gradient or low bis-acrylamide gels (See Figure 2A in reference [9]). Since we run our samples in regular SDS-PAGE, we suspect this could lead to no observed mobility shift of EGFR. Similarly, no mobility shift of EGFR after EGF treatment was observed when the samples were run in regular SDS-PAGE by others (See Figure 1 in reference [10], Figure 2 in [11], etc.).

5. One explanation for the loss of localization Rab5 to EEA1 positive early endosomes in CapZ-KO cells is that there is increase in the rate of maturation of these endosomes. The authors should check if the colocalization of Rab5 with Rab7 affected by CapZ KO? Similarly, are the larger endosomes seen in the Arp2 KO cells, more positive for Rab7? This seems to be particularly true for the CK-636 treated cells in Supplemental Figure 4A, which have a dramatic Golgi-like localization.

As suggested, we examined the colocalization of Rab5 with Rab7 in control, CapZ_β_ KO, or ARP2 KO cells. Since we could not find a suitable Rab7 antibody for immunostaining, we transfected cells with Rab7-mCherry to label Rab7-positive endosomes. As shown in Figure 4—figure supplement 1C, the colocalization of Rab5 with Rab7 was significantly decreased in CapZ_β_ KO cells when compared to the control cells, suggesting that most of the accumulated Rab5-positive vesicles are immature early endosomes. As predicted by the reviewer, most of the enlarged endosomes in ARP2 KO cells are Rab7 positive. Intriguingly, some of these RAB7-positive enlarged vehicles are also RAB5 positive, manifested by the increased colocalization between Rab5 and Rab7 in ARP2 KO cells when compared to the control cells. These results suggest that the enlarged RAB5/RAB7 double-positive vesicles might be the transition vesicles between early endosomes and late endosomes, and the RAB7-only positive enlarged vesicles are late endosomes. Of note, Poteryaev et al. has shown the existence of the Rab5/Rab7 double-positive endosomes during endocytic Rab conversion in *C. elegans* Coelomocytes (see Figure 1 in ref. [12]).

6. The novel interaction between CapZ and the Rabaptin/Rabex GEF complex is the real strength of the paper. The authors have done an excellent job in validating these interactions, however the mechanism proposed of how CapZ regulates Rabaptin/Rabex GEF complex is confusing.a) The authors propose that CapZ regulates Rab5 indirectly by regulating the localization of Rabaptin-5/Rabex-5, but it's unclear if CapZ solely effects localization and/or complex formation. For example, does CapZ facilitate the formation of a functional Rabaptin-Rabex complex? Does CapZ modulate binding of the Rabaptin-Rabex complex to Rab5?

a. In Figures 5H and 5I of the revised manuscript, we showed that the interaction between Rabaptin-5 and Rabex5 was decreased in CapZ KO cells when compared to the control cells by co-IP experiments. Also, the interaction between Rabaptin-5 and Rab5 was decreased in CapZ KO cells when compared to the control cells (Figure 5H). On the other hand, knockdown of either Rabex5 (Figure 5—figure supplement 1G) or Rabaptin-5 (Figure 5—figure supplement 1H) failed to affect the interaction of CapZ with Rabaptin-5 or Rabex5, respectively. These results suggest that CapZ facilitates the formation of the Rabaptin-5/Rabex5 complex, and participates in the binding of Rabaptin-5 or Rabex5 to Rab5.

We agree with the reviewer that it is of great interest to dissect how CapZ modulates Rabex5/Rabaptin-5 GEF activity. We are currently figuring out the way to obtain recombinant Rabex5 protein from insect or mammalian cell systems since our efforts to purify the recombinant Rabex5 protein from the bacterial culture failed. We plan to test whether CapZ interacts with Rabex5 directly, to assess whether CapZ modulates the GEF activity of Rabex5/Rabaptin, and to obtain the crystal structure of CapZ/Rabex5/Rabaptin-5 complex. If successful, these results will be submitted for publication.

b) Does Rabaptin-5 and Rabex-5 colocalizes with CapZDC34 in cells? Presumably, If the interaction between CapZ with said Rab5 GEFs is independent of F-actin, then CapZDC34 should localize equally well to the Rab5 GEF complex as full-length CapZ.

b. In Figures 6F and 6G, we showed that the deletion of C34 from CapZ_β_ did not affect its interaction with Rabex5 and Rabaptin-5, but abolished its interaction with actin and markedly diminished its interaction with Rab5. Of note, CapZ-_∆_C34 exhibited a diffuse expression pattern when it was reconstituted back to CapZ KO cells (the bottom panel in Figure 6E).

c) The increase in Rab5 colocalization with the Arp2/3 complex in CapZ KO cells is quite unexpected for two reasons. (1) it contradicts the assumption that CapZ recruits Rab5 (indirectly through Rabaptin-5/Rabex-5) to these endosomal actin patches. (2) it's indicated that Rabaptin-5 (Figure 5J) is delivered by CapZ (Figure 5H) and needed to facilitate the growth of these endosomes (presumably through fusion). However, without Rab5 GEFs it's impossible to promote the positive feedback loop needed to drive endosomal fusion and/or maturation. Can the authors provide some insight into why this is the case?

c. The increased colocalization of Rab5 with Arp2/3 in CapZ KO cells compared to the control cells suggests the increased actin branches around early endosomes in CapZ KO cells (Figures 3D and 3E).

1. It has been previously shown that early endosomes are associated with F-actin [6, 13], and several proteins are required for their association, such as Annexin A2 and RNtre (a GAP for RAB5) [14, 15]. We, thus, speculate that CapZ could recruit Rabex5 and Rabaptin-5 to F-actin-associated early endosomes when it caps the barbed ends of Factin. This could bring Rabex5 and Rapaptin-5 to close proximity to RAB5 to establish a positive feedback loop to further activate Rab5 for the fusion of early endosome (see Author response image 8). Along this line, we showed that CapZ did not directly interact with RAB5 (Figure 1—figure supplement 1C in the revised manuscript). In Figures 6F and 6G, we showed that CapZ-_∆_C34 lost its ability to bind with F-actin, and only weakly interacted with Rab5. However, CapZ-_∆_C34 kept its interaction with Rabex5 and Rabaptin-5. Thus, our results do not suggest that CapZ recruits Rab5 (indirectly via Rabex5 and Rabaptin-5) to the actin branches around early endosomes.

**Author response image 9. sa2fig9:** CapZ helps recruit Rabex5 and Rapaptin-5 to RAB5-positive early endosomes, thus establishing a positive feedback loop to further activate RAB5 for the early endosome fusion.

2. In Figure 5J (the experiment suggested by the reviewer for the previous revision), we showed that Rabaptin-5 KO decreased the size and increased the number of early endosomes caused by the ARP2 KO. These results suggest that Rabaptin-5 KO inhibited the fusion of early endosomes induced by ARP2 KO. In Figure 5H, we showed that CapZ KO decreased the interaction of Rabaptin-5 with Rabex-5 and RAB5. These results suggest that the reduced interaction of Rabex-5/Rabaptin-5 with Rab5 also contributes to the inhibited early endosome maturation (manifested by the accumulation of small early endosomes) in CapZ KO cells. Thus, the role of Rabatpin5 suggested by these two figures is consistent.

d) The impact of CapZ on early endosomes could also be explained by changes in protein expression of Rab5, Rabaptin-5, Rabex-5, VPS34, etc. Please provide evidence that the expression levels of these proteins have not changed with CapZ KO.

d. In the previous rebuttal letter (Author response image 4C in point 6), we have provided evidence that CapZ KO did not affect the expression of Rab5, Rab7, VPS34, Rabex-5, and Rabaptin 5 (Author response image 10). Figures 2E, 5H, and 5I in the revised manuscript also showed that the expression of RAB5, Rebex-5, or Rabectin-5, was not affected by CapZ KO.

**Author response image 10. sa2fig10:** The expression levels of RAB5A, RAB7A, Rabex5, Rabaptin5, and VPS34 in control or CapZβ-knockout HeLa cells were determined by western blot analysis.

7. The authors need to include a discussion with how their observations fit into the wealth of data on what is known about CapZ in endocytosis and trafficking. The potential role of CapZ in endosome fusion versus fission should be discussed.

We add more information on pages 5-6 of the revised manuscript regarding the role of CapZ in various cellular processes. We also include a discussion to link our observation with the previous knowledge on CapZ in endocytosis and trafficking on page 23 of the revised manuscript.

We discussed the potential role of CapZ in endosome fusion versus fission on page 22 of the revised manuscript as following:

In the CapZ knockout cells, the number of early endosomes was significantly increased but their size was markedly smaller, when compared with the control cells (Figures 2C, 2D, Figure 2—figure supplement 1H, and Figure 3—figure supplement 1D). In addition, Rab5 activity was significantly decreased in CapZ KO cells than the contrl cells (Figure 2E). These results suggest that CapZ is required for the fusion or maturation of early endosomes. Yet, it is possible that CapZ KO might promote the fission of early endosomes, and this could also lead to the increase of small endosomes in CapZ KO cells. This possibility remains to be determined. Interestingly, Derivery et al. identified CapZ in the WASH complex [1]. They demonstrated that WASH helps to nucleate actin onto endosomes to control the fission of early endosomes for endosomal receptor recycling. They showed that WASH knockdown markedly induced tabulation of the endosomes (See Figure 5 in ref. [1]) and inhibited transferrin recycling. Although we found that CapZ knockout inhibited transferrin recycling (Figure S2F), we did not detect the long membrane tubules originating from the endosomes in the CapZ knockout cells (Figures 2D, 3C, Figure 3—figure supplement 1G and 1H). This suggests that CapZ might not participate in the endosome fission for endosomal receptor recycling.

In addition to the points mentioned above, please address the comment #4 (a-f) and #5 of reviewer 3 on the colocalization analysis and methodology and provide the complete information in the manuscript text. Please also address the minor concerns listed by reviewer #3.Reviewer #3:4. Much of the conclusions of the MS rely on colocalization analysis, which is surprisingly not at all described. This is particularly relevant to the above points. Please provide a detailed description as to how colocalization analysis was performed.a. Indicate which software package was employed (presumably Image J).

Image J software was used to perform the colocalization analysis.

b. Indicate whether the analysis was performed on single confocal plane, on Z-stack projection or in 3D? It's indicated in the rebuttal that "only single confocal plane was taken and analyzed unless otherwise stated." If this is true, indicate the criteria that was used to select this plane for analysis. Also, the methods indicate that the SIM images were reconstructed in 3D – were these image sets also analyzed in the same way as the confocal images. It's not stated how these data sets were analyzed.

All imagings were taken and analyzed on a single confocal plane unless otherwise stated in the figure legend. We chose a plane where most signals are detected, and cell morphology looks normal. Although we reconstructed SIM images in 3D, we did not perform quantification and statistical analysis on these images.

c. How were the ROIs determined? Was the entire image plane or individual cells analyzed? If on the entire image plane, how many cells were in the field of view. If performed in 3D was the ROI the same in all image planes of the stack, or was segmentation performed on each pane to identify the true boundary of the cell?d. Did the ROI exclude centrally located organelles, such as the nucleus, perinuclear region, and the Golgi? The use of Mander's to examine co-occurrence of endosomal markers is appropriate so long as it is restricted to peripheral organelles (please refer to Pike et al., Methods 2017).

We captured the single-plane image of the individual cells and analyzed the whole image. Thus no ROIs were selected in the images. There were at least 5~10 cells in the field of view (63x oil lens). Also, we did not quantify the 3D images.

Pike et al. described the workflow to analyze 3D time-lapse images. We agree that it is appropriate to exclude the nucleus region when performing the Manders colocalization coefficient analysis of endosomes markers in these 3D images.

e. Was the background subtracted? Which approach was used?

We did not perform additional background subtractions before distinguishing objects from the background using the JACoP plugin of Image J.

f. Was signal isolation performed in an automated and unbiased way, such as employing the Costes joint histogram approach?

We started with the default threshold for each image in the JACoP plugin of Image J; then we did the manual inspection to ensure that it distinguished most target objects from the background [16]. We also tried the Costes' method to set up a threshold for the image; however, in most cases, the set threshold is too low to distinguish objects from the background.

5. The endosome size data was not statistically analyzed correctly. The data is not normally distributed and showing the mean +/- SD is not appropriate for these figures. Instead, please show a box and whisker plot on top of the data points. Also note, because the data is skewed, the appropriate statistical test is the Mann-Whitney U Test. It may be more beneficial to show this data as a frequency histogram distribution.

All statistical analysis was performed by Graphpad software. We tried a box and whisker plot or a frequency histogram distribution for the endosome size data, but found that dot plots more precisely present the distribution of the endosome size data than other two methods (Author response image 11). Therefore, we selected the dot plots with the mean value present the endosome size distribution.

**Author response image 11. sa2fig11:** The dot plot, a box and whisker plot, and a frequency histogram distribution of the endosome size data in control, CapZβ-knockout, or CapZβ-reconstituted HeLa cells.

The Mann-Whitney U test is applied to compare the mean rank of data, but it does not compare means and distributions. Thus it is not always suitable for median comparison [see reference 1 below]. Although the Mann-Whitney U test could be adopted when data does not follow a Gaussian distribution, the choice of parametric vs. nonparametric is difficult, and normality tests could not simply justify it [see reference 2 below]. Since our endosome size data is large (>1000 in sample size), it matters less for choosing a parametric or nonparametric test [see reference 3 below]. In addition, a T-test is recommended for studies with large sample sizes and skewed data [see reference 4 below]. When the data might not follow a Gaussian distribution, it is also advised first to transform the data with logarithms, and then analyze the transformed values with a conventional test [see reference 5 below]. Therefore, we transformed our endosome size data with log10 before the T-test. We found that it did not change the statistical significance (p<0.05) of our results.

References for statistical analysis:

1. H. J. Motulsky, "The Mann-Whitney test doesn't really compare medians ", GraphPad Statistics Guide. Accessed 18 August 2021. https://www.graphpad.com/guides/prism/latest/statistics/stat_nonparametric_tests_dont_compa.htm

2. H. J. Motulsky, " Advice: Don't automate the decision to use a nonparametric test ", GraphPad Statistics Guide. Accessed 18 August 2021. https://www.graphpad.com/guides/prism/latest/statistics/using_a_normality_test_t o_choo.htm

3. H. J. Motulsky, " Nonparametric tests with small and large samples ", GraphPad Statistics Guide. Accessed 18 August 2021. https://www.graphpad.com/guides/prism/latest/statistics/choosing_parametric_vs_nonpar.htm

4. Fagerland, M.W., t-tests, nonparametric tests, and large studies—a paradox of statistical practice? BMC medical research methodology, 2012. 12(1): p. 1-7.

5. H. J. Motulsky, " Advice: When to choose a nonparametric test ", GraphPad Statistics Guide. Accessed 18 August 2021. https://www.graphpad.com/guides/prism/7/statistics/when_to_choose_a_nonpara metric.htm

**RERERENCES**

Derivery, E., Sousa, C., Gautier, J.J., Lombard, B., Loew, D., and Gautreau, A. (2009). The Arp2/3 activator WASH controls the fission of endosomes through a large multiprotein complex. Dev Cell *17*, 712-723.Goley, E.D., and Welch, M.D. (2006). The ARP2/3 complex: an actin nucleator comes of age. Nat Rev Mol Cell Biol *7*, 713-726.Smith, B.A., Daugherty-Clarke, K., Goode, B.L., and Gelles, J. (2013). Pathway of actin filament branch formation by Arp2/3 complex revealed by single-molecule imaging. Proc Natl Acad Sci U S A *110*, 1285-1290.Almeida, C.G., Yamada, A., Tenza, D., Louvard, D., Raposo, G., and Coudrier, E. (2011). Myosin 1b promotes the formation of post-Golgi carriers by regulating actin assembly and membrane remodelling at the trans-Golgi network. Nat Cell Biol *13*, 779-789.Mi, N., Chen, Y., Wang, S., Chen, M., Zhao, M., Yang, G., Ma, M., Su, Q., Luo, S., Shi, J., et al. (2015). CapZ regulates autophagosomal membrane shaping by promoting actin assembly inside the isolation membrane. Nat Cell Biol *17*, 1112-1123.Kaksonen, M., Toret, C.P., and Drubin, D.G. (2005). A modular design for the clathrin- and actinmediated endocytosis machinery. Cell *123*, 305-320.Olson, E.N., and Nordheim, A. (2010). Linking actin dynamics and gene transcription to drive cellular motile functions. Nat Rev Mol Cell Biol *11*, 353-365.Knoll, B. (2010). Actin-mediated gene expression in neurons: the MRTF-SRF connection. Biol Chem *391*, 591-597.Carter, R.E., and Sorkin, A. (1998). Endocytosis of functional epidermal growth factor receptorgreen fluorescent protein chimera. J Biol Chem *273*, 35000-35007.Wang, W., Zhang, H., Liu, S., Kim, C.K., Xu, Y., Hurley, L.A., Nishikawa, R., Nagane, M., Hu, B., Stegh, A.H., et al. (2017). Internalized CD44s splice isoform attenuates EGFR degradation by targeting Rab7A. Proc Natl Acad Sci U S A *114*, 8366-8371.Kortum, F., Harms, F.L., Hennighausen, N., and Rosenberger, G. (2015). alphaPIX Is a Trafficking Regulator that Balances Recycling and Degradation of the Epidermal Growth Factor Receptor. PLoS One *10*, e0132737.Poteryaev, D., Datta, S., Ackema, K., Zerial, M., and Spang, A. (2010). Identification of the switch in early-to-late endosome transition. Cell *141*, 497-508.Taunton, J., Rowning, B.A., Coughlin, M.L., Wu, M., Moon, R.T., Mitchison, T.J., and Larabell, C.A. (2000). Actin-dependent propulsion of endosomes and lysosomes by recruitment of N-WASP. J Cell Biol *148*, 519-530.Morel, E., Parton, R.G., and Gruenberg, J. (2009). Annexin A2-dependent polymerization of actin mediates endosome biogenesis. Dev Cell *16*, 445-457.Lanzetti, L., Palamidessi, A., Areces, L., Scita, G., and Di Fiore, P.P. (2004). Rab5 is a signalling GTPase involved in actin remodelling by receptor tyrosine kinases. Nature *429*, 309-314.Dunn, K.W., Kamocka, M.M., and McDonald, J.H. (2011). A practical guide to evaluating colocalization in biological microscopy. Am J Physiol Cell Physiol *300*, C723-742.Shinde, S.R., and Maddika, S. (2016). PTEN modulates EGFR late endocytic trafficking and degradation by dephosphorylating Rab7. Nat Commun *7*, 10689.

[Editors' note: further revisions were suggested prior to acceptance, as described below.]

The manuscript has been improved but there are some remaining issues that primarily require text changes, data quantification and figure modifications. These changes are outlined below in the reviewers comments:Reviewer #2:The authors have done an additional set of experiments to address the concerns expressed. Overall, the manuscript is certainly improved and at this stage, only few points to clarify might be needed.1. On the actin around the endosome. The new set of images in figure 3C is certainly a progress in this direction. Yet, it remains conceivable that the apparent increased chance of seeing an endosome adjacent to filamentous actin is simply a reflection of more cellular F-actin actin. Maybe one way to visualized actin, keeping in mind the effect exerted by CAPZ KO (which reduced the size and increase the number of endosomes) could have been treating cells with the Pik5 inhibitor. This said, at this stage, it would seem acceptable to use cautionary words on the increased F-Actin around the endosome following CAPZ genetic silencing w/o performing new experiments.

As suggested, we have rephrased the statement to “CapZ knockout increased F-actin density around RAB5-positive endosomes”, in the revised manuscript.

2. On the possible impact of CAPZ on early internalization steps. The attempts made were clearly inconclusive on the images in response to this point, it is stated that no permeabilization was done to ensure that the detected EGF would be localized at the cell surface. Specifically: "we found that staining fixed cells without permeabilization with phalloidin-647 could demonstrate plasma membrane" yet at 5 minutes there is apparently plenty of EGF signal inside the cells, which should not be detectable in the absence of permeabilization. The experiments on this issue seem inconclusive. There are more precise ways to measure the rate of endocytosis through the use of Iodinated EGF. The manuscript, however, is not focused on early steps on endocytosis, and it would seem not appropriate, at this point, to ask for these experiments. The authors were careful in commenting on the possible role of CAPZ in the early step of internalization and this might be all that is needed.

For EGF chasing experiment, the live cells were first incubated with Alexa Fluor 488–EGF on ice for 30 min, followed by PBS wash to remove the unbound ligand. The cells were then incubated with the warm medium for indicated times to trigger the endocytosis of the EGFEGFR complex before fixation. Thus, at 5 minutes, the EGF signal inside the cells was due to the internalized EGF-EGFR complex after EGF chase, not because of the staining of fixed cells. These results are presented as new Figure 2—figure supplement 1D in the revised manuscript as suggested by reviewer 3.

Reviewer #3:The additional data provided in the rebuttal when fully included, markedly helps with interpreting the roll of CapZ in regulating early endosome fusion and/or maturation. There is a clear effect of CapZ on trafficking and it seems to be driven by the assembly on Rab-5 GEF complex, Rabex-5/Rabaptin-5. The data on this point is excellent and of general interests. However, my main concern is with this insistence to say that CapZ regulates branched actin assembly on endosomes, when it's unnecessary to do so. It also calls attention to the weak point of the paper, and this conclusion is formulated by an over-interpretation, and concerning implementation and colocalization analyses, not to mention the incorrect assumption that CapZ blocks Arp2/3 complex. Indeed, recent data provided in this recent revision suggests that Arp2/3 complex regulates maturation of early endosomes, since Arp2/3 KO leads to accumulation of Rab7 positive endosomes, that were previously misinterpreted as enlarged Rab5 endosomes. This calls into question the fairness of comparing the effects of CapZ KO with Arp2/3 complex KO on endosome number and endosome size in all of Figure 4 and Figure 5J. The correct thing to do would have been restrict the endosome size distribution analysis to early endosomes (as suggested by previously by the reviewers), which would necessarily exclude any Rab5 puncta overlapping with Rab7. Alternatively, the data can be kept as is, but the authors have to strongly state that these Rab5 positive structures also includes both early and late endosomes. Furthermore, it is quite likely, and correctly mentioned by the authors, that the Arp2/3 complex plays a role in a myriad of endosome functions that affects everything from cargo sorting, fission and fusion. As such, this data cannot be used to solely conclude that CapZ KO leads to an increase in branched actin density on early endosomes that impairs their fusion.

In ARP2-knockout cells, there are both RAB7-only positive endosomes (likely the late endosomes) and RAB7-RAB5 double-positive endosomes (Figure 4—figure supplement 1C). These RAB7-RAB5 double-positive endosomes likely represent the transition vesicles between early endosomes and late endosomes [12]. These results suggest that ARP2 KO facilitates the maturation of early endosomes and/or the transition of the early-to-late endosome. Whereas in CapZ-knockout cells, most of the RAB5-positive vesicles are RAB7negative, suggesting that these RAB5-positive vesicles likely represent early endosomes. Therefore, as suggested, we clearly stated what these RAB5-only vesicles, RAB7-only vesicles, and RAB5-RAB7 double-positive vesicles represent on page 15 of the revised manuscript.

We also agree with the reviewer that Arp2/3 complex plays an important role during endosomal trafficking, from cargo sorting, fission to fusion. We have inserted a statement regarding the role of ARP2/3 in endocytosis on page 13 of the revised manuscript.

With regards to point 8.b. of the rebuttal letter, the authors state that they performed colocalization analysis on a single plane for the entire field of view, and each field of view consists of 5-10 cells. Is that correct? This would suggest extra-cellular debris and labeling would have been included in the analysis. Surely the authors did something to correct for this.

During immunostaining, all slides were washed extensively by PBS with care after primary and secondary antibodies incubation, and thus extracellular debris and signals were rare in these slides. Also, during quantification, all images were manually inspected to ensure that only intracellular signals were quantified.